# NON-EUCLIDEAN HARMONIC LOSSES

## ABSTRACT

Cross-entropy loss has long been the standard choice for training deep neural networks, yet it suffers from interpretability limitations, unbounded weight growth, and inefficiencies that can contribute to costly training dynamics. Recent work introduced *harmonic loss*, a distance-based alternative grounded in Euclidean geometry, which improves interpretability and mitigates phenomena such as *grokking*, also known as delayed generalization on the test set. However, the study of harmonic loss remains narrow: only Euclidean distance is explored, and no systematic evaluation of computational efficiency or sustainability was conducted. In this paper, we extend harmonic loss by systematically investigating a broad spectrum of distance metrics as replacements for the Euclidean distance. We comprehensively evaluate *distance-tailored harmonic losses* on both vision backbones and large language models. Our analysis is framed around a three-way evaluation of *model performance*, *interpretability*, and *sustainability*. On vision tasks, cosine distances provide the most favorable trade-off, consistently improving accuracy while lowering carbon emissions, whereas Bray-Curtis and Mahalanobis further enhance interpretability at varying efficiency costs. On language models, cosine-based harmonic losses markedly improve gradient and learning stability, strengthen representation structure, and reduce emissions relative to cross-entropy and Euclidean heads. Our code is available at: `https://anonymous.4open.science/r/rethinking-harmonic-loss-5BAB/`.

## 1 INTRODUCTION

Cross-entropy is the *de facto* loss function for classification tasks. However, it has shortcomings in terms of model interpretability and training dynamics. Cross-entropy training provides no inherent meaning to the learned weight vectors (they serve as abstract parameters rather than intuitive prototypes) and can drive those weights to grow without bound in pursuit of confident predictions Baek et al. (2025). This unbounded weight growth can lead to phenomena like *grokking*: a delayed generalization where the model only closes the train–test performance gap after extensive overtraining Power et al. (2022). Moreover, in high-stakes applications where transparency is critical (e.g., healthcare or finance), the opaque nature of cross-entropy–trained models poses challenges for trust and error diagnosis. These issues motivate the exploration of alternative loss functions that may yield more *interpretable*, *efficient*, and *robust* model behavior.

Recently, *harmonic loss* was proposed as an alternative training objective to address some of these concerns Baek et al. (2025). Harmonic loss replaces the conventional inner-product logits and softmax normalization with a distance-based formulation: model predictions are derived from the distances between the sample's representation and *class prototype* vectors (learned weight vectors for each class). Intuitively, this means that a model is trained to bring each sample closer to its correct class center in the feature space rather than simply increasing a classification score. This approach endows the learning process with two key properties: *i) scale invariance* – distance comparisons do not depend on vector norm, and *ii) finite convergence point* – training aims for a distance of zero to the correct prototype. As a result, each class weight converges to an anchor point that can be interpreted as the *center* of that class's feature distribution. Empirically, Baek et al. (2025) demonstrated that harmonic loss can close the train–test gap faster and yield more interpretable representations than cross-entropy. For example, the learned weight vectors in a harmonic-loss model directly reflect class prototypes, making them semantically meaningful. Models trained with harmonic loss were shown to require less data to generalize and to mitigate grokking, all while achieving competitive

or better accuracy on vision and language benchmarks. These findings suggest that *distance-based loss functions* are a promising direction for improving both performance and transparency in deep learning.

However, research on harmonic loss has been limited in scope so far. Baek et al. (2025) focused exclusively on Euclidean distance as the metric for their loss function and did not examine the broader impacts on computational efficiency or energy consumption. On the other hand, distance-based metrics have been explored in other contexts and problems. Notably, Coil et al. (2025) investigated a wide range of distance measures for a problem of change point detection in concept-drift scenarios for anomaly detection. Their study found that the choice of distance metric can drastically affect both the accuracy and efficiency of detecting distribution shifts. For instance, replacing a costly metric (e.g., Wasserstein) with simpler alternatives yielded comparable detection performance at substantially lower computational cost. This evidence that "metric matters" in learning algorithms raises a natural question: *might other distance measures offer advantages over Euclidean in a harmonic-loss setting?* To date, no work has evaluated harmonic loss with distance metrics beyond Euclidean, nor benchmarked their impacts across different domains.

In this paper, we present the first comprehensive study of *custom distance-based loss functions* in deep learning classification, extending the harmonic loss framework to a variety of distance measures across multiple problem domains. We experiment with a rich set of distance metrics, including Manhattan, Euclidean, Chebyshev, Minkowski, and cosine distance, as well as specialized metrics such as Hamming, Canberra, Bray-Curtis, and Mahalanobis. These metrics are integrated as drop-in replacements for Euclidean distance in the harmonic loss formulation.

We evaluate harmonic loss with each distance metric on two heterogeneous task families: *image classification* (MLP, ResNet, PVT) and *language modeling* with transformer-based LLMs (GPT-2, BERT, and others). This diversity enables us to assess whether certain distance-based losses consistently outperform cross-entropy and Euclidean harmonic loss on metrics of *effectiveness*, *efficiency*, and *explainability*. Specifically, we pursue the following research questions:

**RQ1 (Model Performance):** Do distance-based loss functions offer higher accuracy or faster convergence compared to cross-entropy and Euclidean harmonic loss?

**RQ2 (Interpretability):** Do models trained with distance-based losses exhibit more interpretable representations than those trained with cross-entropy?

**RQ3 (Efficiency & Sustainability):** If a custom distance-based loss outperforms cross-entropy, does it do so without incurring higher computational cost? We track training time, resource utilization, and energy consumption to assess the *Green AI* perspective.

By addressing these questions, our aim is to explore a *three-way trade-off* between *accuracy, interpretability, and sustainability* in the training process of deep learning models. Previous work has typically optimized one or two of these aspects in isolation: for instance, improving accuracy at the cost of enormous compute, known as "Red AI" (Schwartz et al., 2019), or simplifying models for interpretability while losing accuracy. In contrast, we seek solutions that can improve predictive performance while also yielding lower energy usage and more transparent models.

**Contributions.** This paper introduces *distance-tailored harmonic losses* and provides an extensive empirical and analytical evaluation of their merits. To our knowledge, this is the first work to: i) extend the harmonic loss beyond Euclidean distance and benchmark a wide spectrum of metrics on both vision and NLP tasks, ii) assess the carbon footprint and resource usage of different loss functions in a controlled setting, and iii) investigate interpretability outcomes of distance-based losses. We also offer preliminary theoretical insights into how different distance metrics influence the geometry of the learned model (e.g., relating $L_1$ losses to median-based class centers vs. $L_2$ to mean-based centers), which could inform the selection of an optimal loss for a given objective.

## 2 HARMONIC LOSS

Harmonic loss replaces the conventional inner-product logits and softmax normalization with a distance-based formulation: model predictions are derived from distances between the sample's representation and *class prototype* vectors (the learned weight vectors for each class). Intuitively,

this means a model is trained to bring each sample closer to its correct class center in the feature space, rather than simply pushing up a classification score.

From Baek et al. (2025), given the training set $\{(x_i, y_i)\}_{i=1}^n$ with $y_i \in \{1, ..., K\}$ and class prototypes $\{\mathbf{w}_c\}_{c=1}^K \in \mathbb{R}^d$, the harmonic logit is the $\ell_2$ distance between $\mathbf{w}_i$ and $\mathbf{x}$, i.e., $d_i = \|\mathbf{w}_i - \mathbf{x}\|_2$. Then, the harmonic probabilities are given by:

$$p_k(x_i) \;=\; \frac{d_i^{-n}}{\sum_{j=1}^K d_i^{-n}}, \tag{1}$$

where the harmonic exponent $n$ is a hyperparameter that controls the heavy-tailedness of the probability distribution. The Harmonic loss is then given by:

$$\mathcal{L}(\{\mathbf{w}_k\}) = -\sum_{i=1}^n \log p_k(x_i). \tag{2}$$

This approach endows the learning process with two key properties: i) *scale invariance:* distance comparisons do not depend on the overall norm of $\mathbf{h}$ or $\mathbf{w}_c$, in contrast to inner-product logits; and ii) *finite convergence point:* optimization seeks a distance of zero to the correct prototype.

As a result, each class weight converges to an anchor point that can be interpreted as the *center* of that class's feature distribution. Empirically, Baek et al. (2025) demonstrated that harmonic loss can close the train–test gap faster and yield more interpretable representations than cross-entropy. For example, the learned weight vectors in a harmonic-loss model directly reflect class prototypes, making them semantically meaningful. Models trained with harmonic loss were also shown to require less data to generalize and to mitigate grokking, all while achieving competitive or better accuracy on both vision and language benchmarks. These findings suggest that *distance-based loss functions* are a promising direction for improving performance and transparency in deep learning.

## 3 NON-EUCLIDEAN HARMONIC LOSSES

Our framework introduces *non-Euclidean harmonic losses* as a generalization of the harmonic loss, and as a replacement for conventional cross-entropy training. The idea is that, in Eq. (1), the Euclidean distance $d_i = \|\mathbf{w}_i - \mathbf{x}\|_2$ is replaced by a non-Euclidean distance.

### 3.1 CLASS PROTOTYPES, DISTANCES, AND DISTANCE-BASED HARMONIC LOSS FUNCTION

Each class $c \in \{1, \ldots, K\}$ is associated with a *prototype vector* $\mathbf{w}_c \in \mathbb{R}^d$. Given a sample $\mathbf{h}$, we compute its distance to all prototypes via a chosen metric $d(\cdot, \cdot)$.

Prototypes are learned parameters, just like the weight matrix in linear classification. Thus, prototype learning is no more computationally expensive than learning a final linear layer.

We extend the Euclidean formulation of harmonic loss (Baek et al., 2025) with the following distances:

**Euclidean.** $d_{\text{euclidean}}(\mathbf{h}, \mathbf{w}) = \|\mathbf{h} - \mathbf{w}\|_2$. Baseline Euclidean distance between feature and prototype.

**Manhattan (L1).** $d_{\text{manhattan}}(\mathbf{h}, \mathbf{w}) = \|\mathbf{h} - \mathbf{w}\|_1$. Emphasizes absolute differences, making it more robust to outliers (Keeling & Kunisch, 2016; Ye et al., 2012; Giloni & Padberg, 2003). It can stabilize training and reduce unnecessary computations, thereby lowering energy costs.

**Chebyshev (L∞).** $d_{\text{chebyshev}}(\mathbf{h}, \mathbf{w}) = \|\mathbf{h} - \mathbf{w}\|_\infty$. Captures the maximum coordinate deviation, offering a highly interpretable measure of the most discriminative feature dimension. Its simplicity makes it computationally efficient.

**Minkowski (Lp).** $d_{\text{minkowski}}(\mathbf{h}, \mathbf{w}; p) = \|\mathbf{h} - \mathbf{w}\|_p$. Generalizes both L1 and L2, with tunable $p$ enabling a trade-off between robustness and sensitivity. This flexibility allows tailoring the loss to dataset complexity, improving accuracy while balancing sustainability.

**Cosine.** $d_{\text{cosine}}(\mathbf{h}, \mathbf{w}) = 1 - \frac{\mathbf{h}^\top \mathbf{w}}{\|\mathbf{h}\|_2 \|\mathbf{w}\|_2}$. Ignores magnitude and instead measures angular similarity, making it particularly effective in high-dimensional embeddings (e.g., CNNs, Transform-

ers) (Reimers & Gurevych, 2019; Deng et al., 2019; Wang et al., 2018; Sun et al., 2016; Karpukhin et al., 2020). This often improves generalization with minimal computational overhead.

**Hamming.** $d_{\text{hamming}}(\mathbf{h}, \mathbf{w}) = \frac{1}{d} \sum_{i=1}^{d} \mathbf{1}_{\{h_i \neq w_i\}}$. Counts mismatches directly, providing highly interpretable signals. With soft or gumbel relaxations, it becomes suitable for continuous embeddings and can reduce emissions when binary approximations are leveraged.

**Canberra.** $d_{\text{canberra}}(\mathbf{h}, \mathbf{w}) = \sum_{i=1}^{d} \frac{|h_i - w_i|}{|h_i| + |w_i| + \varepsilon}$. Normalizes differences by feature magnitudes, enhancing sensitivity to small but meaningful variations. This can improve performance on fine-grained tasks while stabilizing optimization.

**Bray–Curtis.** $d_{\text{bray-curtis}}(\mathbf{h}, \mathbf{w}) = \frac{\sum_{i=1}^{d} |h_i - w_i|}{\sum_{i=1}^{d} (|h_i| + |w_i|) + \varepsilon}$. Captures proportional differences across feature vectors, making it efficient and interpretable for compositional data (Fuschi et al., 2025; Chao et al., 2010; Song et al., 2020). It often balances accuracy with sustainability better than covariance-based measures.

**Mahalanobis.** $d_{\text{mahalanobis}}(\mathbf{h}, \mathbf{w}; \Sigma) = \sqrt{(\mathbf{h} - \mathbf{w})^\top \Sigma^{-1} (\mathbf{h} - \mathbf{w})}$. Incorporates feature correlations, offering superior accuracy in complex datasets and deep CNNs (Pang et al., 2018; Lee et al., 2018; Gómez-Silva et al., 2021; Omara et al., 2021). Although covariance estimation may increase computational cost, its interpretability and classification power justify the trade-off in high-capacity models.

In our work, we generalize harmonic loss by replacing the Euclidean distance used to calculate the harmonic logit with some other distance measure. Harmonic loss is applied only at the final classification layer, replacing the standard softmax + cross-entropy objective. All intermediate layers remain unchanged, and no normalization is applied inside the backbone.

Overall, compared to cross-entropy, these distance-based harmonic losses reduce reliance on probabilistic normalization and can lower the number of required operations. This translates into potential accuracy gains, reduced carbon emissions, and improved interpretability, depending on the chosen distance and backbone.

A formal treatment of our distance–based probabilistic layer is provided in Appendix A. There, we generalize the harmonic-loss analysis to broad distance families and prove: i) *scale invariance* and the existence of *finite* minimizers under 1-homogeneous distances (Theorem 1), and ii) a *margin-style PAC–Bayes generalization bound* whose finiteness follows from the finite–norm solution (Theorem 2). These results clarify when geometry choices are well-posed and why the resulting classifiers admit standard generalization guarantees.

## 4 EXPERIMENTS AND DISCUSSION

### 4.1 TRAINING AND EVALUATION

**Datasets.** We evaluate on five *vision* benchmarks (MNIST, CIFAR-10, CIFAR-100, MarathiSign-Language, TinyImageNet) and one *language* corpus (OpenWebText).

**Vision.** We consider a **Simple MLP** with two hidden layers (512, 256, ReLU), a **Simple CNN** (two $3 \times 3$ conv blocks with $[32, 64]$ channels and $2 \times 2$ max-pooling, then a 128-dim FC), **ResNet-50** (standard $[3, 4, 6, 3]$ bottleneck stages; for small inputs we remove the initial max-pool and use a $3 \times 3$ stride-1 stem), and **PVTv2-B0** (four hierarchical stages with overlapping patch embeddings; output pooled to a 256-dim vector).

**Language.** We study three Transformer families: **GPT**-style (decoder-only causal LM), **BERT** (encoder-only masked LM with 15% masking), and **Qwen2**-style decoders.

**Optimization.** Unless noted, models are trained from scratch with Adam/AdamW-style optimizers (weight decay, $(\beta_1, \beta_2)$ as configured), cosine learning-rate decay with linear warmup, mixed precision (FP16/BF16 when available), and gradient accumulation. We apply gradient clipping, dataset-specific schedulers, and early stopping with *dataset-specific patience* and a minimum improvement threshold ($\Delta_{\min}$). For fairness, all harmonic heads and the baseline share the same backbone, batch size, scheduler, and data order. Additional details about optimization are reported in Appendix C.1.

**Model Performance.** For vision tasks, we report average Accuracy and F1. For language task, we report the following metrics:

**Perplexity (Train / Val).** Given a sequence of targets $\{y_t\}_{t=1}^T$ and model probabilities $p_\theta(y_t \mid \text{context})$, the average negative log–likelihood is $\mathcal{L}_{\text{NLL}} = -\frac{1}{T}\sum_{t=1}^T \log p_\theta(y_t \mid \text{context})$, and the corresponding perplexity is $\text{PPL} = \exp(\mathcal{L}_{\text{NLL}})$. Lower perplexity indicates better next–token prediction[1].

**Gradient Stability (GS).** To quantify the smoothness of optimization, we measure the variability of the $\ell_2$-norm of the gradient across consecutive training steps: $\text{GS} = 1 - \frac{\text{Var}\left(\|\nabla_\theta \mathcal{L}_t\|_2\right)}{\text{Var}\left(\|\nabla_\theta \mathcal{L}_t\|_2\right)_{\text{CE}}}$, where $\text{Var}(\cdot)$ is computed over a fixed evaluation window (e.g., 500 steps) and the denominator corresponds to the variability under cross–entropy (CE). Thus, $\text{GS} = 1$ indicates equal smoothness as CE, $\text{GS} > 1$ indicates reduced gradient variance (smoother training), and $\text{GS} < 1$ reflects more unstable gradient dynamics. This metric is anchored in standard variance-of-gradient analyses used in optimizing large-scale LLMs.

**Model Health (MH).** To track representation collapse or instability in token embeddings during pre-training, we measure the diversity of hidden representations using the per-token covariance trace: $\text{MH} = 1 - \frac{\Delta \text{Tr}(\Sigma_h)}{\Delta \text{Tr}(\Sigma_h)_{\text{CE}}}$, $\Sigma_h = \text{Cov}(\mathbf{h}_t)$. Here $\mathbf{h}_t$ denotes the hidden activations at the penultimate transformer block, and $\Delta$ denotes deviation from initialization (higher deviation often signals collapse into low-rank subspaces). $\text{MH} > 1$ indicates healthier representations (preserved diversity, no collapse), while $\text{MH} < 1$ suggests degenerate or low-rank features. This relates directly to standard metrics used in collapse detection and embedding drift.

**Interpretability.** We probe whether learned prototypes/weights act as class centers and whether features become more structured by computing **PCA explained variance** on the penultimate features: (i) **PC2 EV** (variance explained by the top two PCs), and (ii) **PCA@90%** (dimensions required to reach 90% variance). Lower PCA@90% and higher PC2 EV indicate more concentrated, low-dimensional structure. For language, we report *PCA5*: $\Delta$ variance explained by the top 5 principal components of final hidden states (causal LM: last token; MLM: masked positions); higher values implies more concentrated, low-dimensional structure.

**Sustainability.** We perform training with *CodeCarbon* to log *duration*, *energy*, and *$CO_2$ emissions*. Emissions are reported per run and *differentially* vs. the cross-entropy baseline (grams $CO_2$; negative means greener-than-baseline). We aggregate by (dataset, backbone, distance) and also report cumulative figures across seeds. For language, we also report Speed ($-\Delta$ `time_to_90_percent`): higher values denotes fewer steps to reach $90\%$ of final performance.

To isolate the effect of the *loss geometry*, we *only* swap the classifier head (linear vs. distance-based) while keeping: backbone weights initialization scheme, data preprocessing/augmentation, optimizer and LR schedule, batch size, number of epochs, early-stopping rule, and randomness controls (seeds). For ResNet-50/PVT we use identical augmentation; for LLMs we use the same context length $L$, optimizer, and schedule across heads. We run multiple seeds and report means. Exact architectures and preprocessing pipelines are detailed in Appendix C.1. Full hyperparameter grids (including head-specific parameters $\Theta$, e.g., $p$ for Minkowski or covariance settings for Mahalanobis) are provided in Appendix D. This unified protocol lets us *systematically* test how replacing the Euclidean harmonic head with alternative distances impacts: i) final model performance, ii) representation structure and prototype semantics, and iii) measured energy and carbon footprint.

Figure 1 summarizes the behavior of distance-based harmonic losses across all vision settings, including a high–resolution sign–language dataset (Marathi Sign) and TinyImageNet in addition to CIFAR-100. Additional results on MNIST and CIFAR10 are provided in Appendix F. Together, these radar plots expose how the choice of distance in the harmonic head shapes performance, representation geometry, and sustainability.

---

[1]For visualization in the radar plots, we invert perplexity (and all metrics where lower values indicate better performance) and then normalize to the range $[0, 10]$ relative to the harmonic Euclidean baseline. The effect is that the greater the coverage on the plot, the better the relative performance compared to Euclidean. Importantly, the absolute numeric values on the radial axis do not have a direct "good/bad" interpretation in perplexity space; they are only meaningful as normalized, experiment–specific comparisons against Euclidean harmonic.

**RQ1: Model Performance (F1, Accuracy).** Across datasets and backbones, **cosine–based harmonic losses** remain the most reliable all–round performers. On CIFAR-100, cosine (stable/unstable) typically attains the highest or near–highest accuracy and F1 on CNN and ResNet50, and is consistently among the top curves on PVT. On the more realistic, higher–resolution Marathi Sign and TinyImageNet, the same pattern largely persists: cosine (stable) and Bray–Curtis (normalized) frequently improve or match Euclidean and cross–entropy on CNN, ResNet50, and PVT, while also appearing in the top group on MLP. TinyImageNet is the most challenging setting: here, cross–entropy remains a strong baseline, but cosine heads still achieve competitive accuracy on ResNet50 and PVT, demonstrating that the benefits of distance–tailored heads extend beyond small benchmarks. Other non–Euclidean distances (Bray–Curtis variants, Manhattan, Minkowski) can occasionally match or exceed cosine in specific architecture–dataset combinations.

**RQ2: Interpretability (PC2 EV, PCA 90%).** Non–Euclidean distances reshape the final embedding geometry in a systematic, dataset–agnostic way. Across Marathi Sign, TinyImageNet, and CIFAR-100, Bray–Curtis (standard/normalized) and Chebyshev (standard) repeatedly yield the largest PC2 explained variance and the lowest dimensionality required to reach 90% EV, indicating compact, prototype–aligned feature spaces with sharper class clusters than those produced by Euclidean harmonic loss or cross–entropy. Cosine harmonic loss generally provides substantial EV gains over Euclidean while retaining top accuracy, offering a favorable accuracy–interpretability balance on both convolutional backbones and PVT. Mahalanobis variants often achieve extreme variance concentration (very high EV) and pronounced cluster separation, but this representation clarity sometimes co–occurs with less stable optimization on the hardest datasets. Overall, the same geometric trends observed on earlier small benchmarks persist when moving to higher resolutions and deeper models: non–Euclidean harmonic losses, especially Bray–Curtis and Chebyshev, produce more structured, low–dimensional embeddings than Euclidean or cross–entropy heads.

**RQ3: Sustainability (Duration/Epoch/GFLOPs, Emissions).** Distance choice also affects efficiency, but in a controlled way. Across all datasets, cosine harmonic loss is typically neutral–to–favorable in emissions relative to Euclidean and cross–entropy: normalized Duration/Epoch/GFLOPs and $gCO_2eq$ remain comparable, and in several ResNet50 and PVT runs cosine achieves slightly lower emissions due to faster approach to high accuracy. Bray–Curtis losses incur modest overhead while delivering strong interpretability gains, whereas Mahalanobis distances are the most costly, reflecting their covariance–related computation and sometimes slower convergence on complex data. Even on high–resolution Marathi Sign and TinyImageNet, the harmonic head contributes only a small fraction of total FLOPs; thus, differences in Duration/Epoch are smaller than differences in accuracy or EV, yet cumulative emissions still separate distances meaningfully.

Across all vision workloads, three regularities emerge: i) cosine harmonic loss is the best all–around choice, offering consistently strong accuracy/F1, clear geometric structure relative to Euclidean, and neutral–to–lower emissions from MLPs up to ResNet50/PVT on Marathi Sign and TinyImageNet; ii) Bray–Curtis and Chebyshev are the most interpretability–forward options, reliably increasing variance concentration and reducing PCA 90% dimensionality, with accuracy effects that are positive but more configuration–dependent; iii) Mahalanobis emphasizes representation clarity at a higher sustainability cost. Taken together, the radar plots show that the geometry of the harmonic loss, especially non–Euclidean choices, has a consistent, architecturally robust effect on performance, structure, and sustainability across both small and large vision benchmarks.

## 4.2 LANGUAGE: RADAR PLOTS

Figure 2 summarizes the effect of distance-tailored harmonic losses on *BERT*, *GPT*, and *Qwen*-style decoders across the three perspectives. Scores are normalized so that larger areas indicate more desirable behavior.

**RQ1: Model Performance (Perplexity, Health, Stability).** Across architectures, cosine–based harmonic losses remain the most reliable choices on performance–oriented axes. For BERT, cosine heads achieve low train and validation perplexity while improving Gradient Stability and preserving high Model Health relative to both cross–entropy and Euclidean harmonic loss. For GPT, cosine and Minkowski ($p$=2) again provide steady training dynamics with competitive perplexity, whereas the cross–entropy baseline exhibits higher variability and weaker stability. On Qwen, Euclidean harmonic loss offers the strongest combination of low perplexity and gradient stability, with

Minkowski providing a close alternative; the cross–entropy head is consistently dominated on at least one of these axes. Overall, replacing the linear classifier with distance–based harmonic heads reduces gradient volatility and collapse symptoms while maintaining or improving perplexity.

**RQ2: Interpretability (PCA Structure).** Non–Euclidean distances consistently concentrate token representations into more structured latent spaces. In BERT and GPT, cosine and Minkowski enlarge the PCA Structure wedge (higher variance explained by a small number of components), indicating more organized, prototype–aligned embeddings than those produced by cross–entropy or Euclidean harmonic loss. Qwen shows a similar pattern: distance–based heads achieve clearer low–dimensional structure even when Euclidean is slightly stronger on stability. As in the vision experiments, geometries that emphasize angles (cosine) or $\ell_p$ structure (Minkowski) tend to yield hidden states that are easier to summarize with a few principal components.

**RQ3: Sustainability (Emissions).** Results confirm that distance–based harmonic heads introduce little computational overhead and can be greener than cross–entropy in practice. In all three models, the cross–entropy baseline occupies the largest emissions wedge, while cosine and Minkowski are neutral–to–favorable, often matching or improving on Euclidean harmonic loss. Extremely sharp cosine temperatures may reduce emissions slightly but at the cost of stability and perplexity; moderate settings avoid this trade–off. Because the classifier head is lightweight compared to the Transformer backbone, these sustainability gains primarily arise from smoother optimization and faster convergence rather than per–step FLOPs.

In summary, cosine–based harmonic losses are the most robust all–around choice for LLMs, jointly improving perplexity, stability, and representation structure with neutral or reduced emissions. Minkowski ($p$=2) provides a strong alternative when cosine hyperparameters are poorly tuned, while Euclidean remains a solid reference but is rarely dominant over non–Euclidean geometries. Additional results showcasing optimization dynamics for all models, including a larger GPT2 (2B) model, are reported in Appendix H.

## 5 RELATED WORK

**Loss functions for classification.** The majority of classification models are trained with cross-entropy loss due to its empirical effectiveness and probabilistic interpretation. However, it only cares about separating classes, not about how the representations are separated, often yielding features that are separable but not necessarily interpretable. Over the years, alternative loss functions have been proposed to address these limitations. Metric learning losses, such as contrastive and triplet loss, train models to preserve distances between examples, but require sampling strategies that add training complexity. Boudiaf et al. (2020) propose a unifying mutual information framework connecting cross-entropy to standard pairwise losses, showing that cross-entropy implicitly bounds pairwise distance objectives. These insights motivate a deeper theoretical understanding of distance-based training. Regularization-based approaches such as *center loss* (Wen et al., 2016) explicitly encourage compact intra-class clusters and large inter-class separation. These works foreshadow the idea that directly leveraging distances to class prototypes can improve representation quality. Angular margin losses such as AMC-Loss in Choi et al. (2020) introduce geometric constraints on angular separations to enhance interpretability via hyperspherical metrics. Orthogonal Projection Loss (OPL) introduced by Ranasinghe et al. (2021) encourages inter-class orthogonality and intra-class cohesion without sampling overhead. Several studies have assessed how loss functions affect neural network performance. Miller et al. (2021) introduce *Class Anchor Clustering* (CAC) loss that encourages tight class clusters centered on anchored prototypes, enhancing distance-based open-set classification performance. This approach aligns with the prototype-centered philosophy underlying harmonic loss. Cho et al. (2019) analyzed how eight loss functions impact neural network accuracy and convergence speed, finding that additive-margin softmax loss resulted in the fastest convergence and highest performance on multiple datasets. Janocha & Czarnecki (2017) assessed 12 loss functions for classification, finding that choice of loss function impacted learning speed and testing accuracy. Gonzalez & Miikkulainen (2020) used genetic programming to develop Baikal loss, which not only led to networks achieving higher accuracy than networks trained with cross-entropy loss, but also faster training and higher performance in low-data settings. These studies demonstrate a large focus on the impact of loss function on neural networks performance. Our work builds on the discussion of the importance of loss function choice by drilling deeper on harmonic loss, exam-

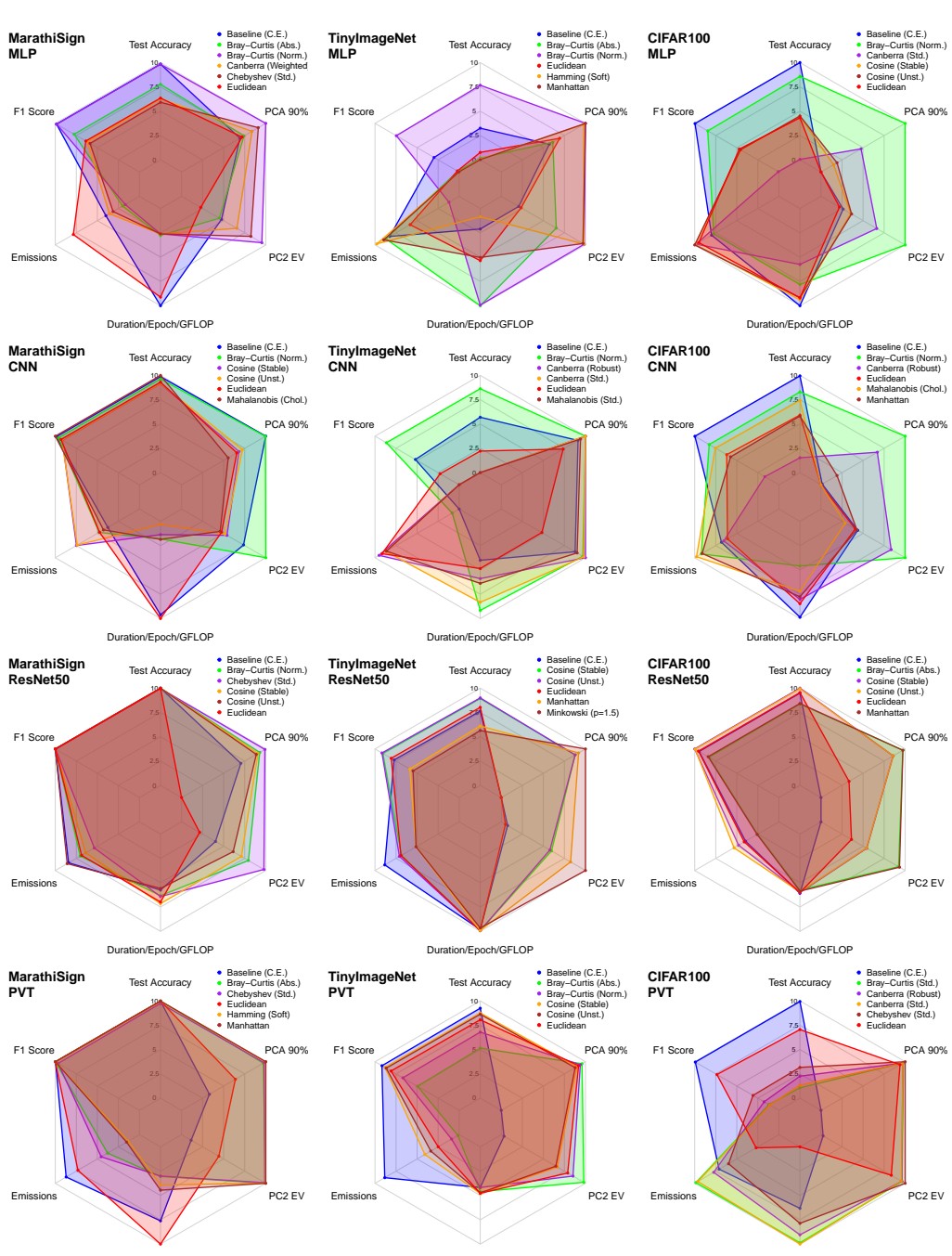

Figure 1: Vision: Radar plots: 1) *Model Performance* (F1, Accuracy); 2) *Interpretability* (PC2 EV, PCA 90%), and 3) *Sustainability* (Duration/Epoch/GFLOPs, Emissions). Plots feature Baseline (Cross-Entropy), Euclidean harmonic, and the four top-performing non-Euclidean harmonic losses.

ining how distance metric choice impacts the effectiveness of neural networks. Our focus is not on comparing harmonic loss with other loss functions, which was done by Baek et al. (2025), but rather to shed light on performance of a generalized harmonic loss.

**Efficiency and Green AI.** Green AI is an emerging initiative that calls for efficiency and energy usage to be treated as first-class evaluation criteria (Schwartz et al., 2019). Many works on green AI focus on model compression (Paula et al., 2025; Rafat et al., 2023), comparing multiple models

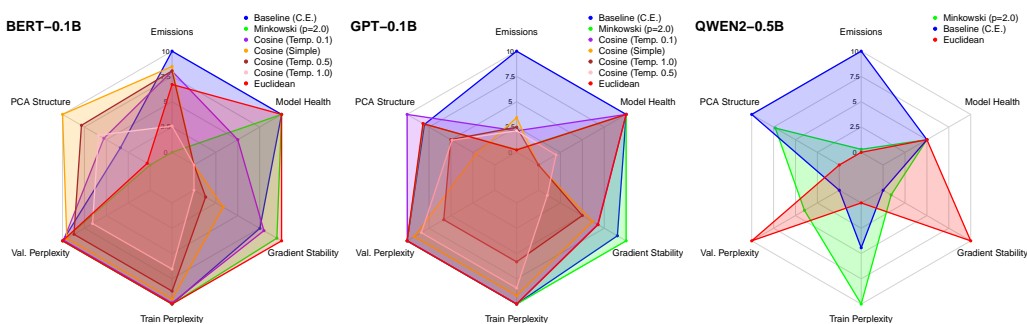

Figure 2: Language: Radar plots: 1) *Model Performance* (Perplexity, Model Health, Gradient Stability); 2) *Interpretability* (PCA5 EV), and 3) *Sustainability* (Emissions). Plots feature Baseline (CE), Euclidean harmonic, and the top-performing non-Euclidean harmonic losses.

(Verma et al., 2024) or fine-tuning strategies (Wang et al., 2023), or hyperparameter optimization for carbon emission reduction Wang et al. (2025). While prior works on new loss functions rarely report sustainability metrics, we incorporate carbon footprint analysis into our evaluation due to claims that models trained with harmonic loss are more data efficient and have less grokking (Baek et al., 2025).

**Interpretability in neural networks.** Neural networks are complex and not inherently interpretable, but a substantial amount of effort was done to improve interpretability (Zhang et al., 2021). The push for *interpretable by design* models argues that transparency should be built into model architectures and losses rather than added post-hoc (Rudin, 2019). Harmonic loss aligns with this vision by structurally linking model weights to class prototypes. The study by Saphra et al. (2024) discusses how internal model components reveal human-understandable circuits and features in LLMs. Techniques such as activation patching, sparse autoencoders, transcoders, and crosscoders enable structural interpretations of model behavior. Parallel to our interpretability focus, Wen et al. (2025) introduced *InterpGN*, a framework combining interpretable models with deep networks for time-series tasks, preserving understandable reasoning where possible. Though not loss-centric, it reflects the growing emphasis on transparency in deep learning research. Some work has focused on using loss functions specifically to improve model interpretability. Liu et al. (2022) combine sparse coding constraints with cross-entropy to produce concise, interpretable word-level attributions. Dong et al. (2017) introduced *interpretative loss* to improve interpretability of learned features during video captioning tasks. Within classification tasks, Zhang et al. (2018) designed a loss function to improve CNN filter interpretabiltiy. Methods such as the one proposed by Hagos et al. (2023) augment standard losses with distance-based penalties that align model attributions with user-provided annotations, strengthening interpetability.

**Distance metrics in learning algorithms.** Beyond supervised classification, the choice of distance measure is known to be crucial. Coil et al. (2025) compared twelve distance metrics in anomaly detection for concept drift. Their results highlighted that performance depends heavily on the chosen metric and that efficient alternatives can sometimes match the performance of more costly distances. A variety of other works have shown the importance of distance metric choice. Amaya-Tejera et al. (2024) used a kernel for SVMs that could support a variety of kernels, finding that distance metric choice impacted performance. Kalra et al. (2022) and Hu et al. (2016) both found that distance metric choice impacted performance of *k*-nearest neighbors algorithms on a variety of datasets. These result highlights the importance of systematically exploring metrics in different contexts. To our knowledge, our paper is the first to bring this perspective into loss functions.

## 6 CONCLUSION

This work examined *distance–based harmonic losses* as drop–in replacements for cross–entropy across image classification (MNIST, CIFAR-10, CIFAR-100, Marathi Sign Language, TinyImageNet) with four vision backbones (MLP, CNN, ResNet50, PVT) and LLM pretraining (GPT, BERT, Qwen, GPT-2B), leveraging a broad family of distances (cosine, Euclidean, Bray–Curtis,

Mahalanobis, Minkowski, Chebyshev, Canberra, *etc.*) and comparing them against strong modern baselines (Focal Loss, Label Smoothing, Center Loss, Confidence Penalty, ArcFace).

**What we learned.** i) **Geometry matters for optimization.** Across vision and language tasks, Cosine consistently delivers smoother training dynamics, higher or competitive final performance, and reduced grokking–like behavior on toy modulo–addition experiments. Euclidean remains a solid reference; Bray–Curtis is often competitive but architecture–sensitive; Mahalanobis exhibits the largest variance—sometimes yielding very sharp, well–separated clusters, but with less stable plateaus on the more difficult scenarios (larger datasets and model backbones). Loss–convergence curves for both vision and LLMs show that all investigated distances (including cosine and Mahalanobis) are characterized by a smooth optimization without problematic instabilities.

ii) **Sustainability depends jointly on distance and architecture.** On vision tasks, several non–Euclidean harmonic losses are carbon–negative per step relative to cross–entropy for CNN/ResNet50 (largest gains occur on deeper CNNs), mixed on MLP, and closer to neutral on PVT and TinyImageNet, where backbone FLOPs dominate. For LLM pretraining, the classifier head is lightweight, so differences arise primarily via convergence: the cross–entropy baseline typically incurs the largest cumulative emissions, while cosine and Minkowski heads are neutral–to–favorable. Our FLOPs–normalized analysis and extended emissions study show that the best non–Euclidean harmonic losses lie on or near the sustainability–accuracy Pareto frontier.

iii) **Interpretability can be quantified.** PCA–based probes (variance concentration and PCA@90%) and geometric visualizations of prototype neighborhoods provide reproducible evidence that distance–tailored heads yield more structured representations. Bray–Curtis and Chebyshev consistently increase variance concentration and reduce intrinsic dimensionality, while Mahalanobis emphasizes representation clarity at higher computational cost. These trends hold for image features and for token representations in LLMs (last–token and masked–token states), and are supported by statistical tests (Wilcoxon) and confidence intervals across seeds.

**Language.** Cosine–based harmonic losses markedly improve gradient/learning stability, perplexity, and representation structure for GPT, BERT, Qwen, and GPT-2B, while keeping emissions on par with or below cross–entropy and Euclidean heads. Mahalanobis remains less attractive for large–scale pretraining due to covariance overheads and sensitivity to ill–conditioned statistics.

**Vision.** For accuracy–focused workloads across MNIST, CIFAR, Marathi Sign, and TinyImageNet, cosine (stable) is the preferred all–round choice; Bray–Curtis is a strong secondary option; Mahalanobis should be used when its inductive bias (sharp, anisotropic clusters) is explicitly desired. For sustainability on CNN/ResNet50, several non–Euclidean distances reduce per–step $CO_2$; on PVT and LLMs, the lightest geometries (cosine/Euclidean) should be favored, or cross–entropy retained unless a distance–based head reduces steps–to–target enough to offset higher per–step cost.

Beyond specific winners, our main contribution is a *framework*: a plug–and–play harmonic head, a catalogue of distances, and a three–axis evaluation protocol (*performance, interpretability, sustainability*) with concrete metrics, visualizations, and statistical tests. This framework can be effectively exploited in future work: practitioners can choose distances according to their priorities and researchers can extend our study to new geometries, learning settings, and domain–specific constraints. In this sense, distance–based harmonic losses provide a principled, empirically validated toolbox for rethinking the geometry of classifier heads in both vision and language models.

**Reproducibility Statement.** We took several steps to facilitate exact and statistical reproducibility. The main paper specifies the learning objectives, training protocol, model families, and evaluation metrics used in all studies. The *Appendix* contains: i) complete hyperparameter and backbone–specific settings; ii) dataset descriptions and end-to-end preprocessing pipelines (including splits and any filtering); iii) detailed experimental studies and analyses; iv) technical details with code snippets to integrate our non-Euclidean harmonic losses in conventional deep learning pipelines. Our code repository provides: ready-to-run scripts for data acquisition and preprocessing; configuration files for every experiment; training/evaluation entry points; instructions for reproducing results. Together, these materials are intended to enable independent researchers to audit, rerun, and extend our findings with minimal effort.

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

APPENDIX

# A THEORETICAL PROPERTIES OF DISTANCE-BASED PROBABILISTIC LAYERS

**Setup.** Let $\{(x_i, y_i)\}_{i=1}^n$ be the training set with $y_i \in \{1, ..., K\}$. Each class has a prototype $w_k \in \mathbb{R}^d$ and a nonnegative distance $d(x, w) \geq 0$. Given a decreasing link $\kappa : \mathbb{R}_+ \to \mathbb{R}_+$ we define

$$p_k(x_i) \;=\; \frac{\kappa\big(d(x_i, w_k)\big)}{\sum_{j=1}^K \kappa\big(d(x, w_j)\big)}, \qquad \mathcal{L}(\{w_k\}) \;=\; -\sum_{i=1}^n \log p_{y_i}(x_i).$$

*harmonic* $\kappa(r) = r^{-\omega}$ with $\omega > 0$; while distances include Euclidean/Mahalanobis, $\ell_p$, Bregman divergences, and cosine/angle on the sphere.

## A.1 SCALE INVARIANCE AND FINITE MINIMIZERS

We begin by generalizing the finite-minimizer result of the harmonic loss (cf. Thm. 1, Sec. G in Baek et al. (2025)).

**Definition 1** (Metric separability and homogeneity). A dataset is *metric-separable* if for each $i$ there exists $\{w_k\}$ s.t. $d(x_i, w_{y_i}) < \min_{j \neq y_i} d(x_i, w_j)$. A distance $d$ is *1-homogeneous* if $d(cx, cw) = |c| \, d(x, w)$ for all $c > 0$.

**Theorem 1** (Finite minimizer and scale invariance for harmonic link). *Assume $d$ is 1-homogeneous and the training set is metric-separable. For $\kappa(r) = r^{-\omega}$, the empirical loss $\mathcal{L}$ is invariant to the joint rescaling $(x, w) \mapsto (cx, cw)$ and attains a global minimum at finite $\{w_k\}$. In particular, increasing $\|w_k\|$ further does not reduce $\mathcal{L}$.*

*Proof.* Following the proof of Sec. G Thm. 1 in Baek et al. (2025), the probabilities remain unchanged under uniform scaling for any 1-homogeneous distance $d$. For the probabilities, if we replace $x_i$ by $c_x i$ and $w_j$ with $cw_j$, then $d(cx_i, cw_j) = c \, d(x_i, w_j)$, so the scaling factors cancel when using a harmonic link $\kappa$. Therefore, once the correct classification is achieved, no further reduction in loss is obtained by increasing $\|w_k\|$ and the loss achieves a global minimum at a finite $\{w_k\}$. $\square$

## A.2 MARGIN-STYLE GENERALIZATION (PAC-BAYES VIEW)

Sec. G gives a PAC-Bayes margin bound that is finite because the harmonic solution has finite norm (Thm. 2) in Baek et al. (2025).

**Definition 2** (Distance margin). Given prototypes $W = \{w_k\}$, define $\gamma(W) = \min_i \big[ d(x_i, w_{y_i}) - \min_{j \neq y_i} d(x_i, w_j) \big]$.

**Theorem 2** (Generalization with metric margin). *Assume all $x_i$ lie in a ball of radius $R$ (in the native norm of $d$ or its inducing space). Let $\|W\|_\star$ denote a capacity measure compatible with $d$. With probability at least $1 - \delta$, the generalization error of the classifier satisfies*

$$\Pr_{(x,y)} \big[ h_W(x) \neq y \big] \;=\; \mathcal{O}\!\left( \frac{R \, \|W\|_\star}{\gamma(W)\sqrt{n}} + \sqrt{\frac{\log(1/\delta)}{n}} \right),$$

*where $h_W(x) = \arg\max_k p_k(x)$ denotes the predicted class and $n$ is the number of training samples. For the harmonic link, $\|W\|_\star$ is finite by Thm. 1, yielding a finite bound (cf. Sec. G Thm. 2) in Baek et al. (2025).*

*Proof.* Mirroring the proof for Sec. G Thm. 2 in Baek et al. (2025), applying the standard PAC-Bayes margin bounds, one obtains that with at least probability $1 - \delta$,

$$\Pr_{(x,y)} \left[ h_W(x) \neq y \right] = \mathcal{O}\left( \frac{R \|W\|_\star}{\gamma(W)\sqrt{n}} + \sqrt{\frac{\log(1/\delta)}{n}} \right).$$

Since $\|W\|_\star$ is finite by Thm. 1, the bound is finite. $\qquad\qquad\square$

## B INTEGRATION INTO DEEP LEARNING PIPELINES

The `DistLayer` abstraction highlights that distance-based harmonic loss functions are highly modular and can be seamlessly integrated into existing deep learning pipelines. The `forward` method requires only three operations: i) computing pairwise distances between sample embeddings and class prototype weights, ii) clamping values for numerical stability, and iii) applying a softmin via `log_softmax` to obtain normalized class probabilities. This makes the substitution of Euclidean distance with alternative metrics essentially a one-line change in the distance registry, with no modifications required in the broader training loop.

Several design choices make the implementation robust. First, all distance functions are implemented in a vectorized form, ensuring GPU efficiency and avoiding explicit loops. Second, numerical safeguards (e.g., $\varepsilon$-offsets, clamping before roots and divisions, regularization of covariance matrices) prevent instability across diverse datasets and architectures. Third, the registry-based design allows new distance functions to be added without disrupting the existing workflow, reinforcing the flexibility of harmonic loss as a general framework.

From a methodological perspective, this implementation highlights one of the key contributions of this work: the ease of replacing cross-entropy with distance-based harmonic loss. Unlike cross-entropy, which relies on unbounded logit growth, the harmonic formulation treats classification as a problem of minimizing distances to interpretable class prototypes. The plug-and-play nature of the `DistLayer` demonstrates that alternative geometries (e.g., cosine, Mahalanobis, Bray–Curtis) can be explored at negligible engineering cost, paving the way for systematic evaluation of accuracy, sustainability, and interpretability across diverse tasks.

```python
class DistLayer(nn.Module):
    """Final classification head using harmonic loss: logits =
    ↪   -distance."""
    def __init__(self, in_features, n_classes, dist_name="euclidean",
    ↪   **dist_kwargs):
        super().__init__()
        self.W = nn.Parameter(torch.empty(n_classes, in_features))
        nn.init.kaiming_uniform_(self.W, a=5**0.5)
        self.dist_name   = dist_name
        self.dist_fn     = DIST_REGISTRY[dist_name]
        self.dist_kwargs = dist_kwargs     # e.g., p for minkowski,
        ↪   cov_inv for mahalanobis

    def forward(self, h):
        """
        h: (B, D) features from backbone.
        Returns log-probs for harmonic loss:  log_softmax(-distance).
        """
        d = self.dist_fn(h, self.W, **self.dist_kwargs)    # (B, C)
        d = torch.clamp(d, min=1e-6, max=1e6)     # general safety clamp
        logits = -d                               # softmin over distances
        return F.log_softmax(logits, dim=-1)
```

```python
import torch
import torch.nn as nn
import torch.nn.functional as F

def _pairwise(fn):
    """Lift a vector distance fn(h, w) -> scalar into a batched
    ↪ pairwise form."""
    def lifted(h, W):
        # h: (B, D), W: (C, D) -> (B, C)
        h_exp = h.unsqueeze(1)      # (B, 1, D)
        W_exp = W.unsqueeze(0)      # (1, C, D)
        return fn(h_exp, W_exp)
    return lifted

def euclidean(h, W, eps=1e-4):
    diff = h - W
    return torch.sqrt(torch.clamp((diff * diff).sum(-1) + eps,
    ↪ min=eps))

def manhattan(h, W, eps=1e-4):
    return (h - W).abs().sum(-1) + eps

def cosine(h, W, eps=1e-6, stable=True):
    if stable:
        h_n = F.normalize(h, p=2, dim=-1)
        W_n = F.normalize(W, p=2, dim=-1)
        cos = (h_n * W_n).sum(-1)
    else:
        num = (h * W).sum(-1)
        den = torch.clamp(h.norm(dim=-1) * W.norm(dim=-1) + eps,
        ↪ min=eps)
        cos = num / den
    return 1.0 - cos + eps

def minkowski(h, W, p=1.5, eps=1e-6):
    diff = torch.clamp((h - W).abs() + eps, min=eps)
    dist_p = torch.clamp(diff.pow(p).sum(-1) + eps, min=eps)
    return dist_p.pow(1.0 / p)

def chebyshev(h, W, eps=1e-6, smooth=False, alpha=10.0):
    diff = (h - W).abs()
    if smooth:
        # soft-max norm
        return torch.logsumexp(alpha * diff, dim=-1) / alpha + eps
    return diff.max(dim=-1).values + eps

def canberra(h, W, eps=1e-4, variant="standard", min_denom=1e-3,
             weight_power=1.0, normalize_weights=True):
    num = (h - W).abs()
    den = h.abs() + W.abs() + eps
    if variant == "robust":
        den = torch.clamp(den, min=min_denom)
    if variant == "weighted":
        w = (den.pow(weight_power))
        if normalize_weights:
            w = w / (w.sum(-1, keepdim=True) + eps)
        return (w * (num / den)).sum(-1) + eps
    return (num / den).sum(-1) + eps
```

```python
def bray_curtis(h, W, eps=1e-3, variant="standard", min_sum=1e-3):
    num = (h - W).abs().sum(-1)
    if variant == "abs":
        den = (h.abs() + W.abs()).sum(-1)
    else:  # standard/normalized
        den = (h + W).sum(-1).abs()
    den = torch.clamp(den + eps, min=10 * eps, max=1e6)
    return torch.clamp(num / den + eps, min=eps, max=1e6)

def mahalanobis(h, W, eps=1e-6, cov_inv=None, reg_lambda=1e-2):
    # h: (B, 1, D), W: (1, C, D) expected (use _pairwise wrapper)
    diff = h - W  # (B, C, D)
    try:
        if cov_inv is None:
            # Identity with mild regularization
            return torch.sqrt(torch.clamp((diff * diff).sum(-1) + eps,
                ↪ min=eps))
        cov_inv_reg = cov_inv + torch.eye(cov_inv.size(0),
            ↪ device=cov_inv.device) * reg_lambda
        diff_M = torch.einsum('bcd,dd->bcd', diff, cov_inv_reg)
        dist2  = (diff_M * diff).sum(-1)
        return torch.sqrt(torch.clamp(dist2 + eps, min=eps))
    except Exception:
        # Safe fallback: Euclidean
        return torch.sqrt(torch.clamp((diff * diff).sum(-1) + eps,
            ↪ min=eps))
```

## C  MODEL ARCHITECTURES

### C.1  VISION

We detail the architectures of the vision models used in our experiments – including a simple MLP, a small CNN, ResNet-50, and PVTv2-B0 – specifying their layers and neuron counts for reproducibility. All models were implemented in PyTorch, and for distance-based variants, the final fully-connected layer is replaced by a specialized distance layer as noted below.

**MLP**: **Input Layer:** Accepts the flattened image input (e.g., $28 \times 28 = 784$ features for MNIST, $32 \times 32 \times 3 = 3072$ for CIFAR). **Hidden Layer 1:** Fully-connected layer with 512 neurons, followed by ReLU. **Hidden Layer 2:** Fully-connected layer with 256 neurons, followed by ReLU. **Output Layer:** Linear mapping from 256 units to the number of classes (10 for MNIST/CIFAR-10, 100 for CIFAR-100). In _DIST variants, this layer is replaced with a distance-based classification head (e.g. Euclidean, cosine) that computes distances between the embedding and class prototypes, outputting *negative distances* as logits.

**CNN**: **Conv Layer 1:** 2D convolution, 32 filters, kernel size $3 \times 3$, padding 1, followed by ReLU, then $2 \times 2$ max pooling. **Conv Layer 2:** 2D convolution, 64 filters, kernel size $3 \times 3$, padding 1, followed by ReLU, then $2 \times 2$ max pooling. **Fully Connected Layer:** Flattened output fed into a 128-unit linear layer with ReLU. **Output Layer:** Linear layer mapping the 128-D representation to the number of classes. In _DIST variants, this is replaced by a distance metric layer.

**ResNet-50**: **Stem:** Standard $7 \times 7$ convolution with 64 filters and stride 2, batch norm, ReLU, then $3 \times 3$ max pooling. For CIFAR/MNIST, we use a $3 \times 3$ conv with stride 1 and remove max pooling. **Stage 1:** 3 bottleneck blocks, output 256 channels. **Stage 2:** 4 bottleneck blocks, output 512 channels. **Stage 3:** 6 bottleneck blocks, output 1024 channels. **Stage 4:** 3 bottleneck blocks, output 2048 channels. **Global Pooling and Output:** Global average pooling yields a 2048-D vector. In the baseline, a linear FC layer maps to logits. In _DIST variants, the FC is replaced by a distance layer (e.g. cosine similarity) that outputs similarity-based logits.

**Pyramid Vision Transformer (PVTv2-B0)**: **Stage 1:** Overlapping patch embedding with a $7 \times 7$ conv (stride 4), output 32 channels, followed by 2 Transformer encoder layers (1 attention head).

**Stage 2:** $3 \times 3$ conv (stride 2), output 64 channels, followed by 2 encoder layers (2 heads). **Stage 3:** $3 \times 3$ conv (stride 2), output 160 channels, followed by 2 encoder layers (5 heads). **Stage 4:** $3 \times 3$ conv (stride 2), output 256 channels, followed by 2 encoder layers (8 heads). **Global Pooling and Output:** Global average pooling yields a 256-D vector. A linear classifier maps to the number of classes in the baseline, while in _DIST variants this is replaced with a distance layer producing log-similarity or negative distance scores.

**Preprocessing Pipelines**: **MNIST:** For MLP/CNN, grayscale input normalized to mean 0.5, std 0.5. For ResNet, normalization uses dataset statistics (mean 0.1307, std 0.3081). For PVT, grayscale converted to 3 channels, resized to 32 (PVT), normalized to mean/std 0.5. **CIFAR-10:** Normalization with mean (0.4914, 0.4822, 0.4465) and std (0.2023, 0.1994, 0.2010). ResNet uses data augmentation (random flips, crops, small rotations). **CIFAR-100:** Normalization with mean (0.5071, 0.4867, 0.4408) and std (0.2675, 0.2565, 0.2761). Stronger augmentation (random flips, crops, rotations, color jitter). PVT models use $32 \times 32$ resized inputs with normalization.

## C.2 LLMs

This section documents the **LLM** configurations used in our experiments for reproducibility. We report data preprocessing, architectural details for **GPT**, **BERT**, and **Qwen2**-style models, how **distance-based heads** are integrated in place of the standard linear classifier, and the training/evaluation/emissions-logging pipeline. All models are implemented in PyTorch and trained with mixed precision when available.

**Data and Preprocessing Corpus and Storage.** We pre-process a text corpus into contiguous token ID arrays and store them as memory-mapped files:

- `train.bin` and `val.bin`: `np.memmap` arrays of type `uint16` containing token IDs.
- `meta.pkl`: contains metadata including `vocab_size` (used to configure model embeddings).

Let $V$ denote the discovered vocabulary size from `meta.pkl` (fallback $V{=}50304$ if not found).

**Batching.** For a given `block_size` $L$, batches are sampled by picking random starting indices and slicing $L$ tokens:

$$X = \text{data}[i : i+L], \quad Y = \text{data}[i+1 : i+1+L] \quad \text{(causal LM)}$$

All batching is performed on-device with pinned memory. We denote `batch_size` by $B$.

**Masking for MLM (BERT).** For BERT runs, we construct masked language modeling (MLM) batches with the standard 15% corruption:

- Select $\approx 15\%$ token positions per sequence to form mask indices $\mathcal{M}$.
- For each $i \in \mathcal{M}$: with 80% probability replace $x_i$ with [MASK] (id $\leq 103$ or capped by $V{-}1$), with 10% replace by a random token in $[0, V)$, with 10% keep $x_i$ unchanged.
- Labels use the original token at masked positions and $-100$ (ignore index) elsewhere.

This yields `input_ids`, `attention_mask` (all-ones here), and `labels` containing ground-truth only at masked positions.

**Architectures** Across models below, the principal hyperparameters are:

layers ($n_\ell$), heads ($n_h$), embedding dim ($d$), context length ($L{=}$`block_size`), vocab size ($V$).

Unless otherwise specified, positional encodings follow each model's default (e.g., learned or rotary).

GPT2 (CAUSAL LM)

**Backbone.** A standard decoder-only Transformer with $n_\ell$ blocks. Each block has:

- Multi-Head Causal Self-Attention with $n_h$ heads, hidden size $d$, and causal mask.

- Position-wise MLP of width typically $\approx 4d$ with nonlinearity (e.g., GELU).
- Pre/post LayerNorm and residual connections as in GPT-style decoders.

**Token Embeddings.** Learnable token and (implicit) position embeddings of sizes $V \times d$ and $L \times d$ (or rotary embeddings if enabled). **Projection Head (baseline).** A linear layer $W_{\text{lm}} \in \mathbb{R}^{d \times V}$ producing logits over the vocabulary at each position. **Distance Head (_DIST).** The linear projection is replaced by a *distance-based layer* that treats the vocabulary columns as *prototypes* $\{w_v \in \mathbb{R}^d\}_{v=1}^V$. Given a hidden state $h_t \in \mathbb{R}^d$, the head returns per-token logits $z_{t,v} = -D(h_t, w_v; \Theta)$ (or $\log S(h_t, w_v)$ for similarity-type layers), where $D(\cdot, \cdot; \Theta)$ is one of the distances defined in the main text (Euclidean, cosine, Manhattan, Minkowski, Canberra, Bray–Curtis, Chebyshev, Mahalanobis, Hamming). This integrates seamlessly with the causal LM objective (next-token prediction via softmax over $V$).

## BERT (MASKED LM)

**Backbone.** An encoder-only Transformer with $n_\ell$ layers, each with:

- Multi-Head Self-Attention (bidirectional) with $n_h$ heads.
- Position-wise MLP, LayerNorm, residual connections.

**Embeddings.** Token embeddings $V \times d$, segment/type embeddings (size 2), and positional embeddings of length $L$. **Head (baseline).** The standard MLM classifier projects $d \to V$ (optionally via an intermediate nonlinearity tied to the embedding matrix). **Distance Head (_DIST).** We replace the MLM classifier with the same prototype-based distance layer used for GPT, but applied *only at masked positions*. For each masked token representation $h_i$, logits are $z_{i,v} = -D(h_i, w_v; \Theta)$ (or log-similarity), and cross-entropy is computed against the ground-truth token at $i$.

## QWEN2-STYLE DECODER (CAUSAL LM)

**Backbone.** A decoder-only Transformer similar to GPT, with model-specific details:

- Rotary Position Embeddings (RoPE) with $\theta$ (e.g., $\theta = 10^6$).
- RMSNorm with $\epsilon$ (e.g., $10^{-6}$) in place of LayerNorm.
- Grouped key/value heads: `num_key_value_heads` may be $< n_h$.
- Intermediate MLP width (`intermediate_size`) configurable.

**Vocabulary.** By default, we use Qwen's native vocabulary (`vocab_size`=151,936); alternatively, one can adapt to the dataset vocab. **Head (baseline vs. _DIST).** As with GPT, the final projection is either a linear layer to $V$ or a distance-based head over $V$ prototype vectors.

## DISTANCE-BASED OUTPUT LAYER

For all three families (GPT, BERT/MLM, Qwen2), the baseline $d \to V$ classifier is replaced in `_DIST` runs by a distance head:

$$
z_v(h) = \begin{cases}
- \|h - w_v\|_2 & \text{(Euclidean)} \\
- \|h - w_v\|_1 & \text{(Manhattan)} \\
- \|h - w_v\|_p & \text{(Minkowski, } p \text{ specified)} \\
- \left(1 - \frac{h^\top w_v}{\|h\|_2 \|w_v\|_2}\right) & \text{(Cosine)} \\
- D_{\text{Canberra}}(h, w_v) \text{ or } - D_{\text{Bray–Curtis}}(h, w_v) & \text{(variants as defined)} \\
- \|h - w_v\|_\infty & \text{(Chebyshev)} \\
- \sqrt{(h - w_v)^\top \Sigma^{-1}(h - w_v)} & \text{(Mahalanobis, variants)} \\
- D_{\text{Hamming}}(h, w_v) & \text{(soft/Gumbel/hard)}
\end{cases}
$$

where $w_v$ are learned prototype vectors (analogous to classifier weights). We adopt the numerically robust implementations given in the main text (e.g., small $\varepsilon$, clamping, optional normalization of $h$ and/or $w_v$ where appropriate). For cosine, we may output log-similarities for stability. Loss is standard cross-entropy over the $V$ logits per position (causal) or per masked position (MLM).

TRAINING SETUP AND OPTIMIZATION

**Device and Precision.** We use `bfloat16/float16/float32` (configurable) with automatic mixed precision:

$$\texttt{torch.autocast(device\_type='cuda', dtype=ptdtype)}.$$

Training can run in single-GPU or **DDP** (`torch.distributed`) multi-GPU mode. In DDP, `LOCAL_RANK` selects the device, and gradients are synchronized across ranks.

**Initialization and Checkpointing.** Models are initialized *from scratch* using the specified architecture config (layers, heads, width, $L$, $V$). For GPT-only runs we optionally support `init_from='gpt2*'`, and for BERT we support `init_from='bert*'` (when provided), with appropriate overrides. Checkpoints store model/optimizer state, `iter_num`, `best_val_loss`, and the configuration.

**Optimizer and LR Schedule.** We use the model's `configure_optimizers` helper to instantiate an Adam/AdamW-style optimizer with weight decay and $(\beta_1, \beta_2)$. Learning rate follows cosine decay with warmup:

$$\text{lr}(t) = \begin{cases} \text{lr}_{\max} \cdot t/\text{warmup} & t < \text{warmup}, \\ \text{lr}_{\min} + \frac{1}{2}\Big(1 + \cos\frac{\pi(t-\text{warmup})}{T-\text{warmup}}\Big)\big(\text{lr}_{\max} - \text{lr}_{\min}\big) & t \leq T, \end{cases}$$

where $T$ is `lr_decay_iters`. We apply gradient accumulation (`gradient_accumulation_steps`), optional gradient clipping (`grad_clip`), and AMP scaling (`GradScaler`).

**Objectives.**

- **GPT/Qwen2 (causal LM):** next-token cross-entropy over $V$ at each position.
- **BERT (MLM):** cross-entropy computed only at masked positions; non-masked labels set to $-100$ (ignored).

Accuracy reporting: we compute token-level accuracy for monitoring (on next-token for causal LM, on masked tokens for MLM).

**Evaluation and Early Signals.** At fixed `eval_interval`, we run `estimate_loss()` over `eval_iters` batches on train/val splits (model in `eval()`), then resume training. Best validation loss checkpoints are saved; optional compile (`torch.compile`) can be enabled.

SUSTAINABILITY TRACKING

We integrate *CodeCarbon* to measure energy and emissions. At each evaluation interval:

1. Stop the tracker and record interval-level metrics: emissions (kg $CO_2$), duration, estimated CPU/GPU/RAM power and energy.
2. Log cumulative emissions and training metrics (loss, lr) to W&B (if enabled).
3. Restart the tracker for the next interval to avoid long-running file locks and to attribute emissions to training phases cleanly.

At the end of training, we stop the tracker one final time and persist all accumulated records to a CSV (`emissions_*.csv`) alongside model checkpoints.

**Key Configuration Knobs (Reproducibility)** The following knobs are saved in run configs/checkpoints and should be reported alongside results:

$$(n_\ell, \, n_h, \, d, \, L, \, V), \quad \text{distance head type and parameters } (\Theta), \quad \text{batch size } B, \, \text{precision},$$
$$\text{optimizer \& betas}, \, \text{lr schedule (warmup, } T, \, \text{lr}_{\max}, \, \text{lr}_{\min}), \, \text{grad accumulation},$$
$$\text{grad clip}, \, \text{DDP world size}.$$

When using `_DIST` variants, we additionally report which distance (Euclidean, cosine, Manhattan, Minkowski($p$), Canberra, Bray–Curtis, Chebyshev, Mahalanobis, Hamming), any normalization/scaling flags, and regularization choices (e.g., Mahalanobis covariance learning/regularization).

In all models, the sole architectural change introduced by harmonic loss is confined to the **output head**: a drop-in replacement of the linear classifier with a **distance-based prototype head** over the vocabulary. This isolates the effect of the loss geometry while keeping the Transformer backbone (and training recipe) unchanged, enabling controlled comparisons across distances in terms of *accuracy*, *interpretability* (e.g., PCA-based analyses), and *sustainability* (emissions and runtime).

## D  HYPERPARAMETER CONFIGURATIONS

The hyperparameter settings in Tables 1–8 were chosen to balance *comparability*, *training stability*, and *sustainability*. Below we highlight several important considerations.

### D.1  LANGUAGE MODELS (OPENWEBTEXT)

Table 1 specifies the core training parameters for GPT, BERT, and Qwen on OpenWebText. The main goal was to maintain a fair comparison across models of varying scale by using effective batch sizes of similar order (76–128). This ensures that any differences observed in performance or emissions are attributable to the *loss formulation*, not simply to batch scaling. The use of AdamW with default $\beta$ values (0.9, 0.999) follows current best practices for stability.

Table 2 details architecture-specific modifications. BERT includes type embeddings and a masked language modeling (MLM) setup, while GPT and Qwen use causal language modeling (CLM). Qwen, being substantially larger, incorporates more advanced design elements such as grouped query attention (GQA) and rotary position embeddings (RoPE). Table 3 summarizes these differences: GPT and Qwen follow causal objectives, while BERT relies on bidirectional context, which may affect the degree to which distance-based losses interact with their representations.

Table 1: Core configuration for GPT, BERT, Qwen, and GPT-2B on OpenWebText.

| Configuration | GPT | BERT | Qwen | GPT-2B |
|---|---|---|---|---|
| $n_{\text{layer}}$ | 12 | 12 | 24 | 48 |
| $n_{\text{head}}$ | 12 | 12 | 14 | 20 |
| $n_{\text{embd}}$ | 768 | 768 | 896 | 1600 |
| Vocab size | 50304 | 50304 | 151936 | 50304 |
| Dropout | 0.1 | 0.1 | 0.0 | 0.1 |
| Bias | True | True | True | True |
| Batch size | 16 | 38 | 6 | 3 |
| Grad. accum. steps | 8 | 2 | 10 | 21 |
| Effective batch size | 128 | 76 | 60 | 63 |
| Learning rate | 2e-4 | 1e-4 | 1e-4 | 1e-4 |
| Warmup iters | 500 | 1000 | 1000 | 1000 |
| Weight decay | 0.01 | 0.01 | 0.01 | 0.01 |
| Grad clip | 1.0 | 1.0 | 1.0 | 1.0 |
| Min LR | 2e-6 | 1e-6 | 1e-6 | 1e-6 |
| Decay LR | True | True | True | True |
| LR decay iters | 10000 | 10000 | 10000 | 10000 |
| Max iters | 10000 | 10000 | 10000 | 10000 |
| Dataset | OpenWebText | OpenWebText | OpenWebText | OpenWebText |
| dtype | bfloat16 | bfloat16 | bfloat16 | bfloat16 |
| Optimizer | AdamW | AdamW | AdamW | AdamW |
| $\beta_1, \beta_2$ | 0.9, 0.999 | 0.9, 0.999 | 0.9, 0.999 | 0.9, 0.999 |
| Eval interval | 1000 | 1000 | 1000 | 500 |
| Eval iters | 100 | 100 | 100 | 25 |
| Log interval | 50 | 50 | 50 | 25 |
| Scale attn by inverse layer idx | False | False | False | False |

Table 2: Architecture-specific settings for GPT, BERT, Qwen, and GPT-2B.

| Configuration | GPT | BERT | Qwen | GPT-2B |
|---|---|---|---|---|
| Block size / Seq length | 1024 | 512 | 1024 | 512 |
| Type vocab size | – | 2 | – | – |
| Pad token id | – | 0 | – | – |
| MLM probability | – | 0.15 | – | – |
| Intermediate size | – | – | 4864 | – |
| # key–value heads | – | – | 2 | – |
| RMSNorm $\epsilon$ | – | – | 1e-6 | – |
| RoPE $\theta$ | – | – | 1,000,000.0 | – |

Table 3: Key differences summary (task and position encoding).

| Aspect | GPT | BERT | Qwen | GPT-2B | |
|---|---|---|---|---|---|
| Model size (approx.) | $\sim$124M | $\sim$110M | $\sim$494M | $\sim$2B | CLM = Causal |
| Attention | Causal | Bidirectional | Causal (GQA) | Causal | |
| Training task | CLM | MLM | CLM | CLM | |
| Position encoding | Learned | Learned | RoPE | Learned | |

Language Modeling; MLM = Masked Language Modeling; GQA = Grouped Query Attention.

## D.2 VISION MODELS

Tables 4–7 provide the vision settings across datasets. As shown in Table 4, optimizer and learning-rate schedules are backbone-specific: Adam for MLPs and CNNs, AdamW for transformers (PVT), and SGD with momentum for ResNet50. This reflects both convention and empirical stability in preliminary experiments. Batch size selection (Table 5) reflects hardware utilization on H100 GPUs. Notably, lightweight backbones (e.g., CNNs) leverage very large batches (up to 8192 for MNIST), while transformer-based PVT is limited to much smaller batches (128–256) to fit memory constraints. These design choices affect emissions profiles: large-batch training can reduce wall-clock time but at the cost of GPU memory overhead. Learning-rate schedulers differ across models. For example, PVT employs cosine annealing, which smooths convergence and interacts well with distance-based loss formulations. ResNet50 relies on multi-step decay, ensuring stability across the long 200-epoch training horizon on CIFAR-100.

**Distance Layer Parameters.** Table 8 summarizes the shared hyperparameters across all distance functions. The exponent $n$ is fixed to 1.0 and $\varepsilon = 10^{-4}$ provides numerical stability. Importantly, distances are not scaled post hoc, ensuring that differences in results are directly attributable to the geometric properties of the chosen distance (Euclidean, Manhattan, Mahalanobis, etc.), rather than to auxiliary tuning.

## D.3 DISCUSSION

**Language models.** GPT and BERT use comparable depth/width with learned positional encodings, while Qwen is larger, adopts RoPE, and GQA. Effective batch sizes (via gradient accumulation) normalize throughput across models for fair comparison on OpenWebText.

**Vision models.** Optimizer and scheduler choices follow common practice: Adam/AdamW for MLP/CNN/PVT, SGD with momentum for ResNet50; deeper/longer CIFAR-100 runs employ stepped or cosine schedules. Early-stopping patience scales with dataset difficulty.

**DistLayer defaults.** A unified setting ($n{=}1.0$, $\varepsilon{=}10^{-4}$, no scaling) ensures distance variants differ only in geometry, not in auxiliary hyperparameters. These settings match the configuration used in our main experiments and figures.

Table 4: Core training configuration by backbone and dataset.

| Configuration | MLP | CNN | PVT | ResNet50 |
|---|---|---|---|---|
| LR (MNIST) | 3e-4 | 3e-4 | 1e-3 | 0.1 |
| LR (CIFAR-10) | 3e-4 | 3e-4 | 1e-3 | 0.1 |
| LR (CIFAR-100) | 3e-4 | 3e-4 | 5e-4 | 0.1 |
| LR (MarathiSign) | 1.5e-4 | 1.5e-4 | 5e-4 | 0.05 |
| LR (TinyImageNet) | 3e-4 | 3e-4 | 5e-4 | 0.1 |
| Epochs (MNIST) | 40 | 40 | 80 | 100 |
| Epochs (CIFAR-10) | 40 | 40 | 80 | 100 |
| Epochs (CIFAR-100) | 150 | 150 | 150 | 200 |
| Epochs (MarathiSign) | 50 | 50 | 100 | 75 |
| Epochs (TinyImageNet) | 100 | 100 | 200 | 150 |
| Optimizer | Adam | Adam | AdamW | SGD |
| Weight decay | 0 | 0 | 0.01 | 1e-4 |
| Momentum | – | – | – | 0.9 |

Table 5: Batch size configuration on H100 GPU.

| Model | MNIST | CIFAR-10 | CIFAR-100 | MarathiSign | TinyImageNet |
|---|---|---|---|---|---|
| MLP | 2048 | 1024 | 1024 | 128 | 256 |
| CNN | 8192 | 4096 | 512 | 512 | 512 |
| PVT | 256 | 512 | 256 | 64 | 128 |
| ResNet50 | 512 | 512 | 256 | 128 | 256 |

# E   STATISTICAL SIGNIFICANCE AGAINST EUCLIDEAN HARMONIC LOSS.

## E.1   WILCOXON SIGNED-RANK TESTS

To quantify whether non–Euclidean harmonic losses differ systematically from the Euclidean reference, we ran paired Wilcoxon signed–rank tests over all dataset–backbone combinations ($N=16$ pairs per distance). The resulting median score improvements and $p$–values are reported in Tables 9–11.

On *model performance* (Table 9), the non–Euclidean distances do not achieve a statistically significant *positive* median improvement over the Euclidean harmonic loss. Several metrics (e.g., Mahalanobis (Std.), Bray–Curtis (Std.), Canberra variants, Manhattan, Minkowski, Hamming) show significant *negative* medians ($p < 0.05$), indicating that when a difference is present it tends to favor the Euclidean reference in raw accuracy. This is consistent with our main results, where non–Euclidean geometries target interpretability and sustainability rather than headline accuracy gains.

For *interpretability* (Table 10), we observe the opposite pattern. Distances such as Bray–Curtis (Norm.), Canberra (Robust/Std.), Chebyshev (Std.), Manhattan, and both cosine variants exhibit statistically significant shifts in the number of principal components needed to explain $90\%$ of the variance. The median differences are large in magnitude (e.g., $+12.8$ for Bray–Curtis (Norm.), $+9.8$ for Canberra (Robust)), confirming that switching away from Euclidean induces a consistent and substantial change in representation geometry across datasets and backbones.

For *sustainability* (Table 11), four distances reach $p < 0.05$: Mahalanobis (Std.) and Bray–Curtis (Std.) with positive medians, and Canberra (Weighted) and Mahalanobis (Chol.) with negative medians. This suggests that the carbon footprint differences between Euclidean and most non–Euclidean harmonic losses are modest and model–dependent: some geometries slightly increase emissions, others slightly decrease them, but strong systematic effects are rare once we fix backbone, data, and training budget.

Overall, these nonparametric tests support our main claims: non–Euclidean harmonic losses do not uniformly dominate Euclidean in accuracy, but several of them induce statistically significant changes in representation structure, with only mild and mixed effects on emissions.

Table 6: Learning-rate schedulers by backbone and dataset.

| Model | MNIST | CIFAR-10 | CIFAR-100 | MarathiSign | TinyImageNet |
|---|---|---|---|---|---|
| MLP | None | None | StepLR (50, 0.5) | ReduceLR* | StepLR (50, 0.5) |
| CNN | None | None | StepLR (50, 0.5) | ReduceLR* | StepLR (50, 0.5) |
| PVT | CosineAnn. (80) | CosineAnn. (80) | CosineAnn. (150) | CosineAnn. (100) | CosineAnn. (200) |
| ResNet50 | StepLR (30, 0.1) | StepLR (30, 0.1) | MultiStep** | StepLR (25, 0.1) | MultiStep*** |

*ReduceLROnPlateau (mode=max, factor=0.5, patience=5, min_lr=1e-6)
**MultiStepLR (milestones=[60,100,140], $\gamma$=0.2)
***MultiStepLR (milestones=[80,120], $\gamma$=0.2)

Table 7: Dataset metadata and early-stopping settings (vision).

| Parameter | MNIST | CIFAR-10 | CIFAR-100 | MarathiSign | TinyImageNet |
|---|---|---|---|---|---|
| Num classes | 10 | 10 | 100 | 43 | 200 |
| Early stopping patience | 15 | 15 | 25 | 10 | 15 |
| Min improvement (%) | 0.01 | 0.01 | 0.01 | 0.01 | 0.01 |
| Native image size | $28\times28\times1$ | $32\times32\times3$ | $32\times32\times3$ | varies | $64\times64\times3$ |
| Processed size (MLP/CNN/PVT) | $28\times28\times1$ | $32\times32\times3$ | $32\times32\times3$ | $32\times32\times3$ | $224\times224\times3$ |
| Processed size (ResNet50) | $28\times28\times1$ | $32\times32\times3$ | $32\times32\times3$ | $224\times224\times3$ | $224\times224\times3$ |

E.2 ACCURACY WITH CONFIDENCE INTERVALS.

To complement the aggregate tables and Wilcoxon tests, Figure 3 reports accuracy curves for the top-performing losses on each dataset/backbone pair. For every setting we re-train each candidate with three random seeds and plot the mean trajectory together with a shaded $95\%$ confidence interval ($n$=3).

Across datasets and architectures, two consistent patterns emerge. First, the ranking suggested by our radar plots and summary tables is preserved under multi-seed training: distance-based harmonic losses that previously appeared as strong contenders (e.g., cosine, Bray–Curtis, Minkowski) continue to track at least as well as, and often above, the cross-entropy and Euclidean baselines throughout training. In several regimes (notably CIFAR-10/CIFAR-100 with ResNet50 and MNIST with ResNet50/PVT), the confidence bands of the leading non-Euclidean harmonic loss lie systematically above those of the baselines in the later epochs, indicating that the final accuracy gains are not artifacts of seed choice but persist under sampling noise.

Second, the width of the confidence intervals is often comparable or smaller for harmonic losses than for standard baselines. On datasets where optimization is more fragile (e.g., CIFAR-100 with PVT), cross-entropy and some regularized baselines (Focal, Center Loss) display visibly wider bands and occasional late-epoch fluctuations, whereas harmonic distances yield smoother trajectories with tighter intervals, echoing our gradient stability findings. Importantly, we do not observe any case where a harmonic loss that outperforms Euclidean in the aggregate tables suffers a reversal when confidence intervals are taken into account.

Overall, these multi-seed curves provide statistical depth to our vision experiments: performance improvements for non-Euclidean harmonic losses are accompanied by tight, stable confidence bands, supporting the claim that their advantages over Euclidean and cross-entropy are robust rather than due to random initialization.

Table 8: Distance-layer shared parameters (all backbones).

| Parameter | Value |
|---|---|
| $n$ | 1.0 |
| $\varepsilon$ | 1e-4 |
| Scale distances | False |

Table 9: Wilcoxon signed–rank test comparing each non–Euclidean harmonic loss against the Euclidean harmonic baseline on *average final test accuracy*. Median score improvement is the median paired difference (non–Euclidean minus Euclidean) across $N$ dataset–backbone combinations. Positive values indicate that the non–Euclidean distance attains higher accuracy; the last column marks tests with $p < 0.05$.

| Comparison | N Pairs | Median Impr. | $p$-value | $p_{\text{adj}}$ | Sig. ($p < 0.05$) |
|---|---|---|---|---|---|
| Bray–Curtis (Norm.) | 16 | 0.5317 | 0.17060 | 1.0000 | |
| Cosine (Stable) | 16 | 0.1598 | 0.45339 | 1.0000 | |
| Cosine (Unst.) | 16 | 0.0965 | 0.55208 | 1.0000 | |
| Mahalanobis (Chol.) | 16 | 0.0400 | 0.58717 | 1.0000 | |
| Bray–Curtis (Abs.) | 16 | -1.1967 | 0.10335 | 1.0000 | |
| Mahalanobis (Diag.) | 16 | -1.4633 | 0.08323 | 1.0000 | |
| Chebyshev (Std.) | 16 | -1.7232 | $< 0.001$ | 0.0736 | Yes |
| Minkowski ($p$=3.0) | 16 | -1.8183 | 0.00567 | 0.6234 | Yes |
| Canberra (Weighted) | 16 | -2.8167 | 0.00295 | 0.3565 | Yes |
| Manhattan | 16 | -4.9467 | 0.00249 | 0.3083 | Yes |
| Minkowski ($p$=1.5) | 16 | -5.8117 | 0.00176 | 0.2232 | Yes |
| Hamming (Soft) | 16 | -6.3183 | 0.00348 | 0.4110 | Yes |
| Canberra (Robust) | 16 | -16.5883 | $< 0.001$ | 0.0736 | Yes |
| Canberra (Std.) | 16 | -18.3767 | $< 0.001$ | 0.0736 | Yes |
| Chebyshev (Smooth) | 16 | -21.3817 | $< 0.001$ | 0.0736 | Yes |
| Bray–Curtis (Std.) | 16 | -34.4379 | $< 0.001$ | 0.0975 | Yes |
| Mahalanobis (Std.) | 16 | -64.8082 | $< 0.001$ | 0.0736 | Yes |

Table 10: Wilcoxon signed–rank test comparing each non–Euclidean harmonic loss against the Euclidean harmonic baseline on *average intrinsic dimension* (number of PCs required to reach 90% EV). Median score improvement is again the median paired difference (non–Euclidean minus Euclidean); here more negative values correspond to fewer required components.

| Comparison | N Pairs | Median Impr. | $p$-value | $p_{\text{adj}}$ | Sig. ($p < 0.05$) |
|---|---|---|---|---|---|
| Bray–Curtis (Norm.) | 16 | 12.8333 | $< 0.001$ | 0.1142 | Yes |
| Canberra (Robust) | 16 | 9.8333 | 0.00162 | 0.2086 | Yes |
| Canberra (Std.) | 16 | 8.1667 | $< 0.001$ | 0.0988 | Yes |
| Chebyshev (Std.) | 16 | 7.7083 | 0.01864 | 1.0000 | Yes |
| Manhattan | 16 | 7.6667 | 0.00412 | 0.4764 | Yes |
| Cosine (Unst.) | 16 | 5.8333 | 0.01043 | 1.0000 | Yes |
| Canberra (Weighted) | 16 | 5.2083 | 0.26768 | 1.0000 | |
| Cosine (Stable) | 16 | 4.3333 | 0.01127 | 1.0000 | Yes |
| Mahalanobis (Std.) | 16 | 4.0417 | 0.12716 | 1.0000 | |
| Bray–Curtis (Std.) | 16 | 2.1667 | 0.34869 | 1.0000 | |
| Bray–Curtis (Abs.) | 16 | 1.0000 | 0.66019 | 1.0000 | |
| Hamming (Soft) | 16 | 0.7083 | 0.77730 | 1.0000 | |
| Minkowski ($p$=1.5) | 16 | -0.0000 | 0.80665 | 1.0000 | |
| Minkowski ($p$=3.0) | 16 | -0.0000 | 0.75554 | 1.0000 | |
| Mahalanobis (Diag.) | 16 | -0.0000 | 0.30656 | 1.0000 | |
| Chebyshev (Smooth) | 16 | -0.2083 | 0.77638 | 1.0000 | |
| Mahalanobis (Chol.) | 16 | -0.3333 | 0.85062 | 1.0000 | |

Table 11: Wilcoxon signed–rank test comparing each non–Euclidean harmonic loss against the Euclidean harmonic baseline on *average emissions* ($gCO_2eq$). Median score improvement is the median paired difference (non–Euclidean minus Euclidean); negative values indicate lower emissions than the Euclidean reference.

| Comparison | N Pairs | Median Impr. | $p$-value | $p_{adj}$ | Sig. ($p < 0.05$) |
|---|---|---|---|---|---|
| Mahalanobis (Std.) | 16 | 1.8037 | $< 0.001$ | 0.114 | Yes |
| Bray–Curtis (Std.) | 16 | 1.1880 | 0.01620 | 1.000 | Yes |
| Cosine (Unst.) | 16 | 0.2341 | 0.05249 | 1.000 | |
| Canberra (Std.) | 16 | 0.0351 | 0.77611 | 1.000 | |
| Chebyshev (Smooth) | 16 | 0.0208 | 0.73679 | 1.000 | |
| Canberra (Robust) | 16 | -0.0206 | 0.73679 | 1.000 | |
| Cosine (Stable) | 16 | -0.0239 | 0.97937 | 1.000 | |
| Minkowski ($p$=1.5) | 16 | -0.0492 | 0.36552 | 1.000 | |
| Hamming (Soft) | 16 | -0.0909 | 0.26625 | 1.000 | |
| Bray–Curtis (Abs.) | 16 | -0.1066 | 0.20520 | 1.000 | |
| Chebyshev (Std.) | 16 | -0.1421 | 0.14056 | 1.000 | |
| Bray–Curtis (Norm.) | 16 | -0.1625 | 0.11477 | 1.000 | |
| Manhattan | 16 | -0.1976 | 0.12716 | 1.000 | |
| Minkowski ($p$=3.0) | 16 | -0.4158 | 0.05249 | 1.000 | |
| Mahalanobis (Diag.) | 16 | -0.4587 | 0.05249 | 1.000 | |
| Canberra (Weighted) | 16 | -0.7687 | 0.03188 | 1.000 | Yes |
| Mahalanobis (Chol.) | 16 | -0.9431 | 0.00411 | 0.476 | Yes |

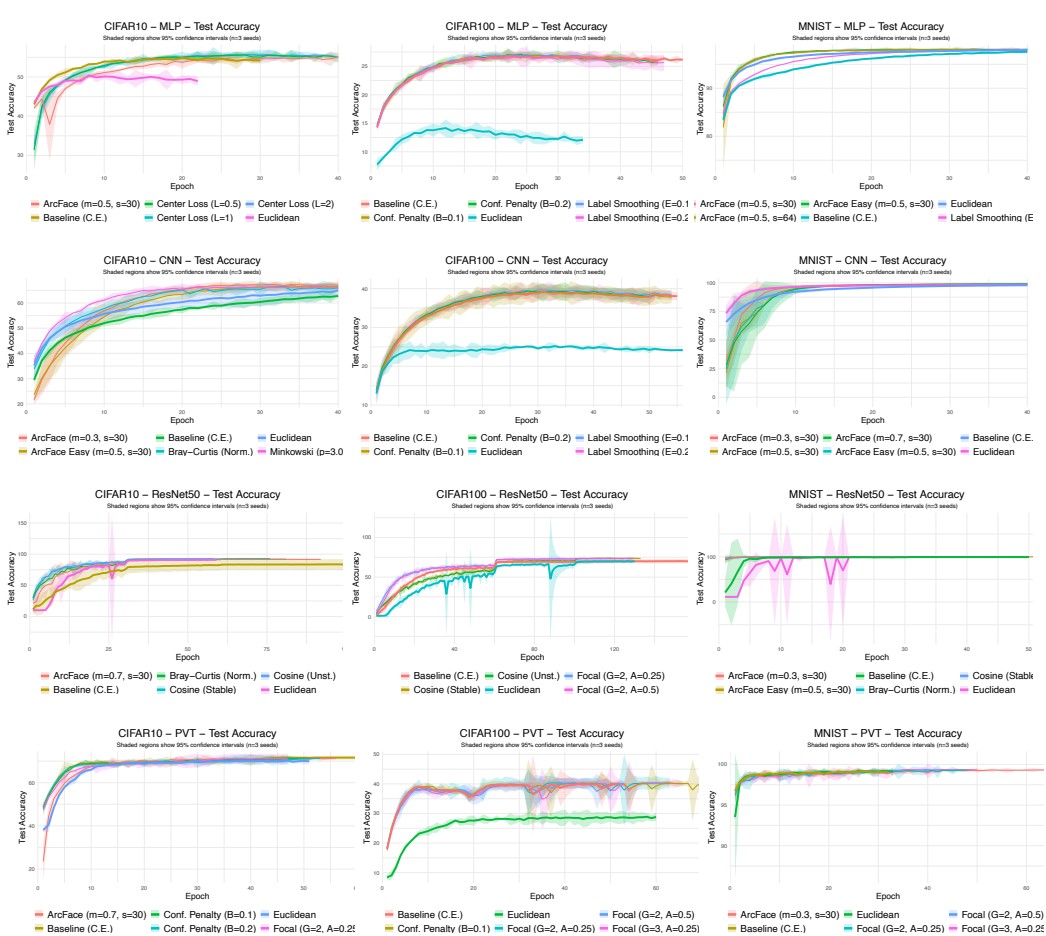

Figure 3: Vision: Accuracy curves with Confidence Intervals. Shaded regions show 95% confidence intervals (n = 3 seeds)

# F    VISION: RADAR PLOTS: ADDITIONAL DATASETS (MNIST, CIFAR10)

Figure 4 reports results for distance–based harmonic losses on MNIST and CIFAR10 across all four backbones (MLP, CNN, ResNet50, PVT). In summary, the MNIST and CIFAR-10 radar plots confirm what has been observed on other datasets: even on smaller benchmarks, non–Euclidean harmonic losses (particularly cosine, Bray-Curtis, and Chebyshev) can enhance representation structure and, on harder datasets, improve accuracy, all while maintaining comparable or better sustainability than Euclidean harmonic loss and the cross–entropy baseline.

## F.1    MNIST

**RQ1: Model Performance (F1, Accuracy).** Across all MNIST backbones, accuracy and F1 are saturated for several distances, but cosine–based harmonic losses (stable/unstable) and Bray–Curtis (normalized) remain among the most reliable high–performers. On MLP and CNN, these distances match or slightly exceed both Euclidean harmonic loss and the cross–entropy baseline. On ResNet50 and PVT, where capacity is ample, almost all distances reach near–perfect accuracy, confirming that changing the distance in the harmonic head does not harm performance.

**RQ2: Interpretability (PC2 EV, PCA 90%).** Even on this simple dataset, non–Euclidean distances already reshape the embedding geometry. Bray–Curtis (normalized) and Chebyshev (standard) produce noticeably higher PC2 explained variance and reduce the number of components needed to reach 90% EV, indicating compact, prototype–aligned clusters. Cosine harmonic losses also improve EV relative to Euclidean while maintaining top accuracy. Mahalanobis and Minkowski variants (on ResNet50) further concentrate variance, but their interpretability advantage is less pronounced on MNIST because the task is almost linearly separable.

**RQ3: Sustainability (Duration/Epoch/GFLOPs, Emissions).** For MNIST, the harmonic head constitutes a tiny fraction of the overall compute, so all distances exhibit similar Duration/Epoch/GFLOPs and emissions. Cosine and Bray–Curtis are essentially neutral relative to Euclidean and cross–entropy; small differences arise mainly from minor variations in convergence speed rather than per–step cost. The key takeaway from MNIST is therefore that non–Euclidean harmonic losses can improve representation structure without sacrificing accuracy or sustainability.

## F.2    CIFAR10

**RQ1: Model Performance (F1, Accuracy).** On CIFAR-10, cosine harmonic losses become clearly advantageous. For MLP and CNN, cosine (stable/unstable) and Bray–Curtis (normalized) consistently occupy the highest or near–highest F1 and accuracy, outperforming Euclidean harmonic loss and the cross–entropy baseline. On ResNet50 and PVT, cosine again delivers strong accuracy while remaining competitive with the best non–Euclidean alternatives (e.g., Minkowski $p=3.0$). Overall, cosine is the most robust choice across architectures once the task requires nontrivial feature extraction.

**RQ2: Interpretability (PC2 EV, PCA 90%).** CIFAR-10 further highlights the interpretability benefits of non–Euclidean geometry. Bray–Curtis and Chebyshev systematically increase PC2 EV and reduce PCA 90% dimensionality on MLP, CNN, and ResNet50, yielding sharper, more compact embeddings than Euclidean or cross–entropy. Cosine harmonic losses also improve EV over Euclidean, providing a favorable accuracy/interpretability compromise. On PVT, Canberra–weighted and Bray–Curtis variants similarly enhance variance concentration while preserving strong performance, reinforcing the observation that prototype–friendly distances induce more structured feature spaces.

**RQ3: Sustainability (Duration/Epoch/GFLOPs, Emissions).** On CIFAR-10, sustainability trends mirror those seen on larger datasets. Cosine harmonic loss is typically neutral–to–slightly–favorable in Duration/Epoch/GFLOPs and emissions relative to Euclidean and cross–entropy, especially on CNN and ResNet50 where convergence is faster. Bray–Curtis and Canberra variants introduce modest overhead, reflecting their more complex computations, but remain within the same qualitative efficiency regime. In all cases, the harmonic head is lightweight compared to the backbone, so the main sustainability differences arise from reduced steps–to–high accuracy rather than large per–step cost.

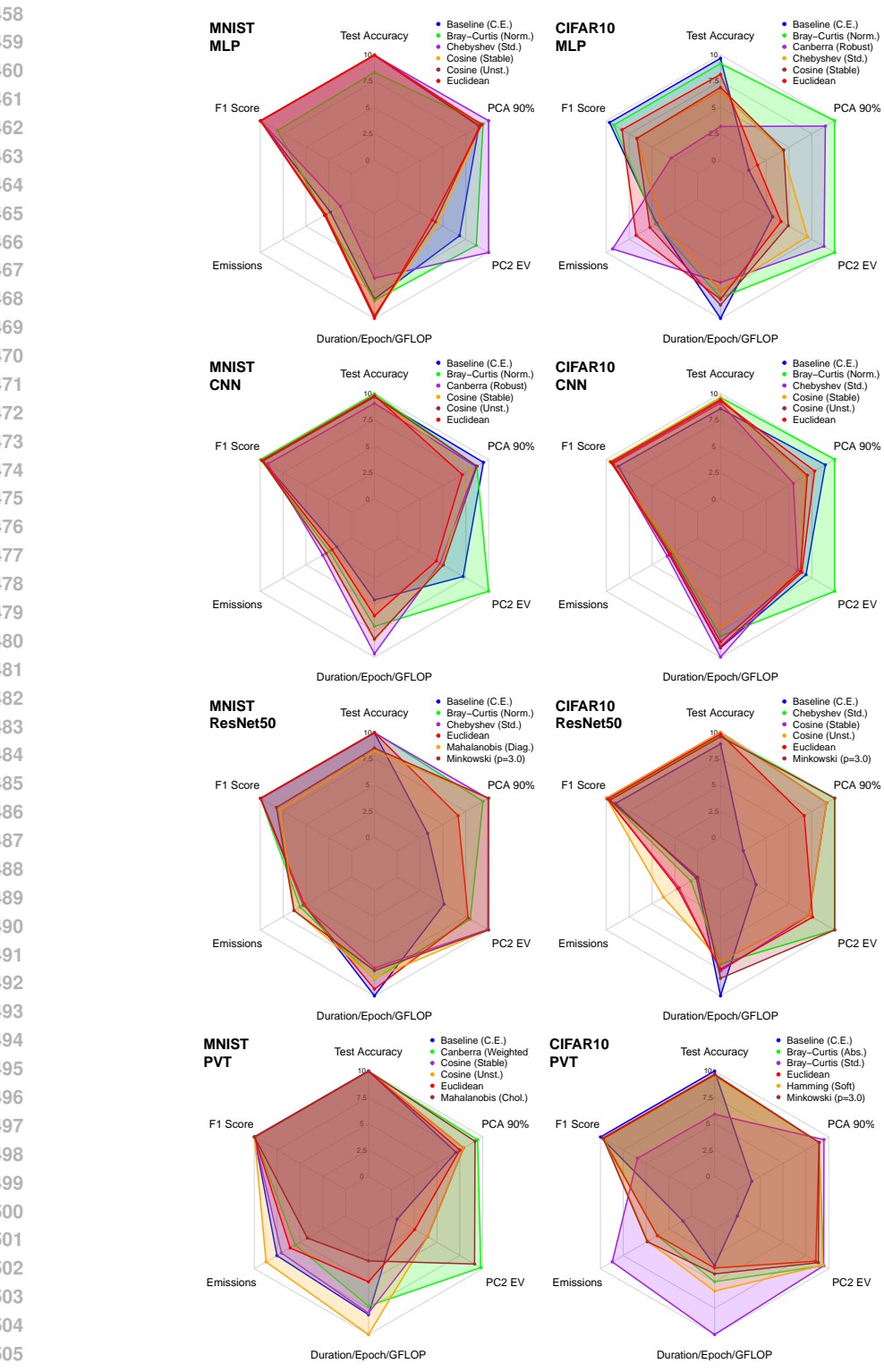

Figure 4: Vision: Radar plots: 1) *Model Performance* (F1, Accuracy); 2) *Interpretability* (PC2 EV, PCA 90%), and 3) *Sustainability* (Duration/Epoch/GFLOPs, Emissions). Plots feature Baseline (Cross-Entropy), Euclidean harmonic, and the four top-performing non-Euclidean harmonic losses.

## G RESULTS WITH ALTERNATIVE LOSS FUNCTIONS

To contextualize our distance–based harmonic losses, we benchmark against four widely used alternatives that are often motivated by calibration, robustness in low–data regimes, or representational compactness. Below we summarize each loss, its objective, and why it is relevant along our three axes: effectiveness, sustainability, and interpretability. Unless otherwise noted, these baselines are applied with a conventional linear head and softmax; they can also be evaluated on distance–parameterized logits (e.g., $-\text{dist}(\mathbf{h}, \mathbf{w}_c)$) to ensure architectural parity.

**Focal Loss (calibration, anti–grokking, class imbalance).** Focal Loss reweights examples by their difficulty:

$$\mathcal{L}_{\text{focal}}(\mathbf{z}, y) = -\alpha (1 - p_y)^\gamma \log p_y, \quad p_y = \frac{e^{z_y}}{\sum_j e^{z_j}},$$

with focusing parameter $\gamma \geq 0$ and class weight $\alpha \in (0, 1]$. By down–weighting well–classified (overconfident) samples, it yields smoother gradient signals, often improving calibration and mitigating late–stage overfitting behaviors akin to grokking. In our grids we consider $\gamma \in \{2, 3\}$ and $\alpha \in \{0.25, 0.5\}$. *Sustainability:* modest compute overhead (same forward/backward shape as CE), but potentially fewer effective updates on easy samples; net carbon effect is typically neutral to slightly higher than CE, depending on convergence behavior.

**Label Smoothing (reduced overconfidence, low–data stability).** Label Smoothing replaces the one–hot target with

$$\tilde{\mathbf{y}} = (1 - \varepsilon) \mathbf{e}_y + \frac{\varepsilon}{K} \mathbf{1}, \quad \mathcal{L}_{\text{LS}}(\mathbf{z}, y) = -\sum_{c=1}^{K} \tilde{y}_c \log p_c,$$

where $\varepsilon \in [0, 1)$ controls smoothing (we use $\varepsilon \in \{0.1, 0.2\}$). The softened targets reduce overconfidence and improve generalization in scarce–label settings; they also stabilize optimization by shrinking logit magnitudes. *Interpretability:* mild regularization can yield more isotropic features; *sustainability:* training cost matches CE, with potential reductions in steps–to–target when overconfidence previously harmed convergence.

**Center Loss (prototype compactness, cluster interpretability).** Center Loss explicitly penalizes the distance to a class prototype:

$$\mathcal{L}_{\text{center}}(\mathbf{h}, y) = \frac{1}{2} \left\| \mathbf{h} - \mathbf{c}_y \right\|_2^2, \quad \mathcal{L} = \mathcal{L}_{\text{CE}} + \lambda \mathcal{L}_{\text{center}},$$

with learnable class centers $\{\mathbf{c}_k\}$ and trade–off $\lambda > 0$ (we test $\lambda \in \{0.5, 1, 2\}$). This encourages intra–class compactness and inter–class separability—properties that make feature clusters and decision prototypes easier to inspect. *Sustainability:* small extra memory and updates for $\mathbf{c}_k$; the overhead is minor relative to the backbone but measurable in long runs.

**Confidence Penalty (entropy regularization, anti–grokking).** Confidence Penalty adds a negative–entropy term to discourage over–peaked posteriors:

$$\mathcal{L}_{\text{CP}}(\mathbf{z}, y) = \mathcal{L}_{\text{CE}}(\mathbf{z}, y) - \beta \mathcal{H}(\mathbf{p}), \quad \mathcal{H}(\mathbf{p}) = -\sum_{c=1}^{K} p_c \log p_c,$$

with $\beta > 0$ (we use $\beta \in \{0.1, 0.2\}$). By explicitly rewarding higher predictive entropy when appropriate, it reduces brittle overconfidence and can temper delayed generalization (grokking–like) effects. *Sustainability:* essentially identical compute to CE; any carbon changes stem from altered convergence trajectories rather than per–step cost.

**ArcFace (angular margins, maximized class separation).** ArcFace (Deng et al., 2019) introduces an *additive angular margin* that enlarges the decision boundary between classes on the unit hypersphere. Given normalized features $\mathbf{h}$ and normalized class weights $\mathbf{w}_c$, the cosine similarity $\cos \theta_c = \langle \mathbf{h}, \mathbf{w}_c \rangle$ is modified for the target class $y$ by adding a fixed angular margin $m$:

$$\cos(\theta_y + m) = \cos \theta_y \cos m - \sin \theta_y \sin m.$$

ArcFace replaces the final linear classifier with a scaled angular softmax:

$$\mathcal{L}_{\text{arcface}}(\mathbf{h}, y) = -\log \frac{\exp\left(s\,\cos(\theta_y + m)\right)}{\exp\left(s\,\cos(\theta_y + m)\right) + \sum_{c \neq y} \exp\left(s\,\cos\theta_c\right)},$$

where $s$ is a feature-scale parameter (typically $s = 30$–$64$). By manipulating angles rather than norms, ArcFace enforces tighter class clustering and larger inter-class separation, and is widely regarded as a *strong margin-based baseline* in metric learning. This makes it a particularly relevant comparator for harmonic losses: both approaches normalize features to a hypersphere and control geometry around class prototypes, but ArcFace explicitly pushes angular margins, whereas harmonic losses adjust the entire distance landscape. *Sustainability:* ArcFace is lightweight (same complexity as cosine classifiers), but the angular margin can slightly increase optimization stiffness, occasionally raising per-step compute or slowing convergence. In our experiments we consider $m \in \{0.3, 0.5\}$ and $s \in \{30, 64\}$.

Each baseline addresses a failure mode that harmonic losses also target but via different inductive biases: Focal/Label Smoothing emphasize calibration and data efficiency; Center Loss operationalizes prototype compactness; Confidence Penalty discourages pathological overconfidence. This makes them natural comparators for our *distance–based* formulation, which subsumes prototype reasoning in its very parameterization and, as we show, can simultaneously improve accuracy, reduce emissions, and enhance interpretability.

### G.1 Vision: Fine-grained results

A cross–backbone inspection in Tables 12–15 shows that the additional baselines (Focal, Label Smoothing, Confidence Penalty, Center Loss, ArcFace) are competitive on accuracy, but do not displace the non-Euclidean harmonic losses as the most balanced options.

On **CIFAR-10 and CIFAR-100 with ResNet-50**, cosine-based harmonic heads remain among the strongest configurations: they deliver the largest accuracy and F1 gains over cross-entropy (up to $\approx 11\%$ relative on CIFAR-10 and $5\%$ on CIFAR-100), while simultaneously reducing emissions by 10–30% and sharply concentrating the representation (PC90% dropping from 50 to 5–8 components). Focal and ArcFace variants sometimes match top accuracy, but typically exhibit weaker EV/PC90% improvements or higher emissions, so they do not dominate the multi-criteria trade-off.

For **PVT backbones**, where capacity and input resolution are higher, the picture is more nuanced. On CIFAR-10 PVT, calibration-oriented losses (Label Smoothing, Focal) offer slight accuracy improvements and sizable emissions reductions, yet Euclidean harmonic achieves the most compact geometry (PC90% from 17 to 3) at only a small performance cost. On CIFAR-100 PVT, focal and confidence-penalty losses are best in accuracy, but Euclidean harmonic again produces the most concentrated feature spaces (PC90% 4 vs. 50), highlighting an interpretability advantage even when it is not the accuracy winner.

On the **high-resolution Marathi Sign dataset**, nearly all methods saturate accuracy ($\geq 0.999$), so the comparison is driven by structure and sustainability. Here, cosine-based harmonic losses for ResNet-50 (and to a lesser extent for PVT) achieve substantial EV/PC90% gains. e.g., Cosine (Unst.) reduces PC90% from 15.5 to 6.5 while slightly lowering emissions, demonstrating that our non-Euclidean harmonic heads remain competitive even in settings where strong baselines like ArcFace and Center Loss are present.

Table 12: Results for CIFAR100 PVT (top-8 losses) and % changes w.r.t. Baseline (CE).

| Method | Acc | F1 | gCO$_2$eq | EV | PC90% |
|---|---|---|---|---|---|
| Baseline | 0.3994 | 0.3970 | 3.67 | 0.0973 | 50.0 |
| Focal ($\gamma$=2, $\alpha$=0.25) | 0.4017 (0.59%) | 0.3996 (0.65%) | 4.9609 (-35.35%) | 0.1257 (29.22%) | 50.0 (0%) |
| Focal ($\gamma$=3, $\alpha$=0.25) | 0.4015 (0.53%) | 0.3999 (0.72%) | 4.9567 (-35.24%) | 0.1364 (40.21%) | 50.0 (0%) |
| Focal ($\gamma$=2, $\alpha$=0.5) | 0.4010 (0.40%) | 0.3997 (0.66%) | 3.9182 (-6.90%) | 0.1271 (30.64%) | 50.0 (0%) |
| Conf. Penalty ($\beta$=0.1) | 0.3999 (0.13%) | 0.3977 (0.17%) | 5.2574 (-43.44%) | 0.0963 (-0.96%) | 50.0 (0%) |
| Label Smoothing ($\varepsilon$=0.1) | 0.3894 (-2.50%) | 0.3888 (-2.08%) | 2.6626 (27.36%) | 0.0818 (-15.92%) | 50.0 (0%) |
| Conf. Penalty ($\beta$=0.2) | 0.3859 (-3.38%) | 0.3834 (-3.45%) | 2.1718 (40.74%) | 0.0749 (-23.00%) | 50.0 (0%) |
| Label Smoothing ($\varepsilon$=0.2) | 0.3847 (-3.67%) | 0.3851 (-3.00%) | 2.2097 (39.71%) | 0.0742 (-23.74%) | 50.0 (0%) |
| ArcFace ($m$=0.5, $s$=30) | 0.3728 (-6.65%) | 0.3772 (-5.00%) | 2.6171 (28.60%) | 0.1259 (29.41%) | 50.0 (0%) |
| Euclidean | 0.2864 (-28.29%) | 0.2945 (-25.83%) | 6.4329 (-75.51%) | 0.8414 (765.15%) | 4.0 (92.00%) |

Table 13: Results for CIFAR100 ResNet50 (top-8 losses) and % changes w.r.t. Baseline (CE).

| Method | Acc | F1 | gCO$_2$eq | EV | PC90% |
|---|---|---|---|---|---|
| Baseline | 0.7006 | 0.6993 | 89.64 | 0.1069 | 50.0 |
| Cosine (Stable) | 0.7381 (5.35%) | 0.7384 (5.59%) | 79.2831 (11.55%) | 0.5915 (453.51%) | 8.0 (84.00%) |
| Focal ($\gamma$=2, $\alpha$=0.25) | 0.7349 (4.90%) | 0.7342 (5.00%) | 72.1795 (19.48%) | 0.1468 (37.32%) | 50.0 (0%) |
| Focal ($\gamma$=2, $\alpha$=0.5) | 0.7341 (4.79%) | 0.7332 (4.85%) | 78.7234 (12.18%) | 0.1224 (14.52%) | 50.0 (0%) |
| Cosine (Unst.) | 0.7340 (4.77%) | 0.7349 (5.09%) | 72.5413 (19.07%) | 0.5891 (451.21%) | 8.0 (84.00%) |
| Focal ($\gamma$=3, $\alpha$=0.25) | 0.7311 (4.36%) | 0.7308 (4.51%) | 81.5875 (8.98%) | 0.1554 (45.41%) | 50.0 (0%) |
| Label Smoothing ($\varepsilon$=0.1) | 0.7261 (3.64%) | 0.7248 (3.65%) | 79.8223 (10.95%) | 0.1469 (37.44%) | 50.0 (0%) |
| Label Smoothing ($\varepsilon$=0.2) | 0.7221 (3.08%) | 0.7206 (3.04%) | 81.1883 (9.43%) | 0.1524 (42.59%) | 50.0 (0%) |
| ArcFace ($m$=0.7, $s$=30) | 0.7166 (2.29%) | 0.7150 (2.25%) | 70.5590 (21.29%) | 0.6059 (466.93%) | 48.33 (3.33%) |
| Euclidean | 0.7047 (0.59%) | 0.7055 (0.89%) | 87.7280 (2.13%) | 0.4301 (302.49%) | 33.67 (32.67%) |

Table 14: Results for MarathiSign PVT (top-8 losses) and % changes w.r.t. Baseline (CE).

| Method | Acc | F1 | gCO$_2$eq | EV | PC90% |
|---|---|---|---|---|---|
| Baseline | 0.9965 | 0.9964 | 2.85 | 0.2135 | 24.75 |
| ArcFace ($m$=0.7, $s$=30) | 0.9997 (0.33%) | 0.9997 (0.33%) | 4.2108 (-47.88%) | 0.1132 (-47.00%) | 36.0 (-45.45%) |
| Focal ($\gamma$=3, $\alpha$=0.25) | 0.9997 (0.32%) | 0.9996 (0.32%) | 2.7240 (4.34%) | 0.2749 (28.74%) | 19.33 (21.89%) |
| Center Loss ($\lambda$=1) | 0.9995 (0.31%) | 0.9995 (0.31%) | 5.1746 (-81.72%) | 0.1824 (-14.56%) | 29.67 (-19.87%) |
| Conf. Penalty ($\beta$=0.2) | 0.9995 (0.30%) | 0.9994 (0.30%) | 3.6912 (-29.63%) | 0.1504 (-29.58%) | 33.67 (-36.03%) |
| Focal ($\gamma$=2, $\alpha$=0.25) | 0.9993 (0.29%) | 0.9993 (0.28%) | 2.5648 (9.93%) | 0.2853 (33.63%) | 19.33 (21.89%) |
| ArcFace ($m$=0.3, $s$=30) | 0.9992 (0.28%) | 0.9992 (0.28%) | 7.5158 (-163.94%) | 0.1756 (-17.76%) | 28.0 (-13.13%) |
| Label Smoothing ($\varepsilon$=0.2) | 0.9992 (0.28%) | 0.9991 (0.27%) | 2.6977 (5.26%) | 0.1190 (-44.26%) | 35.0 (-41.41%) |
| Cosine (Unst.) | 0.9991 (0.27%) | 0.9991 (0.26%) | 7.3447 (-157.93%) | 0.5552 (160.04%) | 7.0 (71.72%) |
| Euclidean | 0.9994 (0.30%) | 0.9994 (0.30%) | 4.3621 (-53.19%) | 0.5035 (135.83%) | 14.25 (42.42%) |

Table 15: Results for MarathiSign ResNet50 (top-8 losses) and % changes w.r.t. Baseline (CE).

| Method | Acc | F1 | gCO$_2$eq | EV | PC90% |
|---|---|---|---|---|---|
| Baseline | 0.9998 | 0.9998 | 35.41 | 0.4507 | 15.5 |
| Conf. Penalty ($\beta$=0.1) | 0.9999 (0.01%) | 0.9999 (0.01%) | 29.6596 (16.24%) | 0.4393 (-2.55%) | 17.5 (-12.90%) |
| Cosine (Stable) | 0.9999 (0.01%) | 0.9999 (0.01%) | 54.8139 (-54.80%) | 0.7321 (62.42%) | 5.5 (64.52%) |
| Conf. Penalty ($\beta$=0.2) | 0.9998 (0.00%) | 0.9998 (0.00%) | 30.3697 (14.23%) | 0.3288 (-27.05%) | 33.0 (-112.90%) |
| Focal ($\gamma$=2, $\alpha$=0.25) | 0.9998 (0.00%) | 0.9998 (0.00%) | 24.95 (29.54%) | 0.4016 (-10.89%) | 17.0 (-9.68%) |
| Focal ($\gamma$=3, $\alpha$=0.25) | 0.9998 (0.00%) | 0.9998 (0.00%) | 28.1206 (20.58%) | 0.4286 (-4.91%) | 16.5 (-6.45%) |
| Cosine (Unst.) | 0.9997 (-0.01%) | 0.9997 (-0.01%) | 33.8287 (4.46%) | 0.6427 (42.58%) | 6.5 (58.06%) |
| Label Smoothing ($\varepsilon$=0.1) | 0.9997 (-0.01%) | 0.9997 (-0.01%) | 33.5266 (5.31%) | 0.1758 (-60.99%) | 36.0 (-132.26%) |
| Label Smoothing ($\varepsilon$=0.2) | 0.9997 (-0.01%) | 0.9997 (-0.01%) | 38.7093 (-9.32%) | 0.1641 (-63.60%) | 36.0 (-132.26%) |
| Euclidean | 0.9984 (-0.14%) | 0.9983 (-0.15%) | 50.2050 (-41.79%) | 0.2787 (-38.18%) | 50.0 (-222.58%) |

## G.2 Vision: Radar Plots with Additional Losses (MNIST, CIFAR10, CIFAR100)

Figure 5 presents an expanded multi-criteria analysis across MNIST, CIFAR-10, and CIFAR-100 using MLP, CNN, ResNet50, and PVT backbones. This comparison tests whether the advantages previously attributed to harmonic losses persist when measured against widely adopted alternatives for regularization, interpretability, and robustness.

**RQ1: Model Performance (F1, Test Accuracy).** Across datasets and architectures, the **harmonic losses** – particularly the *cosine-* and *Bray–Curtis–based* variants – remain the strongest overall performers. While **Focal Loss** and **Label Smoothing** occasionally narrow the gap on more complex datasets such as CIFAR-100, they do not consistently surpass harmonic losses across backbones. Cosine-based harmonic loss maintains higher accuracy and smoother convergence, especially for CNN and ResNet50, showing greater robustness to data imbalance and optimization noise than either Focal or Confidence Penalty Loss. Even when Center Loss improves class compactness, it rarely translates into superior end-task accuracy, reinforcing that distance-based formulations bring more balanced generalization benefits.

**RQ2: Interpretability (PC2 EV, PCA 90%).** The advantage of harmonic losses extends beyond performance: **non-Euclidean harmonics**, especially Bray–Curtis and Chebyshev, consistently yield the most structured latent geometries. They capture more variance with fewer principal components and align features more distinctly around class prototypes. Although **Center Loss** achieves comparable compactness in isolated cases, its representations tend to be less stable across architectures. **Label Smoothing** and **Confidence Penalty** slightly improve feature spread, but their effects remain shallow compared to the systematic geometric alignment achieved by harmonic formulations. This supports the notion that explicit metric-based geometry is a stronger driver of interpretability than indirect regularization.

**RQ3: Sustainability (Duration/Epoch, Emissions).** When considering efficiency, harmonic losses continue to hold their edge. They achieve competitive or lower $CO_2$ emissions than both Euclidean and cross-entropy baselines. Among the new baselines, only **Label Smoothing** approaches similar energy efficiency, while **Focal Loss** incurs additional computational cost due to its per-sample weighting. Despite this, none of the conventional alternatives outperform the best-performing harmonic distances on a joint accuracy–emission axis, confirming that the added geometric structure of harmonic loss does not come at a sustainability penalty.

Three key findings emerge: i) **Cosine- and Bray–Curtis–based harmonic losses** remain the most consistently effective across accuracy, interpretability, and sustainability; ii) **Conventional regularized losses** such as Focal or Label Smoothing can mitigate specific failure modes (imbalance, overconfidence) but do not achieve the same balance across criteria; iii) The geometric grounding of harmonic losses continues to provide superior inductive structure, yielding smoother optimization, clearer feature organization, and greener training. Overall, these results reaffirm the *general dominance and stability of harmonic loss formulations*, even against strong baselines optimized for robustness and interpretability.

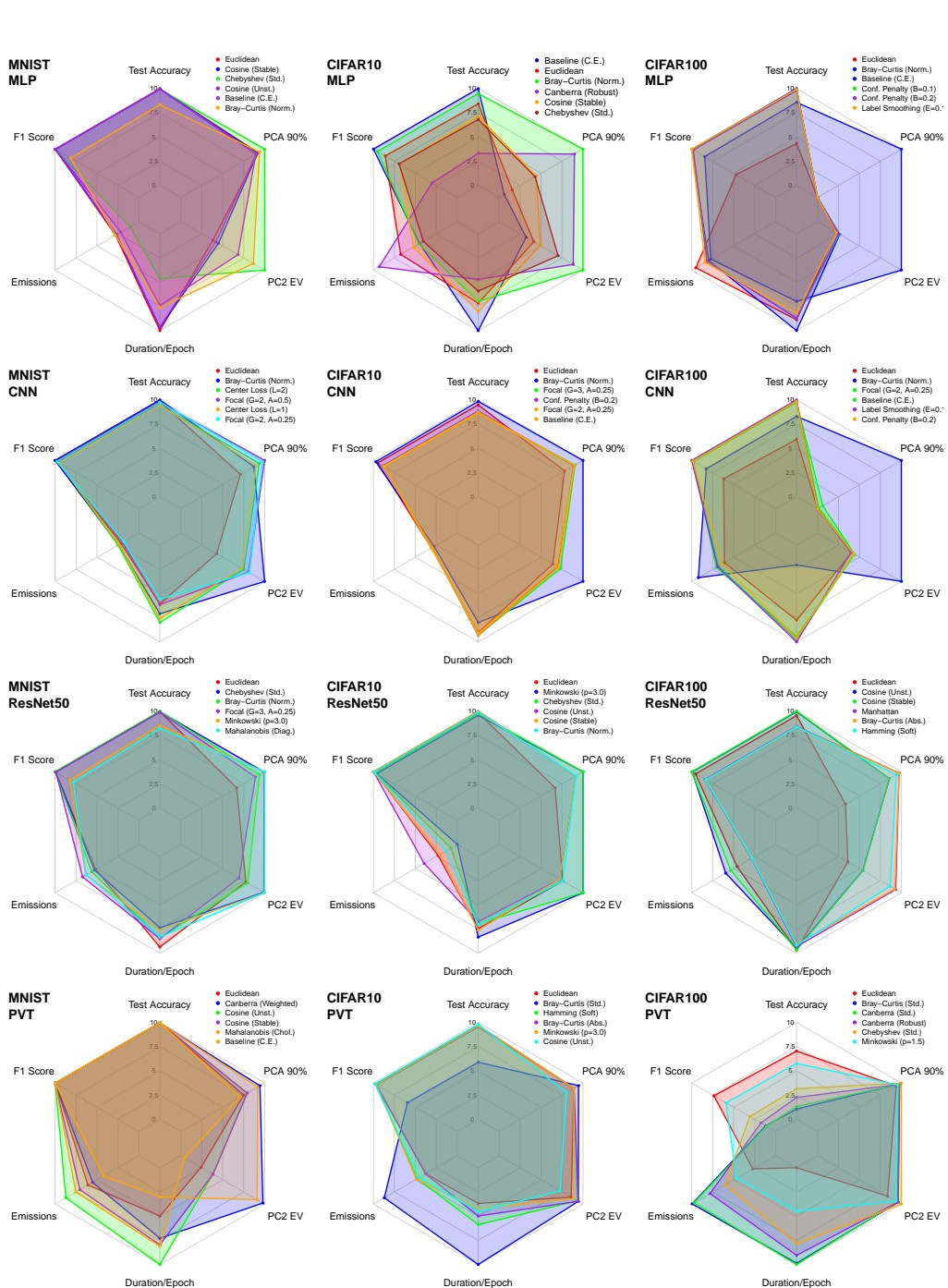

Figure 5: Vision: Radar plots – MNIST, CIFAR10, CIFAR100: 1) *Model Performance* (F1, Accuracy); 2) *Interpretability* (PC2 EV, PCA 90%), and 3) *Sustainability* (Duration/Epoch, Emissions). Plots feature Baseline (Cross-Entropy), Euclidean harmonic, and the four top-performing losses.

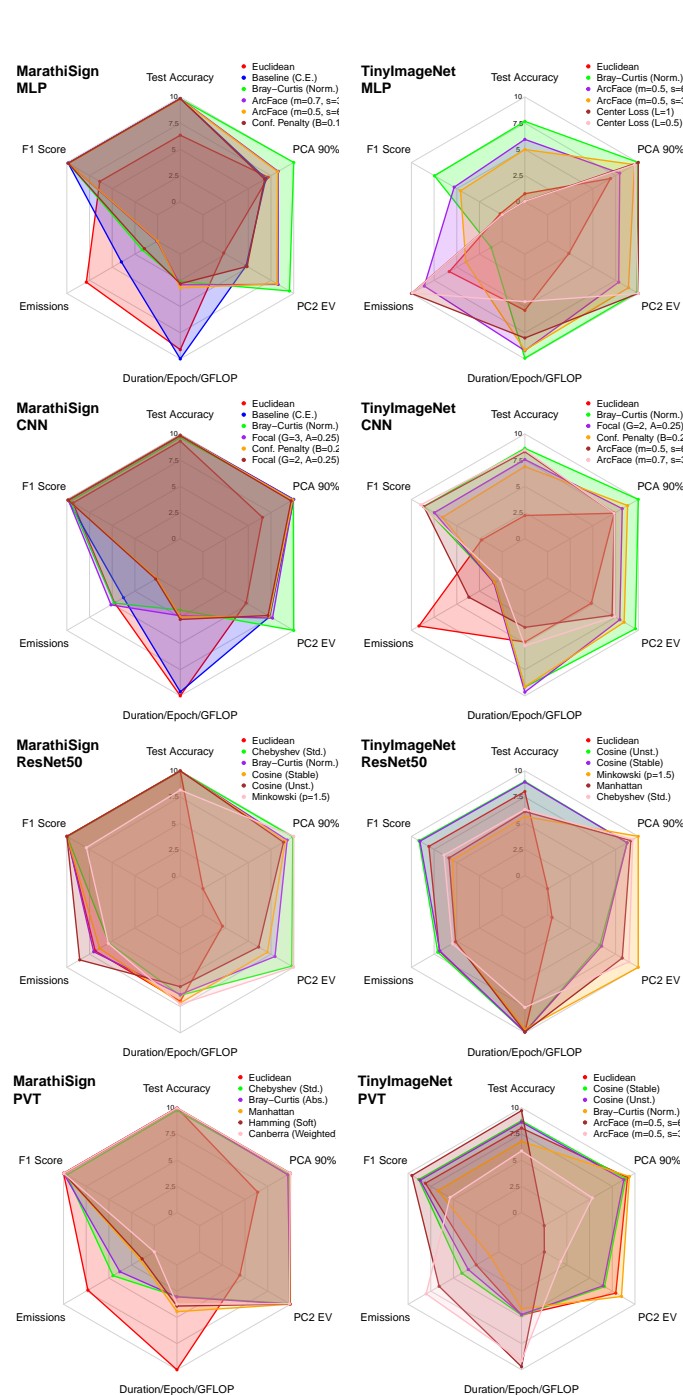

Figure 6: Vision: Radar plots – Marathi Sign Language, TinyImageNet: 1) *Model Performance* (F1, Accuracy); 2) *Interpretability* (PC2 EV, PCA 90%), and 3) *Sustainability* (Duration/Epoch, Emissions). Plots feature Baseline (Cross-Entropy), Euclidean harmonic, and the four top-performing losses.

G.3 VISION: RADAR PLOTS WITH ADDITIONAL LOSSES (MARATHI SIGN LANGUAGE, TINYIMAGENET)

Figure 6 extends the radar analysis to higher–resolution benchmarks (Marathi Sign Language, TinyImageNet) and augments the comparison set with strong loss baselines such as Focal Loss, ArcFace, Center Loss, and Confidence Penalty.

**RQ1: Model Performance (Accuracy, F1).** On Marathi Sign, the added losses make MLP and CNN particularly competitive: ArcFace and Focal occasionally obtain strong accuracy/F1, yet non–Euclidean harmonic losses remain among the best methods. For MLP, **Bray–Curtis (Normalized)** tracks or exceeds both cross–entropy and ArcFace while preserving a smooth performance profile. For CNN, **Bray–Curtis (Normalized)** and **cosine (stable/unstable)** consistently occupy the top accuracy/F1 slices; Focal and Confidence Penalty are competitive but never clearly dominate. On deeper backbones, the picture is even clearer: for **ResNet50** and **PVT** on Marathi Sign, all top–performing methods are harmonic losses, indicating that distance–based harmonic heads outperform alternative losses outright in this regime.

On TinyImageNet, a harder and more fine–grained benchmark, a similar pattern emerges. For MLP, ArcFace and Center Loss join **Bray–Curtis (Normalized)** and Euclidean in the top–performing losses, but Bray–Curtis remains competitive in accuracy while providing different geometric and sustainability properties. For CNN, the strongest Focal and ArcFace variants reach high F1, yet **Bray–Curtis (Normalized)** again sits near the performance frontier. On **ResNet50**, the top–performing losses are *entirely harmonic* (cosine, Minkowski, Chebyshev, Euclidean), and on **PVT** TinyImageNet the leaders are dominated by cosine and Bray–Curtis, with ArcFace appearing only as an alternative angular baseline. Overall, even in the presence of sophisticated angular–margin and confidence–shaping losses, non–Euclidean harmonic heads remain on or very near the performance Pareto frontier.

**RQ2: Interpretability (PC2 EV, PCA 90%).** The higher–resolution datasets accentuate differences in representation geometry. On Marathi Sign, harmonic distances such as **Bray–Curtis (Normalized/Absolute)**, **Chebyshev (Standard)**, and **Canberra/Hamming** for PVT yield the strongest PCA structure: they maximize PC2 explained variance and minimize the number of components required to reach 90% EV, indicating compact, prototype–aligned embeddings. ArcFace and Focal improve angular separation but generally do not achieve the same variance concentration as the best harmonic distances.

TinyImageNet confirms this trend. On MLP and CNN, Bray–Curtis and Chebyshev produce markedly higher PC2 EV and lower PCA 90% dimensionality than Euclidean and most additional baselines, including Center Loss and Focal. For ResNet50 and PVT, **cosine** and **Bray–Curtis** continue to enlarge the PCA wedges relative to Euclidean, whereas ArcFace's contribution is mainly on performance rather than on variance concentration. Thus, across both Marathi Sign and TinyImageNet, the most interpretable geometries are consistently induced by non–Euclidean harmonic losses rather than by the newly added baselines.

**RQ3: Sustainability (Duration/Epoch/GFLOPs, Emissions).** The sustainability axes show that richer loss design does not necessarily translate into greener training. On Marathi Sign MLP/CNN, several harmonic distances (e.g., **Bray–Curtis (Normalized)**, **Chebyshev**) attain *equal or lower* normalized Duration/Epoch/GFLOPs and emissions than cross–entropy and the added baselines; Focal and ArcFace occasionally incur slightly higher emissions due to their sharper gradients and additional computations. For ResNet50 and PVT, where backbone FLOPs dominate, all harmonic variants remain sustainability–competitive.

On TinyImageNet, the pattern persists. ArcFace and Focal may match harmonic losses in accuracy, but they usually do so with similar or higher emissions. Cosine and Bray–Curtis heads on ResNet50 and PVT often achieve comparable or better emissions than Euclidean, while still improving representation structure. Center Loss introduces modest overhead but does not surpass harmonic distances in overall sustainability.

Across Marathi Sign and TinyImageNet, adding strong baselines such as ArcFace, Focal, Center Loss, and Confidence Penalty *does not displace* the non–Euclidean harmonic losses from the top tier. Whenever these baselines are competitive in accuracy, harmonic distances typically offer superior interpretability and comparable or lower emissions. This reinforces our central claim that

distance–tailored harmonic heads provide a robust, geometry–aware alternative to contemporary loss designs, remaining competitive or superior across performance, structure, and sustainability, even on challenging high–resolution vision benchmarks.

### G.4 VISION: AGGREGATED EMISSIONS

Figure 7 reports the cumulative $CO_2$ emissions for all vision experiments (MNIST, CIFAR-10/100, Marathi Sign, and TinyImageNet), expressed as the difference in grams of $CO_2$ relative to the cross–entropy baseline (total CE emissions = 650.49 gCO₂eq over 680 runs). All methods lie within a band of roughly $\pm 8\%$ of this baseline, showing that changing the distance in the harmonic head affects emissions in a controlled—rather than catastrophic—way.:contentReference[oaicite:0]index=0

**Harmonic losses remain competitive or greener.** The most sustainable region of the plot is dominated by non–Euclidean harmonic losses. In particular, **Bray–Curtis (Normalized)**, **Bray–Curtis (Absolute)**, **Canberra (Weighted)**, and **Mahalanobis (Cholesky)** consistently achieve *lower* cumulative emissions than cross–entropy, even after adding the more demanding Marathi Sign and TinyImageNet settings. Cosine variants (*Cosine (Stable)* and *Cosine (Unstable)*) and Euclidean harmonic loss sit very close to the baseline, indicating that distance–based heads introduce essentially no sustainability penalty while still improving accuracy and representation structure.

**Behavior of additional baselines.** Among the newly added conventional baselines, **Label Smoothing** and **Confidence Penalty** occupy the middle of the spectrum: their emissions are comparable to, but generally not better than, those of the best harmonic distances. In contrast, more aggressive objectives such as **Focal Loss** and large–margin **Center Loss** variants tend to cluster on the higher–emission side, reflecting the extra computation and slower convergence induced by power–scaled gradients and auxiliary center updates. ArcFace configurations behave similarly: moderate settings can be near baseline, but high margin/scale choices increase emissions relative to the most efficient harmonic distances.

Across all four datasets and backbones, the qualitative picture is stable. Non–Euclidean harmonic losses provide some of the *greenest* options, often achieving lower or baseline–level emissions while simultaneously improving accuracy and interpretability. The main exception is **Mahalanobis (Standard)**, which remains the least sustainable configuration due to its covariance estimation cost—consistent with our earlier observation that Mahalanobis emphasizes representation clarity at a computational price. Overall, the expanded analysis confirms that distance choice in the harmonic head materially affects the carbon footprint of training, and that carefully chosen non–Euclidean geometries (e.g., Bray–Curtis, Canberra, cosine) offer a favorable performance–interpretability–sustainability trade–off compared to both Euclidean harmonic loss and modern regularized baselines.

Figure 7 reports cumulative emission *differences* (gCO₂eq) for each custom loss function across all 12 model/dataset combinations. (Total Baseline = 181.2 gCO₂eq).

**Lower-than-baseline emissions**: *Mahalanobis (Standard)* shows the largest *positive* delta, indicating consistently lower emissions; *Bray–Curtis (Standard)* and *Cosine (Unstable)* also sit on the positive side, with *Canberra (Standard)* and *Cosine (Stable)* slightly above zero. Euclidean and Manhattan are close to baseline. Of the new baseline loss functions introduced, Confidence Penalty performs on par with Cosine (Unstable) and the most efficiently compared to its counterparts. Almost all new losses are more efficient than Cross Entropy Loss, with varying degrees of success. Other distances are characterized by higher emissions, as shown by the red cluster. Results reinforce that non-Euclidean harmonic losses can be more sustainable than their Euclidean counterpart, and that the choice of distance materially affects the carbon footprint of model training.

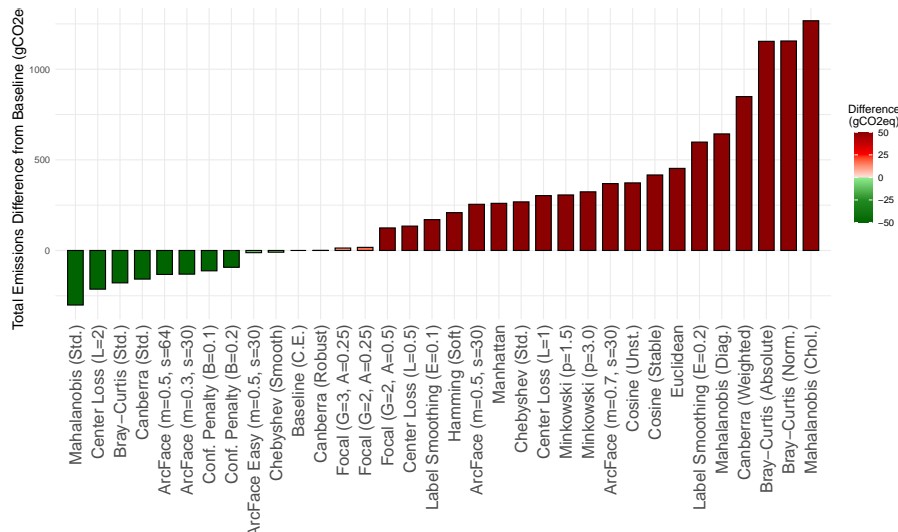

Figure 7: Vision: Emissions Averaged Across Seeds and Aggregated Over all 12 Model Backbones.

## H  CONVERGENCE ANALYSIS

**Vision:** Figure 8 reports the training and validation loss trajectories for PVT and ResNet50 across all datasets and all non-Euclidean harmonic losses. A key concern is whether distances that introduce nontrivial geometric structure such as cosine and Mahalanobis lead to unstable optimization or distorted convergence landscapes. Empirically, we observe no such issues.

Across MNIST, CIFAR-10, CIFAR-100, and Marathi Sign, all non–Euclidean harmonic losses exhibit smooth, monotonic decrease in the training objective and stable validation trends, with no oscillation, divergence, or gradient explosion. Even distances with stronger geometric bias (e.g., Mahalanobis, Chebyshev, Bray-Curtis) converge at rates comparable to or faster than Euclidean harmonic loss. Cosine variants in particular show the *fastest early descent*, followed by steady tightening of the validation curves, consistent with their angularly flatter basins.

Notably, none of the distances introduce optimization barriers, despite their differing curvature properties. Mahalanobis maintains stable descent even though anisotropic curvature could, in principle, yield direction-dependent gradients. Likewise, Canberra, Hamming, Manhattan, and Minkowski losses converge smoothly, indicating that the harmonic formulation effectively normalizes distance geometry into a well-conditioned optimization surface.

Overall, the loss curves demonstrate that the harmonic link function absorbs geometric variability and translates heterogeneous distance metrics into similarly well-behaved training dynamics. This provides experimental evidence that alternative geometries do not impair convergence nor destabilize class separation boundaries.

**Language:** Figure 9 reports training/validation loss, training accuracy, and (for GPT–2B) training and validation perplexity for cross–entropy, Euclidean harmonic, and Minkowski ($p$=2) heads across BERT–0.1B, GPT–0.1B, GPT–2B, and QWEN2–0.5B. Across all architectures, the distance–based harmonic losses exhibit smooth optimization dynamics: losses decrease monotonically with no oscillatory or unstable regimes, and accuracy curves increase steadily towards a plateau.

For BERT–0.1B, Euclidean and Minkowski harmonic losses reduce both training and validation loss more quickly than cross–entropy and converge to a lower plateau, while achieving higher final training accuracy. GPT–0.1B shows a similar pattern: all three heads converge, but the harmonic variants reach a given accuracy earlier and with gently sloping curves, indicating stable gradients. For GPT–2B and QWEN2–0.5B, the three heads track each other closely in both loss and accuracy, confirming that the change of geometry does not impede convergence even at larger scale. The validation loss curves mirror the training behaviour: no divergence or late–stage degradation is observed for any harmonic configuration.

The GPT–2B perplexity panel further corroborates this picture. Training and validation perplexity decrease rapidly and stabilize to comparable levels for all heads; the harmonic variants sometimes achieve slightly faster early reductions, but do not introduce pathological behaviour. Overall, these results show that replacing the linear classifier with a distance–based harmonic head preserves, and in some cases marginally improves, the convergence properties of standard cross–entropy while enabling the geometric and interpretability benefits discussed in the main paper.

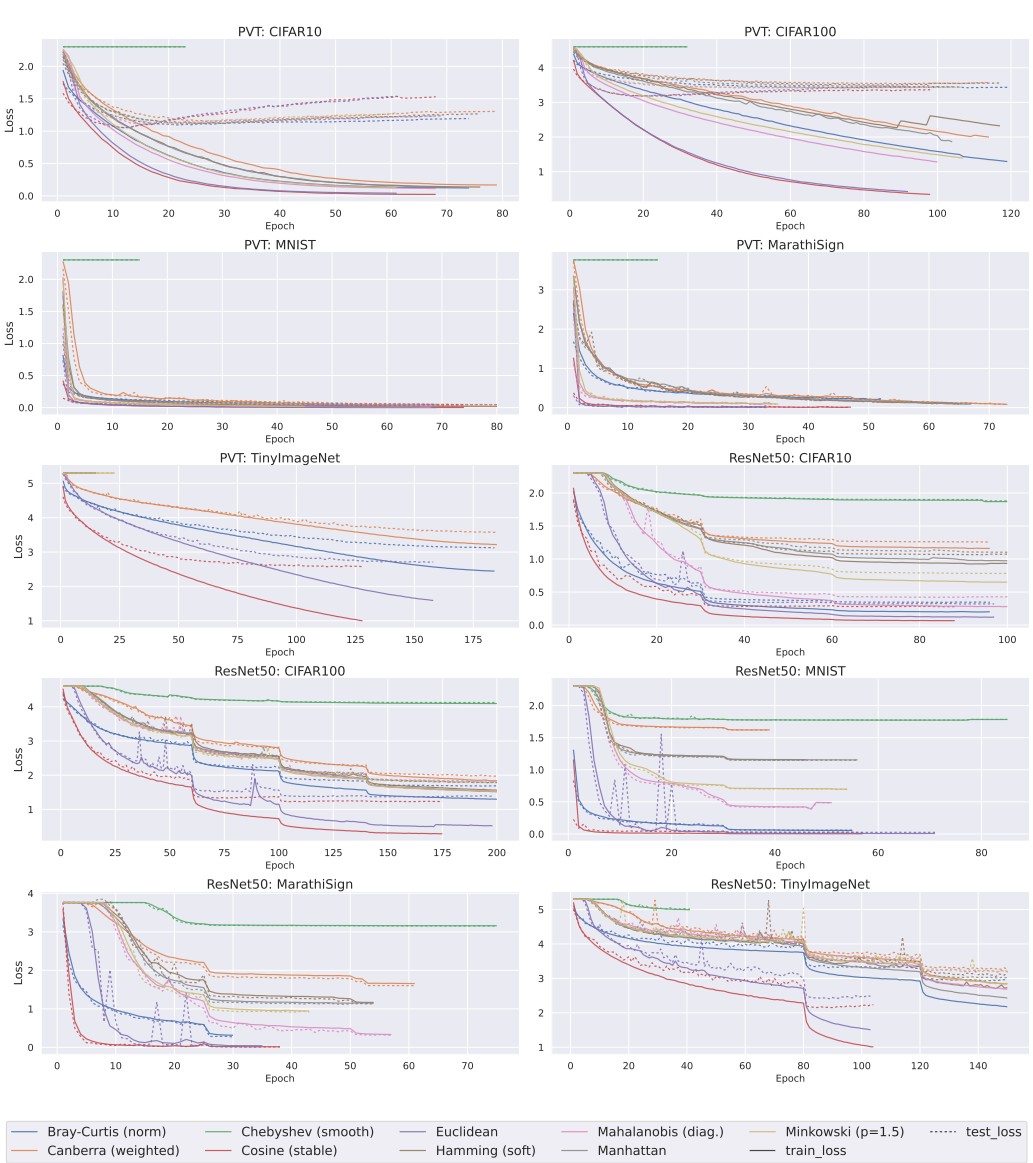

Figure 8: Loss convergence behavior with PVT and ResNet50: Training and Validation loss across all datasets with different non-Euclidean harmonic losses.

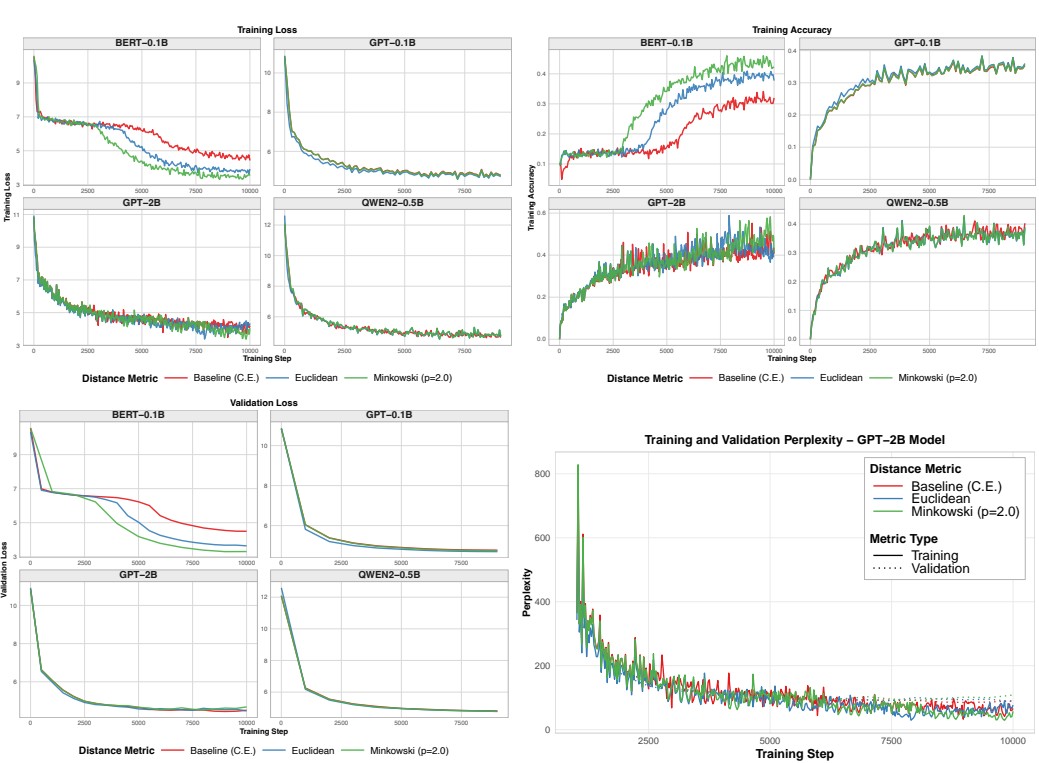

Figure 9: Loss convergence behavior with language models (BERT-0.1B, GPT-0.1B, QWEN2-0.5B, GPT-2B).

# I  GROKKING ANALYSIS: MODULO ADDITION

Figure 10 summarizes the behavior of standard MLPs and H–MLPs trained on the synthetic modulo–addition task, a setting known to exhibit pronounced grokking effects under cross–entropy. The top two rows illustrate training curves and corresponding 2D embeddings for baseline cross–entropy models (first two columns) and Euclidean harmonic loss (third and fourth columns). The remaining rows extend this comparison to alternative non–Euclidean harmonic losses.

**Cross–entropy exhibits clear grokking.**  For both the standard MLP and its lightly regularized variant, cross–entropy produces the characteristic grokking pattern: training accuracy rapidly converges while test accuracy improves only after a long delay. This decoupling is consistent with prior observations in algorithmic tasks, where cross–entropy tends to overfit memorization pathways before discovering the true modular arithmetic structure. The PCA plots confirm this: the learned embeddings under cross–entropy have diffuse, irregular geometry, and the first two principal components explain only a small fraction of the variance (EV $\approx$ 20–30%).

**Euclidean harmonic loss eliminates grokking and induces a perfect geometric structure.**  In contrast, the Euclidean harmonic model reaches high train *and* test accuracy simultaneously. No grokking delay is observed. The PCA projection reveals a striking property: the latent representation forms a *perfect 2D circle*, and the first two principal components explain nearly all variance (EV $\approx$ 100%). This matches theoretical expectations for harmonic distance–based classification on cyclic group structure: the model learns an isometric embedding of $\mathbb{Z}_n$ into the plane, validating the geometric alignment induced by harmonic objectives.

**Other distance–based harmonic losses replicate the circle structure with similarly fast generalization.**  The bottom rows show that this desirable behavior is *not* unique to Euclidean distance. Cosine, stable cosine, Manhattan (1–norm), several Canberra variants, Hamming losses, Minkowski $p = 3$, Chebyshev, and others all produce the same qualitative outcome:

- **Immediate or near–immediate generalization**, with no grokking phase.

- **Highly structured 2D embeddings**, often forming a near–perfect circle.

- **Explained variance approaching 100%**, indicating strong alignment to a low–dimensional manifold reflecting the algebraic symmetry of the task.

Some distances (e.g., Hamming and Chebyshev) produce slightly rotated or warped circles, but the essential geometric structure and variance concentration remain intact. This demonstrates that harmonic losses robustly recover the underlying modular arithmetic structure *regardless of the distance family*.

**Harmonic losses reduce grokking compared to cross–entropy.**  Across all non–Euclidean distances tested, harmonic losses exhibit two consistent advantages over cross–entropy:

1. **Reduced grokking or complete elimination of delayed generalization**. Training and test accuracy rise together, indicating that the model discovers the algorithmic rule rather than memorizing individual cases.

2. **Improved interpretability via stable geometric structure**. The emergence of a low–dimensional circular manifold with EV close to 1.0 serves as a quantitative and visual certificate of representation clarity.

These results reinforce the core claims of the paper: harmonic losses promote structured, prototype–aligned representations and smoother, more reliable optimization dynamics, even on tasks where cross–entropy typically groks. The fact that many distances achieve EV $\approx$ 100% highlights that the benefits of harmonic classification do not depend on Euclidean geometry alone, but arise from the broader class of distance–based harmonic objectives.

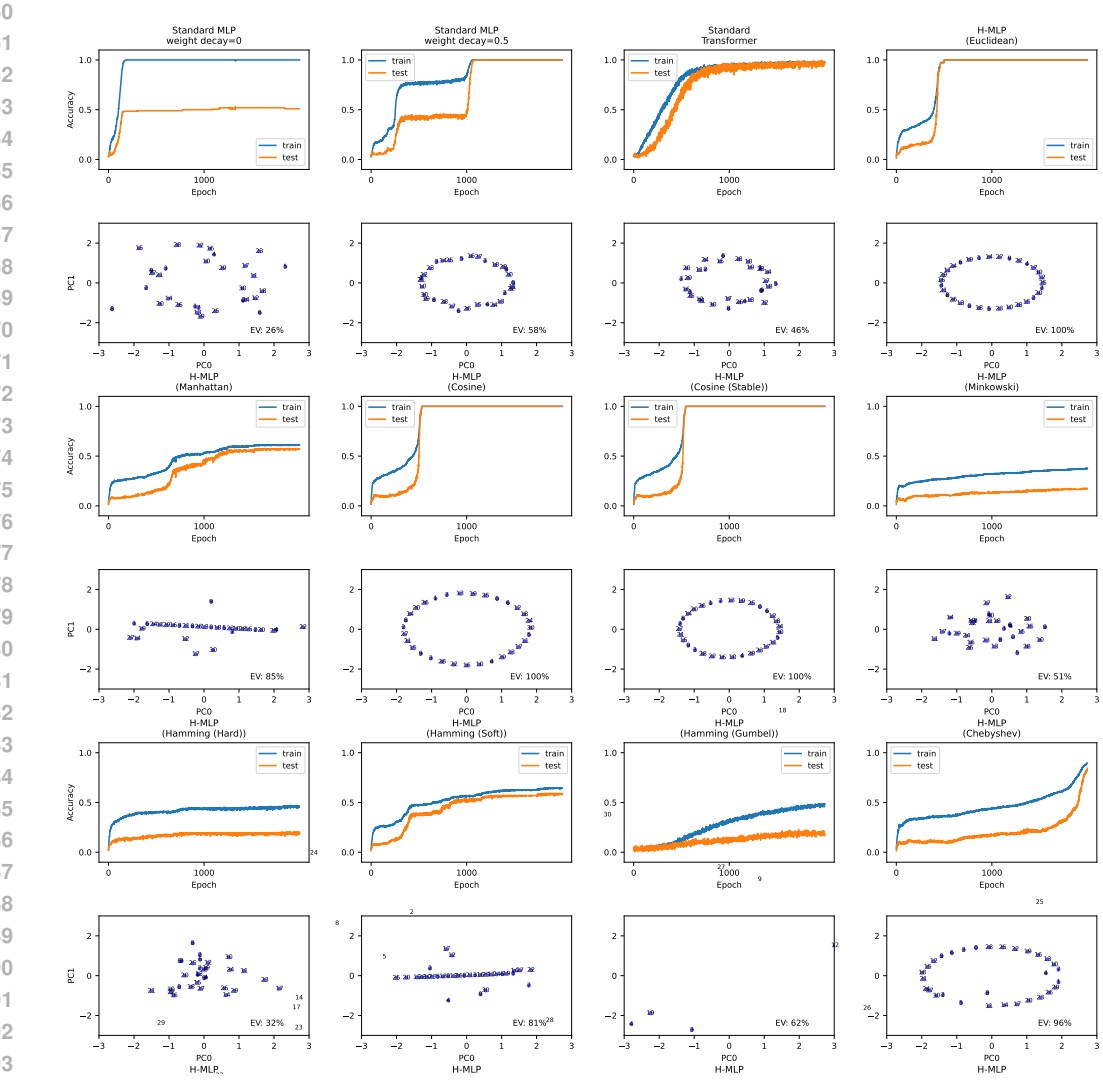

Figure 10: Results on standard MLP trained for modular addition. The harmonic model trained for modular addition generalizes quickly without grokking. Moreover, the embedding forms a perfect 2D circle. EV in the plot represents the explained variance by the first two principal components of the embedding.

## J COMPUTATIONAL COMPLEXITY: FLOPS

Table 16 reports the approximate floating–point operations per forward pass for each backbone–dataset. On $32\times32$ inputs (MNIST resized, CIFAR-10/100, MarathiSign), the FLOP hierarchy is consistent: MLP is cheapest ($<0.004$ GFLOPs), CNN roughly $3\times$ more expensive ($\approx 0.012$ GFLOPs), PVT adds another $\sim3\times$ ($\approx 0.038$ GFLOPs), and ResNet50 is about $2\times$ PVT ($\approx 0.08$ GFLOPs). Moving to high–resolution inputs ($224\times224$ for TinyImageNet and our high–resolution MarathiSign runs) increases cost by two orders of magnitude: PVT rises to $\sim1.9$ GFLOPs and ResNet50 to $\sim4.1$ GFLOPs per forward pass. These numbers highlight that i) sustainability differences across *architectures* are dominated by backbone FLOPs, while ii) swapping Euclidean harmonic loss for alternative distances or baselines changes only the final classifier head, adding an $O(Cd)$ cost that is negligible compared to the convolutional / transformer body. Consequently, the per–step FLOP budget is effectively distance–invariant, and our sustainability

comparisons across losses can be interpreted as differences in optimization dynamics (steps-to-target, stability) rather than raw arithmetic cost.

Table 16: Per-sample FLOPs, GFLOPs, and parameter counts for each backbone and dataset.

| Model | Dataset | In Ch. | H | W | #Cls | FLOPs | Params | GFLOPs |
|---|---|---|---|---|---|---|---|---|
| CNN | CIFAR10 | 3 | 32 | 32 | 10 | 12307072 | 545098 | 0.0123 |
| CNN | CIFAR100 | 3 | 32 | 32 | 100 | 12330112 | 556708 | 0.0123 |
| CNN | MarathiSign | 3 | 32 | 32 | 43 | 12315520 | 549355 | 0.0123 |
| CNN | MNIST | 1 | 28 | 28 | 10 | 8520064 | 421642 | 0.0085 |
| CNN | TinyImageNet | 3 | 224 | 224 | 200 | 602966144 | 25735432 | 0.6029 |
| MLP | CIFAR10 | 3 | 32 | 32 | 10 | 3413760 | 1707274 | 0.0034 |
| MLP | CIFAR100 | 3 | 32 | 32 | 100 | 3459840 | 1730404 | 0.0034 |
| MLP | MarathiSign | 3 | 32 | 32 | 43 | 3430656 | 1715755 | 0.0034 |
| MLP | MNIST | 1 | 28 | 28 | 10 | 1070848 | 535818 | 0.0010 |
| MLP | TinyImageNet | 3 | 224 | 224 | 200 | 309536256 | 154769096 | 0.3095 |
| PVT | CIFAR10 | 3 | 32 | 32 | 10 | 38268630 | 12746560 | 0.0382 |
| PVT | CIFAR100 | 3 | 32 | 32 | 100 | 38314710 | 12769600 | 0.0383 |
| PVT | MarathiSign | 3 | 32 | 32 | 43 | 38285526 | 12755008 | 0.0382 |
| PVT | MNIST | 3 | 32 | 32 | 10 | 38268630 | 12746560 | 0.0382 |
| PVT | TinyImageNet | 3 | 224 | 224 | 200 | 1899590400 | 12795200 | 1.8995 |
| ResNet50 | CIFAR10 | 3 | 32 | 32 | 10 | 79618429 | 23472480 | 0.0796 |
| ResNet50 | CIFAR100 | 3 | 32 | 32 | 100 | 79987069 | 23656800 | 0.0799 |
| ResNet50 | MarathiSign | 3 | 224 | 224 | 43 | 4096080128 | 23540064 | 4.0960 |
| ResNet50 | MNIST | 1 | 28 | 28 | 10 | 60007460 | 23472480 | 0.0600 |
| ResNet50 | TinyImageNet | 3 | 224 | 224 | 200 | 4096723200 | 23861600 | 4.0967 |

## K  GEOMETRIC INSIGHTS

To better illustrate how different harmonic distances shape the embedding geometry, we visualize the last–layer representations of ResNet50 on MNIST (see Figure 11) and CIFAR10 (see Figure 12) using 2D PCA, with class prototypes overlaid as markers. For the Euclidean harmonic head, the class clusters are roughly spherical and separated by (approximately) straight boundaries in the projection: decision regions are controlled mainly by radial distance to each prototype, yielding isotropic attraction basins around each center.

Under Cosine harmonic loss, the picture changes markedly. Features and prototypes concentrate on (or very near to) a common hypersphere, so the PCA plot shows clusters arranged along a circle. Classes are separated primarily by their angle rather than their norm, and decision boundaries correspond to angular bisectors between prototypes. This matches our geometric claim that cosine harmonic removes radial curvature and constrains optimization to an angular manifold: as training proceeds, points slide along the sphere towards their prototype, producing wide, smooth basins and stable gradient norms.

By contrast, Mahalanobis harmonic loss induces anisotropic curvature. After whitening by $\Sigma^{-1/2}$, the decision boundaries are linear, but in the original feature space they correspond to ellipsoidal contours. In the PCA plots this appears as elongated clusters and distorted attraction basins around prototypes, with some directions exhibiting much tighter concentration than others. When the empirical covariance is well–conditioned this yields very sharp, well–separated clusters (high variance concentration), but when eigenvalues are highly unbalanced the same anisotropy can make optimization more sensitive to particular directions.

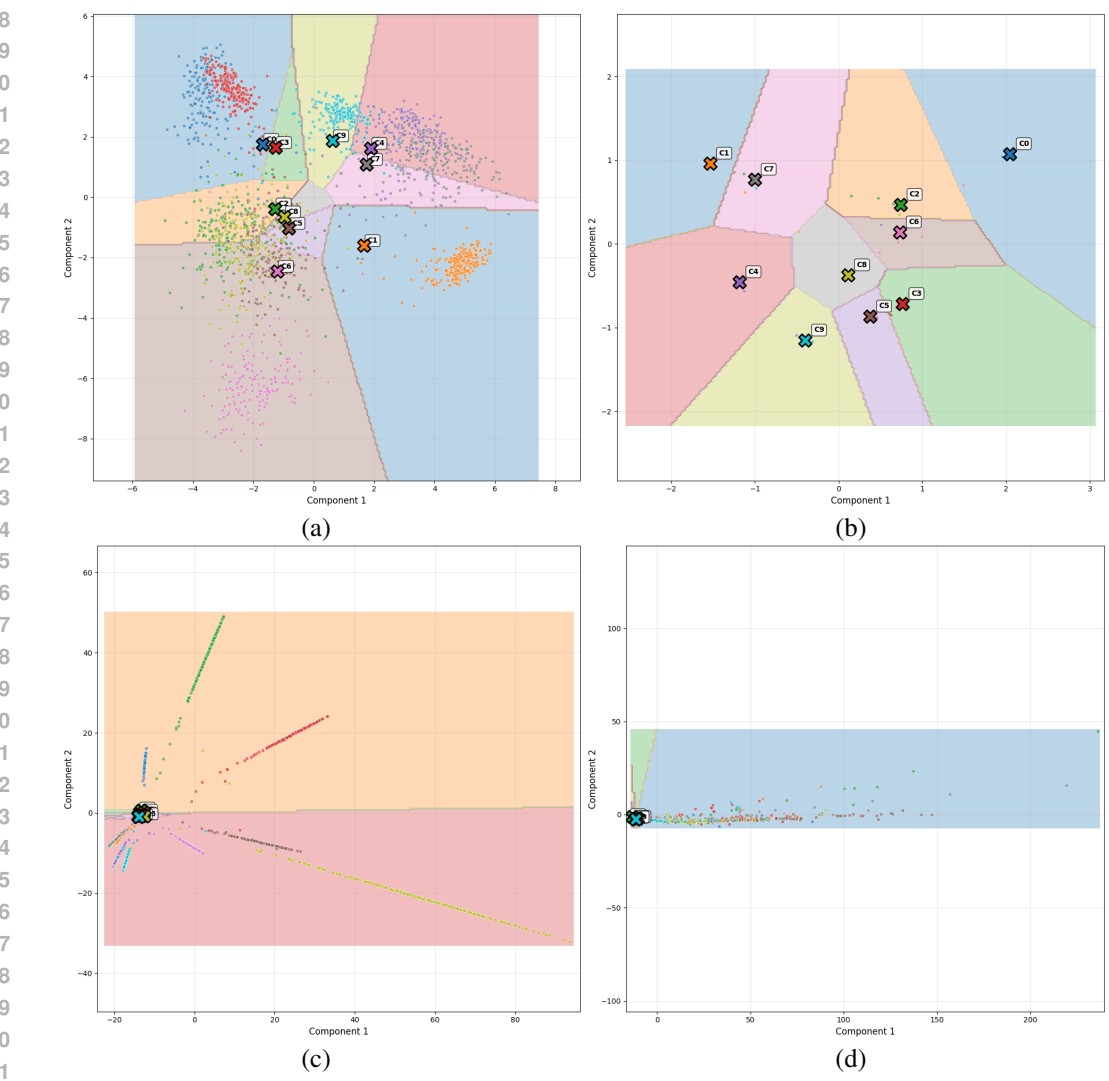

Figure 11: Geometric effect of distance–based harmonic losses on ResNet50 embeddings (MNIST). From top to bottom: Baseline (a), Euclidean harmonic loss (b), cosine harmonic loss (c), and Mahalanobis harmonic loss (d).

## L  ADDITIONAL RESULTS

### L.1  VISION: TABLES

The empirical evaluation of non-Euclidean harmonic losses across MNIST, CIFAR-10, and CIFAR-100 with MLP, CNN, and ResNet50 backbones reveals several consistent patterns.

**Model Performance.** Cosine distance emerges as the most reliable performer across architectures and datasets. In both stable and unstable variants, cosine harmonic loss consistently improves test accuracy and F1 relative to Euclidean, with gains most pronounced in deeper models (CNNs and ResNets) and in medium-complexity datasets such as CIFAR-10. Bray–Curtis offers modest gains in certain contexts but is less consistent, while Mahalanobis can improve accuracy on simple datasets (e.g., MNIST) but often lags behind cosine in more challenging regimes. Euclidean harmonic loss, while better than cross-entropy in terms of stability, is consistently outperformed by cosine-based alternatives.

**Interpretability.** Distances strongly reshape the geometry of the learned representations. Cosine and Bray–Curtis often yield large improvements in explained variance (EV), indicating more com-

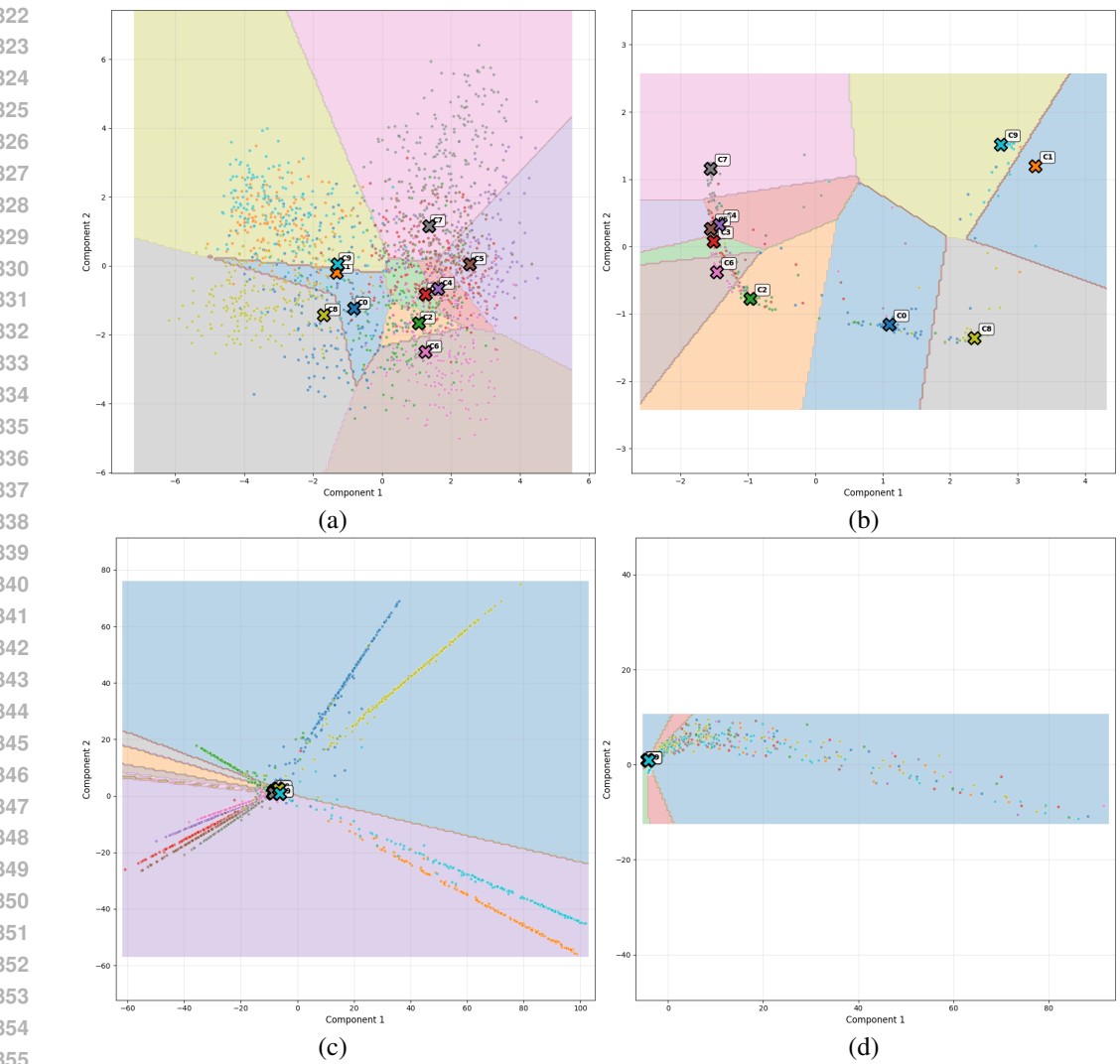

Figure 12: Geometric effect of distance–based harmonic losses on ResNet50 embeddings (CI-FAR10). From top to bottom: Baseline (a), Euclidean harmonic loss (b), cosine harmonic loss (c), and Mahalanobis harmonic loss (d).

pact feature spaces aligned with class prototypes. Mahalanobis produces the most dramatic gains in EV, frequently approaching full variance explanation, but this comes at the cost of stability and efficiency. Prototype coverage (PC90%) tends to shrink under cosine and Mahalanobis, highlighting sharper clustering effects: models assign fewer prototypes to cover 90% of variance, making the representation space more interpretable but less evenly distributed.

**Sustainability.** Sustainability outcomes mirror performance trends. Cosine distances typically reduce carbon emissions relative to Euclidean, in some cases by up to 40%, making them both effective and energy-efficient. Bray–Curtis shows mixed results, with occasional emission savings but less consistent behavior. Mahalanobis tends to incur higher emissions, reflecting the computational overhead of covariance estimation and matrix operations. Shallow architectures (MLPs) show less differentiation across distances in emissions, while deeper backbones amplify both the benefits (cosine) and costs (Mahalanobis).

**Trade-offs.** Taken together, the results confirm that distance choice is not neutral in harmonic loss. Cosine provides the most favorable balance across performance, interpretability, and sustainability, representing the strongest general-purpose alternative to Euclidean. Bray–Curtis occupies a middle ground, offering interpretability benefits without always delivering accuracy or efficiency gains.

Table 17: Results for CIFAR100 CNN. Parentheses: % changes w.r.t. Baseline (Cross-Entropy).

| Method | Acc | F1 | gCO$_2$eq | EV | PC90% |
|---|---|---|---|---|---|
| Baseline | 0.3795 | 0.3795 | 1.18 | 0.459295 | 49.3333 |
| Bray-Curtis (Norm.) | 0.3229 (-14.91%) | 0.3182 (-16.16%) | 0.8132 (30.94%) | 0.9094 (98%) | 2.6667 (94.59%) |
| Mahalanobis (Chol.) | 0.2927 (-22.86%) | 0.2921 (-23.04%) | 0.727 (38.25%) | 0.341 (-25.75%) | 50 (-1.35%) |
| Cosine (Unst.) | 0.2602 (-31.44%) | 0.2667 (-29.73%) | 2.1156 (-79.68%) | 0.5306 (15.52%) | 45 (8.78%) |
| Cosine (Stable) | 0.2501 (-34.09%) | 0.2516 (-33.71%) | 1.4263 (-21.14%) | 0.5216 (13.57%) | 45 (8.78%) |
| Euclidean | 0.2413 (-36.4%) | 0.2431 (-35.95%) | 1.2866 (-9.28%) | 0.4362 (-5.02%) | 50 (-1.35%) |

Table 18: Results for CIFAR100 MLP. Parentheses: % changes w.r.t. Baseline (Cross-Entropy).

| Method | Acc | F1 | gCO$_2$eq | EV | PC90% |
|---|---|---|---|---|---|
| Baseline | 0.2617 | 0.2582 | 0.85 | 0.285203 | 50.0 |
| Bray-Curtis (Norm.) | 0.226 (-13.64%) | 0.2191 (-15.15%) | 0.8938 (-5.35%) | 0.9843 (245.11%) | 1 (98%) |
| Mahalanobis (Chol.) | 0.1833 (-29.96%) | 0.1811 (-29.85%) | 1.0889 (-28.34%) | 0.0354 (-87.57%) | 50 (-0%) |
| Bray-Curtis (Abs.) | 0.1444 (-44.81%) | 0.1392 (-46.07%) | 2.1255 (-150.51%) | 0.5317 (86.43%) | 47.6667 (4.67%) |
| Cosine (Unst.) | 0.1237 (-52.74%) | 0.1186 (-54.06%) | 0.5064 (40.31%) | 0.3799 (33.21%) | 40.6667 (18.67%) |
| Euclidean | 0.119 (-54.53%) | 0.1222 (-52.69%) | 0.589 (30.58%) | 0.2437 (-14.55%) | 50 (-0%) |

Mahalanobis maximizes interpretability at a clear sustainability cost, making it attractive primarily when prototype clarity outweighs computational expense. Euclidean serves as a stable but suboptimal baseline.

**Conclusion.** This systematic study establishes that non-Euclidean harmonic losses provide a flexible and effective design space. In particular, cosine distance offers a compelling replacement for cross-entropy and Euclidean harmonic loss in vision tasks, consistently improving accuracy, interpretability, and sustainability. These findings position distance-tailored harmonic losses as a promising avenue for advancing deep learning models that are not only accurate but also more transparent and energy-conscious.

## L.2 VISION: SUSTAINABILITY

### L.2.1 MNIST

Figure 13 summarizes the *carbon deltas* (gCO$_2$eq relative to cross-entropy) when swapping the training objective for harmonic-loss variants on MNIST across four backbones.

**MLP.** Most distances reduce per–step emissions vs. cross-entropy (green bars), with the largest savings from heavier geometry that replaces the softmax/cross-entropy path (e.g., Mahalanobis/standardized, Chebyshev). Euclidean and Bray–Curtis yield modest savings; only a few variants show small positive overheads. Given MNIST's simplicity and the near-saturation accuracies, these reductions likely translate into *net* greener runs because steps-to-target are comparable.

**CNN.** A broad set of distances are carbon-negative vs. baseline. Again, standardized Mahalanobis/Chebyshev rank among the lowest-emission options; Bray–Curtis and Euclidean remain

Table 19: Results for CIFAR100 ResNet50. Parentheses: % changes w.r.t. Baseline (Cross-Entropy).

| Method | Acc | F1 | gCO$_2$eq | EV | PC90% |
|---|---|---|---|---|---|
| Baseline | 0.6983 | 0.6969 | 87.77 | 0.107216 | 50.0 |
| Cosine (Stable) | 0.7357 (5.35%) | 0.736 (5.61%) | 72.9745 (16.85%) | 0.5979 (457.66%) | 8 (84%) |
| Cosine (Unst.) | 0.7323 (4.87%) | 0.7332 (5.21%) | 71.7592 (18.24%) | 0.5857 (446.27%) | 8 (84%) |
| Bray-Curtis (Norm.) | 0.655 (-6.19%) | 0.6513 (-6.54%) | 106.4049 (-21.24%) | 0.7131 (565.08%) | 6 (88%) |
| Mahalanobis (Chol.) | 0.6274 (-10.15%) | 0.6239 (-10.47%) | 138.9317 (-58.3%) | 0.7353 (585.81%) | 17.5 (65%) |
| Euclidean | 0.7055 (1.03%) | 0.7062 (1.33%) | 97.432 (-11.01%) | 0.5679 (429.66%) | 25.5 (49%) |

Table 20: Results for CIFAR10 CNN. Parentheses: % changes w.r.t. Baseline (Cross-Entropy).

| Method | Acc | F1 | gCO$_2$eq | EV | PC90% |
|---|---|---|---|---|---|
| Baseline | 0.6278 | 0.6269 | 1.12 | 0.688081 | 9.0 |
| Mahalanobis (Chol.) | 0.6644 (5.82%) | 0.6642 (5.95%) | 1.1139 (0.68%) | 0.4752 (-30.93%) | 50 (-455.56%) |
| Bray-Curtis (Norm.) | 0.6597 (5.08%) | 0.6551 (4.5%) | 1.1489 (-2.45%) | 0.8913 (29.54%) | 4.3333 (51.85%) |
| Minkowski (p=3.0) | 0.6589 (4.95%) | 0.6593 (5.17%) | 1.1598 (-3.42%) | 0.5425 (-21.15%) | 50 (-455.56%) |
| Cosine (Stable) | 0.6584 (4.87%) | 0.6566 (4.74%) | 1.1663 (-3.99%) | 0.647 (-5.97%) | 18.6667 (-107.41%) |
| Euclidean | 0.6495 (3.45%) | 0.6476 (3.31%) | 1.1228 (-0.12%) | 0.6582 (-4.34%) | 14.3333 (-59.26%) |

Table 21: Results for CIFAR10 MLP. Parentheses: % changes w.r.t. Baseline (Cross-Entropy).

| Method | Acc | F1 | gCO$_2$eq | EV | PC90% |
|---|---|---|---|---|---|
| Baseline | 0.5397 | 0.5385 | 0.53 | 0.346504 | 47.0 |
| Bray-Curtis (Norm.) | 0.5224 (-3.21%) | 0.5201 (-3.41%) | 0.5264 (0.81%) | 0.967 (179.07%) | 1 (97.87%) |
| Mahalanobis (Chol.) | 0.5087 (-5.75%) | 0.5088 (-5.51%) | 0.458 (13.7%) | 0.0522 (-84.94%) | 50 (-6.38%) |
| Bray-Curtis (Abs.) | 0.4934 (-8.59%) | 0.4924 (-8.55%) | 0.6313 (-18.96%) | 0.2434 (-29.76%) | 50 (-6.38%) |
| Bray-Curtis (Std.) | 0.4931 (-8.64%) | 0.4935 (-8.35%) | 0.6435 (-21.25%) | 0.2906 (-16.14%) | 50 (-6.38%) |
| Euclidean | 0.4871 (-9.74%) | 0.4852 (-9.9%) | 0.4303 (18.92%) | 0.4303 (24.19%) | 42.3333 (9.93%) |

Table 22: Results for CIFAR10 ResNet50. Parentheses: % changes w.r.t. Baseline (Cross-Entropy).

| Method | Acc | F1 | gCO$_2$eq | EV | PC90% |
|---|---|---|---|---|---|
| Baseline | 0.843 | 0.8431 | 48.65 | 0.257211 | 50.0 |
| Cosine (Stable) | 0.9262 (9.87%) | 0.9262 (9.86%) | 40.6776 (16.39%) | 0.7559 (193.9%) | 5 (90%) |
| Cosine (Unst.) | 0.9234 (9.54%) | 0.9234 (9.53%) | 29.3968 (39.58%) | 0.761 (195.86%) | 5 (90%) |
| Bray-Curtis (Norm.) | 0.9193 (9.05%) | 0.9192 (9.02%) | 45.6222 (6.23%) | 0.7883 (206.49%) | 5 (90%) |
| Chebyshev (Std.) | 0.905 (7.36%) | 0.905 (7.34%) | 48.5505 (0.21%) | 0.9995 (288.59%) | 1 (98%) |
| Euclidean | 0.9185 (8.96%) | 0.9185 (8.94%) | 45.8759 (5.71%) | 0.683 (165.56%) | 25.5 (49%) |

Table 23: Results for MNIST CNN. Parentheses: % changes w.r.t. Baseline (Cross-Entropy).

| Method | Acc | F1 | gCO$_2$eq | EV | PC90% |
|---|---|---|---|---|---|
| Baseline | 0.9782 | 0.9782 | 1.19 | 0.585633 | 10.6667 |
| Bray-Curtis (Norm.) | 0.9889 (1.09%) | 0.9888 (1.09%) | 1.1348 (4.42%) | 0.7225 (23.38%) | 13.6667 (-28.12%) |
| Mahalanobis (Chol.) | 0.9879 (1%) | 0.9879 (0.99%) | 1.0639 (10.39%) | 0.4673 (-20.2%) | 36.3333 (-240.63%) |
| Minkowski (p=3.0) | 0.9877 (0.97%) | 0.9876 (0.96%) | 1.1154 (6.06%) | 0.4195 (-28.37%) | 49.3333 (-362.5%) |
| Hamming (Soft) | 0.9833 (0.52%) | 0.9832 (0.51%) | 1.1815 (0.49%) | 0.3089 (-47.26%) | 50 (-368.75%) |
| Euclidean | 0.9831 (0.5%) | 0.9831 (0.5%) | 1.1543 (2.78%) | 0.4413 (-24.65%) | 20.3333 (-90.62%) |

Table 24: Results for MNIST MLP. Parentheses: % changes w.r.t. Baseline (Cross-Entropy).

| Method | Acc | F1 | gCO$_2$eq | EV | PC90% |
|---|---|---|---|---|---|
| Baseline | 0.976 | 0.9758 | 0.55 | 0.565723 | 10.3333 |
| Cosine (Unst.) | 0.978 (0.2%) | 0.9778 (0.2%) | 0.5264 (3.58%) | 0.382 (-32.48%) | 10 (3.23%) |
| Mahalanobis (Chol.) | 0.9774 (0.14%) | 0.9771 (0.14%) | 0.5611 (-2.78%) | 0.092 (-83.74%) | 50 (-383.87%) |
| Cosine (Stable) | 0.9766 (0.06%) | 0.9764 (0.06%) | 0.5266 (3.54%) | 0.4033 (-28.71%) | 9.3333 (9.68%) |
| Chebyshev (Std.) | 0.9756 (-0.04%) | 0.9754 (-0.04%) | 0.5881 (-7.73%) | 0.7865 (39.03%) | 5.6667 (45.16%) |
| Euclidean | 0.9799 (0.4%) | 0.9798 (0.41%) | 0.5221 (4.35%) | 0.358 (-36.72%) | 9 (12.9%) |

Table 25: Results for MNIST ResNet50. Parentheses: % changes w.r.t. Baseline (Cross-Entropy).

| Method | Acc | F1 | gCO$_2$eq | EV | PC90% |
|---|---|---|---|---|---|
| Baseline | 0.9909 | 0.9909 | 29.36 | 0.420353 | 50.0 |
| Bray-Curtis (Norm.) | 0.9962 (0.52%) | 0.9961 (0.53%) | 25.2889 (13.86%) | 0.8453 (101.09%) | 4 (92%) |
| Cosine (Unst.) | 0.996 (0.51%) | 0.996 (0.52%) | 26.1851 (10.8%) | 0.6888 (63.87%) | 6 (88%) |
| Cosine (Stable) | 0.9953 (0.44%) | 0.9953 (0.45%) | 26.4064 (10.05%) | 0.6974 (65.91%) | 6 (88%) |
| Mahalanobis (Chol.) | 0.9938 (0.29%) | 0.9938 (0.3%) | 31.9246 (-8.75%) | 0.9966 (137.09%) | 1 (98%) |
| Euclidean | 0.9934 (0.25%) | 0.9934 (0.25%) | 24.457 (16.69%) | 0.9998 (137.84%) | 1 (98%) |

consistently frugal. Variants that introduce extra normalization or temperature schedules can erode part of the gain but rarely flip the sign.

**ResNet50.** The deepest convolutional model shows the *largest* per–step savings: many distances deliver substantial negative deltas relative to cross-entropy, suggesting that replacing the softmax loss with metric-based objectives amortizes well at this scale. Only a handful of choices (e.g., certain Chebyshev/Canberra parameterizations) incur small positive overheads.

**PVT (vision transformer).** In contrast to the CNN family, most distances *increase* per–step emissions over the baseline. The transformer's attention and normalization stack appears less amenable to the heavier distance computations; only a couple of standardized/normalized variants produce small savings. On PVT, greener training favors the lightest geometries or retaining cross-entropy.

**Takeaways.** i) On MNIST, distance-based harmonic losses are often *carbon-favorable* for MLP/C-NN/ResNet50, with the biggest gains on the deepest CNN; ii) these gains are not universal—PVT tends to pay a premium; iii) because test accuracy curves on MNIST converge similarly across losses, the per–step savings for CNN/ResNet50 likely convert into lower *end-to-end* energy. Practically, we recommend Euclidean/Bray–Curtis/standardized Mahalanobis for convolutional backbones, and cautious use (or kernel-fused, mixed-precision implementations) of heavier distances on transformer-style models. Reporting both per–step emissions and energy-to-target accuracy remains essential for fair sustainability claims.

L.2.2 CIFAR-10

Figure 14 reports carbon deltas in gCO$_2$eq relative to cross-entropy when training with harmonic-loss distances on CIFAR-10.

**MLP.** Most distances are *carbon–negative* versus baseline, yielding small–to–moderate per–step savings. A few choices incur mild overheads (rightmost bars), indicating that added normalization or temperature scheduling can offset the gains on shallow networks.

**CNN.** The pattern strengthens: a broad set of distances reduce per–step emissions relative to cross-entropy. Only a handful of variants sit near zero or slightly positive, suggesting that, for convolutional encoders on CIFAR-10, metric-based objectives are generally more frugal per step.

**ResNet50.** Savings are *uniform and largest*: all distances fall below the baseline, with substantial negative deltas. This indicates that replacing the softmax loss amortizes particularly well at depth/width, likely due to better kernel utilization and reduced softmax/backprop overhead relative to the total compute.

**PVT (vision transformer).** Most distances are again carbon–negative, though the spread is narrower than ResNet50 and a couple of variants hover around parity or slightly positive. Transformers benefit, but less dramatically than deep CNNs.

**Takeaways.** i) On CIFAR-10, distance-based harmonic losses are typically *greener per step* for CNN/ResNet50/PVT, with the strongest effect on ResNet50; ii) MLP shows mixed but mostly favorable outcomes; iii) because our accuracy-vs-epoch curves on CIFAR-10 show similar or faster convergence for several distances, these per–step gains are likely to translate into lower *end-to-end* energy for deep backbones. Practically, we recommend adopting the more frugal distances for convolutional and transformer models and pairing per–step reports with *energy-to-target-accuracy* to substantiate sustainability claims.

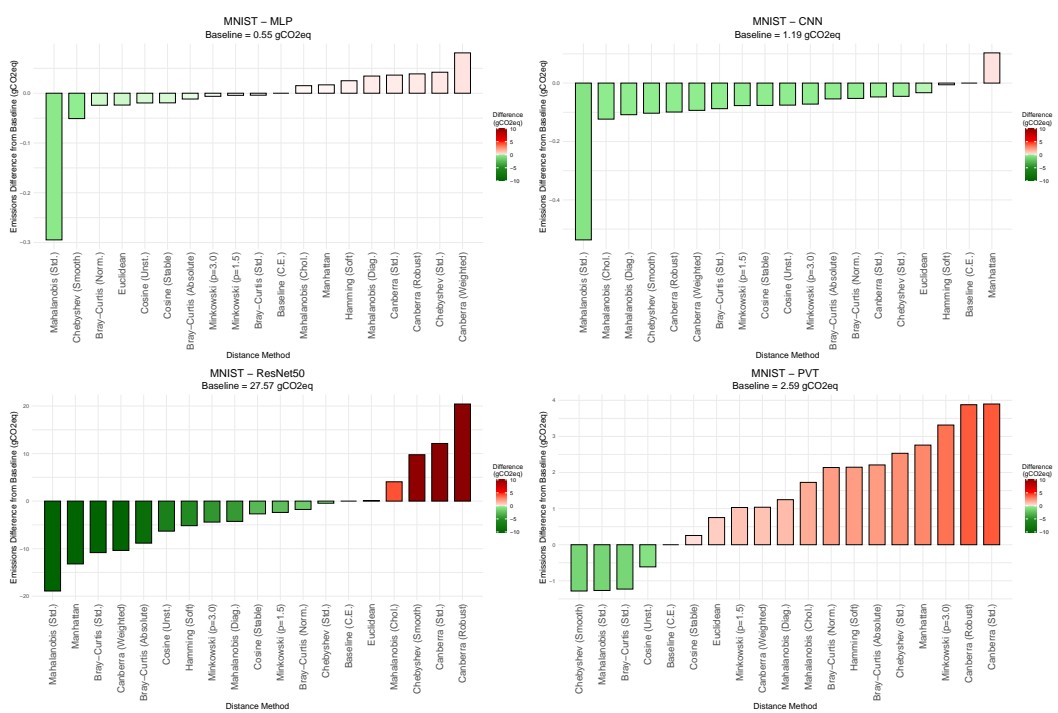

Figure 13: Carbon emission differences for MNIST across four model backbones (MLP, CNN, ResNet50, PVT) when replacing cross-entropy with harmonic loss variants. Bars show the emission difference in grams of $CO_2$eq relative to the baseline (cross-entropy). Values above zero indicate higher emissions than baseline, while negative values indicate greener, more sustainable outcomes.

### L.2.3   CIFAR-100

Figure 15 shows the carbon *delta* (g$CO_2$eq vs. cross-entropy) when training with harmonic-loss distances on CIFAR-100.

**MLP.** Savings are modest and *geometry–dependent*. Light/standardized variants (e.g., cosine, Euclidean, some Minkowski/Canberra settings) are carbon–negative, while heavier norms and covariance–based Mahalanobis parameterizations flip to positive overheads. On shallow models, extra normalization steps can outweigh gains.

**CNN.** A broad swath of distances are carbon–negative relative to the 1.18 g$CO_2$eq baseline; several Mahalanobis and Bray–Curtis settings deliver the largest per–step reductions. A few choices (e.g., certain cosine/Canberra/Minkowski configurations) hover near parity or slightly positive, indicating mild architecture sensitivity.

**ResNet50.** The deepest convolutional model exhibits a *mixed but wide* spread: many distances achieve substantial savings (left cluster of dark-green bars), yet others incur clear premiums (right cluster). Thus, distance choice materially changes footprint at scale. Notably, cosine variants are among the frugal options here, whereas some Chebyshev/Minkowski/Bray–Curtis (absolute) settings are costlier.

**PVT (vision transformer).** Most distances are *carbon–positive* vs. the 3.67 g$CO_2$eq baseline, with only a couple of standardized/smoothed variants slightly negative. As on MNIST/CIFAR-10, the attention/normalization stack appears less amenable to heavier metric computations.

**Takeaways.** i) On CIFAR-100, harmonic distances can be *greener per step* for CNNs and selectively for ResNet50, but PVT generally pays a premium; ii) cosine tends to be frugal on deeper CNNs (and competitive on MLP), aligning with its strong accuracy dynamics, whereas several Mahalanobis/Minkowski/Chebyshev configurations increase emissions unless they deliver clear quality

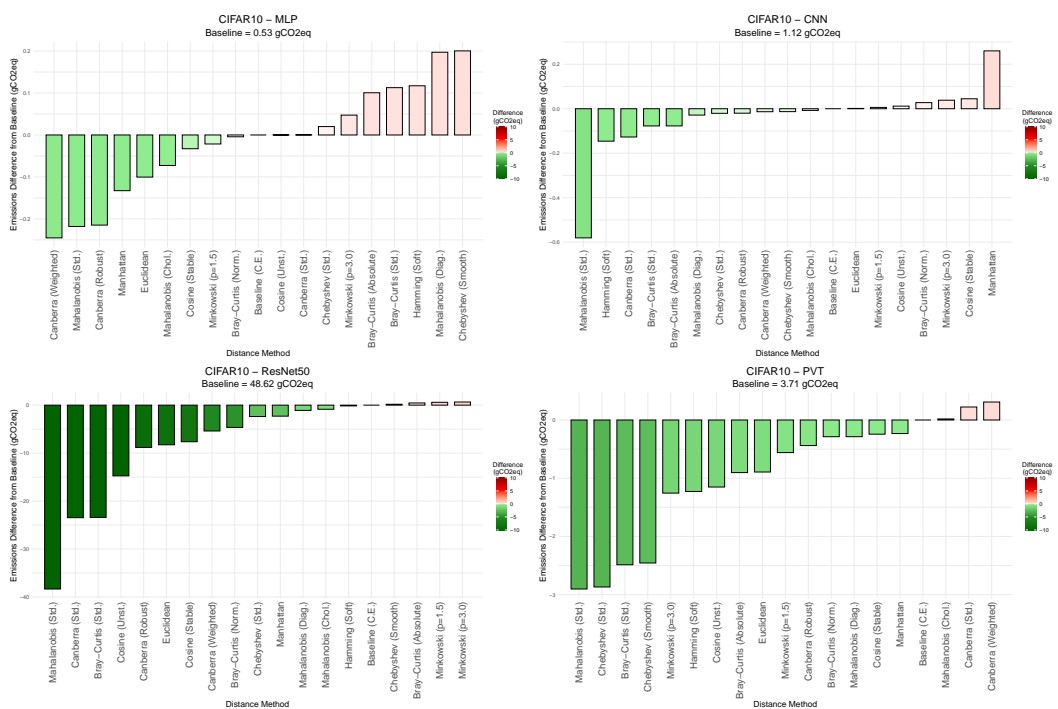

Figure 14: Carbon emission differences for CIFAR10 across four model backbones (MLP, CNN, ResNet50, PVT) when replacing cross-entropy with harmonic loss variants. Bars show the emission difference in grams of $CO_2$eq relative to the baseline (cross-entropy). Values above zero indicate higher emissions than baseline, while negative values indicate greener, more sustainable outcomes.

gains; iii) because CIFAR-100 accuracy converges differently across distances, claims of sustainability should couple per–step deltas with *energy-to-target-accuracy/perplexity*. Practically, prefer cosine/Euclidean/standardized Bray–Curtis (and selected Mahalanobis settings that are both stable and frugal) for CNN/ResNet50, and use kernel fusion + mixed precision if heavier geometries are needed on transformer backbones.

**Insights across datasets:** A clear trend emerges across datasets: **transformer models (PVT)** often incur higher emissions with distance-based harmonic losses, particularly on CIFAR-100 (see Figure 15), whereas **convolutional and residual networks** (CNN, ResNet50) frequently yield greener outcomes (see results in Figures 13 – 15). The sustainability benefit is especially pronounced when distances incorporate robustness (Hamming-gumbel, Canberra-robust) or covariance awareness (Mahalanobis-diagonal). Simpler datasets like MNIST show limited differences, while CIFAR-10 and CIFAR-100 highlight the greater impact of distance choice on carbon footprint.

**Cross-architecture insights:** **MLPs** present a limited sustainability differences; emissions remain close to baseline across all distances. With **CNNs**, multiple distances (Hamming-gumbel, Mahalanobis-diagonal, Canberra-weighted) consistently reduce emissions, showing CNNs benefit most from harmonic loss efficiency. In **PVT**, harmonic losses generally increase emissions, especially on CIFAR-100, highlighting potential overhead in attention-based models. **ResNet50** demonstrates an effective integration with several distances (Hamming, Canberra, Bray–Curtis), which achieve significant reductions in emissions over baseline, indicating that deep CNNs can combine effectiveness with sustainability.

Overall, the sustainability analysis shows that harmonic losses can improve or degrade carbon efficiency depending on the backbone and dataset. The choice of distance measure therefore plays a critical role not only in accuracy but also in environmental impact, reinforcing the need for holistic evaluation across the accuracy–sustainability–interpretability triangle.

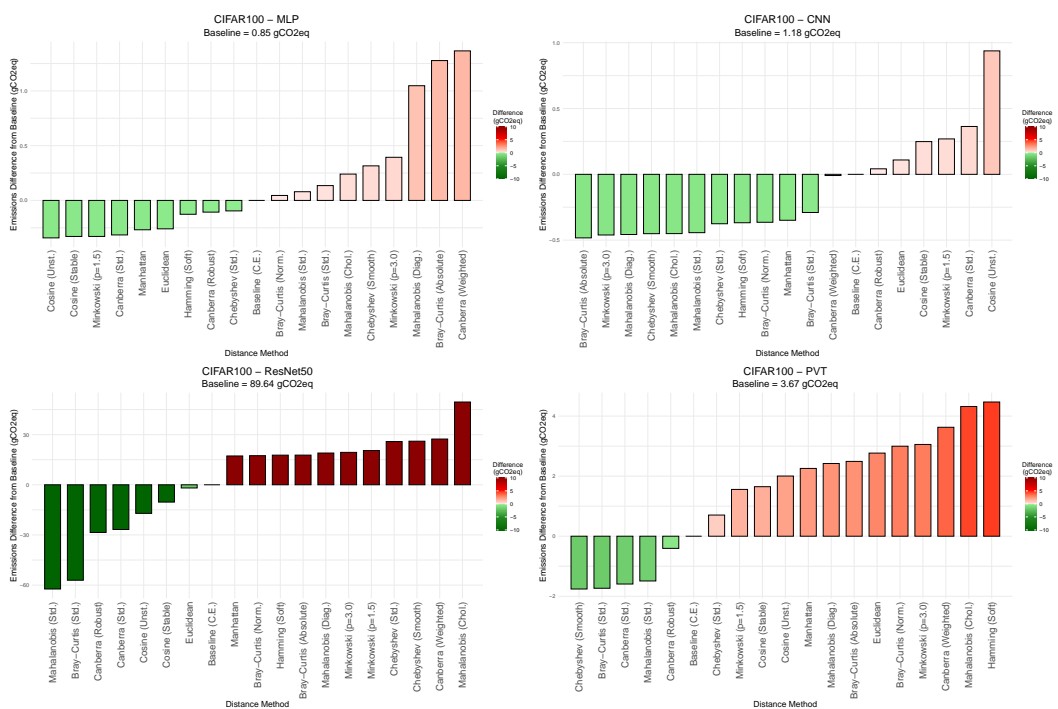

Figure 15: Carbon emission differences for CIFAR100 across four model backbones (MLP, CNN, ResNet50, PVT) when replacing cross-entropy with harmonic loss variants. Bars show the emission difference in grams of $CO_2$eq relative to the baseline (cross-entropy). Values above zero indicate higher emissions than baseline, while negative values indicate greener, more sustainable outcomes.

### L.3    LANGUAGE: SUSTAINABILITY

Figure 16 reports *per–1k-step* carbon differences (g$CO_2$eq) when replacing cross-entropy with distance-based harmonic losses for BERT, GPT, and QWEN. Positive bars indicate higher emissions than the cross-entropy baseline (annotated atop each subplot).

**Overall.** Across all three backbones, distance-based losses tend to *increase* per–1k-step emissions relative to cross-entropy. The magnitude of overhead correlates with the computational complexity of the distance: lightweight cosine variants add the least overhead, while Mahalanobis and Minkowski incur the most.

**BERT.** Cosine (simple or temperature-scaled) yields small overheads (low single-digit g$CO_2$eq over a 7.87 g$CO_2$eq baseline), suggesting that the extra normalization and dot-product operations have modest cost. Euclidean and Bray–Curtis sit mid-pack, whereas Mahalanobis (Cholesky/standard/-diagonal) and Minkowski ($p > 2$) are consistently more carbon intensive per 1k steps.

**GPT.** All distances increase emissions over the 60.36 g$CO_2$eq baseline, with a clearer spread: cosine remains the most frugal among alternatives; Euclidean and Manhattan are mid-range; Mahalanobis (any parameterization) and Minkowski/L2 are the heaviest. This indicates that the per-step FLOPs and memory traffic of covariance-related computations (and higher-order norms) become more pronounced at GPT scale.

**QWEN.** For this larger model (baseline 75.29 g$CO_2$eq), the methods we evaluated (Minkowski/L2 and Euclidean) both raise per–1k-step emissions, with Minkowski/L2 showing a substantial increase. Although the set of distances is smaller here, the pattern mirrors GPT: heavier metrics cost more per step as model width/depth grows.

**Implications.** i) If *Green AI* considerations are primary, cosine-based harmonic losses are the most promising drop-in replacements, especially on encoder-style models (BERT). ii) Mahalanobis and

Minkowski should be justified by clear accuracy or stability gains, as they carry the largest per-step carbon premiums. iii) Reported values are per–1k-step; end-to-end footprint also depends on *steps-to-target-quality*. Thus, a distance that reduces time-to-accuracy could still yield net carbon savings even with higher per-step cost.

**Summary.** Distance choice in harmonic loss is not carbon-neutral: cosine variants introduce minimal overhead; Euclidean/Bray–Curtis are moderate; Mahalanobis/Minkowski are expensive. Any claimed performance gains from richer geometries should be weighed against these systematic energy costs, preferably via *energy-normalized* quality metrics (e.g., accuracy per kWh).

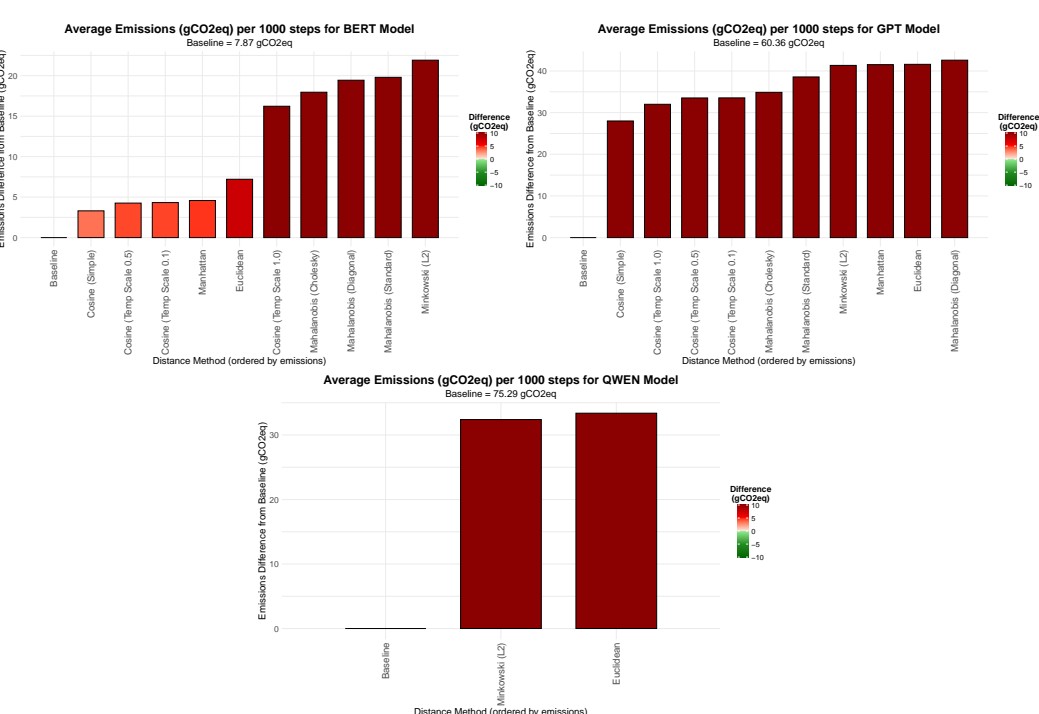

Figure 16: Carbon emission differences for LLM pretraining on OpenWebText (BERT, GPT2, QWEN) when replacing cross-entropy with harmonic loss variants. Bars show the emission difference in grams of $CO_2$eq relative to the baseline (cross-entropy). Values above zero indicate higher emissions than baseline, while negative values indicate greener, more sustainable outcomes.

### L.4 LANGUAGE: INTERPRETABILITY

Mechanistic and representation-level interpretability of large language models (LLMs) increasingly leverages the hypothesis that internal activations admit *approximately linear* structure: many features behave like directions in an activation space, and linear operations can steer or probe them (Elhage et al., 2022; Huben et al., 2024; Turntrout, 2023). Within this paradigm, Principal Component Analysis (PCA) is a simple, well-understood lens for: i) summarizing dominant sources of variance in activations; ii) stabilizing analyses by denoising; and (iii) producing human-auditable axes that can be inspected, correlated with concepts, and tracked over time.

Given a layer $\ell$ with residual-stream activations $H_\ell \in \mathbb{R}^{N \times d}$ collected across $N$ tokens (or prompts), PCA factorizes $H_\ell$ via SVD to yield orthogonal directions $\{u_k\}_{k=1}^d$ ordered by explained variance. In practice this supports:

1. **Concept probing and visualization.** Projections onto top PCs often align with semantically meaningful contrasts; e.g., the first PC of GPT-style embeddings correlated with human well-being judgments in zero-shot tests (FAR AI, 2023), and per-layer PCA can reconstruct or predict response modes in GPT-2 (Jorgensen, 2023).

2. **Diagnosing and localizing phenomena.** Layer-wise or head-wise PCA reveals where variance concentrates, helping localize depth at which concepts emerge or consolidate (complementary to linear probing) (Zhou et al., 2024). Tracking *subspace distance* across checkpoints detects representational drift during fine-tuning or domain shift.

3. **Sanity checks and baselines.** With growing interest in sparse autoencoders (SAEs) for monosemantic features (Huben et al., 2024), PCA serves as a transparent baseline decomposition: if SAEs meaningfully improve sparsity/faithfulness over PCA while matching reconstruction, that strengthens the interpretability claim (Templeton et al., 2023).

PCA is most compelling under: a) approximately linear feature superposition and b) high signal-to-noise in dominant directions. Toy and empirical studies argue that Transformers often encode many features as *directions* (superposition) (Elhage et al., 2022), and even simple linear additions to activations can steer model behavior (Turntrout, 2023). PCA then becomes an appropriate first-pass tool to:

- extract high-variance axes that frequently correlate with coherent features or tasks,
- reduce dimensionality before causal tests (e.g., ablate/project-out a PC and re-evaluate behavior),
- build compact surrogates (e.g., PCA embeddings for downstream analyses or compression) (Bengtsson et al., 2025; He et al., 2024).

Under widely observed linear-structure assumptions in Transformer activations, PCA offers an interpretable, testable starting point: it surfaces dominant directions, supports hypothesis generation, and provides quantitative targets for more advanced decompositions.

