# OpenReview forum: "Non-Euclidean Harmonic Losses"
_ICLR.cc/2026/Conference — ICLR 2026 Conference Desk Rejected Submission_

### Official Review · Reviewer_jU1M · 2025-10-31

**Soundness:** 3
**Presentation:** 3
**Contribution:** 3
**Rating:** 6
**Confidence:** 3

**Summary:**

This paper extends harmonic loss beyond its original Euclidean form to explore how different distance metrics influence deep-learning model performance, interpretability, and sustainability. The authors replace the Euclidean distance in harmonic loss with a broad family of metrics (cosine, Manhattan, Chebyshev, Minkowski, Bray–Curtis, Mahalanobis, etc.) and systematically benchmark them on both vision (MNIST, CIFAR-10/100, ResNet, PVT) and language (BERT, GPT, Qwen) tasks. Results show that cosine-based harmonic losses consistently yield smoother optimization, improved gradient stability, clearer feature structures, and lower carbon emissions compared to cross-entropy and Euclidean baselines. Bray–Curtis distances enhance interpretability by concentrating feature space geometry, while Mahalanobis offers strong representation clarity at a computational cost. Theoretical analysis establishes finite minimizers and PAC-Bayes generalization bounds for 1-homogeneous distances, ensuring well-posed learning. Overall, the study demonstrates that choice of geometric distance crucially affects performance–interpretability–sustainability trade-offs, positioning non-Euclidean harmonic losses as simple, interpretable, and energy-efficient replacements for cross-entropy in both vision and language models

**Strengths:**

1.	Comprehensive extension of harmonic loss — The paper systematically generalizes harmonic loss beyond Euclidean distance, offering the first large-scale comparison of diverse distance metrics across both vision and language domains.
	2.	Strong empirical validation — Extensive experiments on multiple architectures (CNNs, ResNet, PVT, GPT, BERT, Qwen) demonstrate consistent advantages of cosine-based harmonic losses in accuracy, stability, and sustainability.
	3.	Improved interpretability — The proposed framework yields more structured, low-dimensional representations where class prototypes have clear geometric meaning, aligning with interpretability-by-design principles.
	4.	Energy efficiency and Green AI contribution — By including CodeCarbon-based measurements, the paper quantifies training emissions and shows that certain non-Euclidean distances reduce carbon footprint compared to cross-entropy.
	5.	Solid theoretical grounding — The authors provide proofs for scale invariance, finite minimizers, and PAC–Bayes generalization bounds, giving mathematical support to the empirical results.
	6.	Practical simplicity and reproducibility — The non-Euclidean harmonic losses can be implemented as a “drop-in replacement” for standard classifier heads with minimal code changes, and the open-source repository ensures reproducibility and accessibility for future research.

**Weaknesses:**

1.	Limited novelty in concept — The work mainly extends an existing loss (harmonic loss) rather than introducing a fundamentally new framework.
	2.	Empirical results lack statistical depth — The paper reports averages but provides limited statistical significance analysis or error bars.
	3.	High computational cost for some metrics — Distances like Mahalanobis add complexity and may reduce scalability despite interpretability gains.
	4.	Insufficient real-world validation — Experiments are limited to standard benchmarks (MNIST, CIFAR, OpenWebText) without testing on large-scale or practical applications.

**Questions:**

Please check weaknesses, and try to argue them, I will definitely read your response, good luck!

---

> ### Author Response · Authors · 2025-11-22
>
> We thank the reviewer for the positive and encouraging comments, especially regarding the clarity of presentation, strong empirical validation, interpretability gains, sustainability contributions, theoretical grounding, and practical simplicity of our approach. Below we address all concerns:
>
> ## W1/Q1
> We agree that our work builds upon prior harmonic loss formulations. However, our contribution is not merely substituting the metric. We believe that our work sheds lights on at least three additional perspectives not considered in previous works:  i) *Geometric generalization*: We show that harmonic loss extends coherently beyond Euclidean geometry to any 1-homogeneous distance, and we provide new theoretical results that apply to a broad class of distances; ii) *Multi-domain systematic characterization*: Ours is the first study that evaluates many distances across vision and LLM pretraining under a unified tri-axis protocol (performance + interpretability + sustainability); iii) *New empirical phenomena*: We uncover several previously unreported findings, including: a) cosine harmonic loss reduces emissions while improving stability on both CNNs and LLMs, b) Bray-Curtis maximizes EV and prototype clarity across all architectures, and c) Mahalanobis creates extreme concentration that amplifies prototype hierarchy but raises energy cost.  These insights emerge only when analyzing these distances side-by-side. Thus, we argue that, while the conceptual modification is simple, the geometric, empirical, and theoretical insights gained are substantial and novel.
>
> ## W2/Q2
> Thank you for this helpful suggestion. We included statistical validation (Wilcoxon Statistical Tests) in the revised version of the manuscript (see **Appendix E**).  Additionally, for each setting, we provide confidence intervals across three random seeds and plot the mean trajectory together with a shaded 95% CI.
>
> ## W3/Q3
> We acknowledge that Mahalanobis variants increase per-step compute. This is consistent with their covariance estimation and adaptive geometry. However, other distances, such as Cosine and Bray-Curtis, which are consistently strong in our study, have equal or lower computational cost than Euclidean. Thus, while some distances are more expensive, others offer a more convenient trade-off between efficiency/accuracy/explainability. To address this comment, we refined our three-axis discussion of findings.
>
> ## W4/Q4
> This is a fair point. We wish to emphasize that our goal was to establish a controlled, geometry-focused study across modalities. To address this comment, we are performing experiments with larger GPT models (significantly increased number of heads, leading to larger number of parameters), and extending our vision experiments with larger real-world datasets such as Marathi Sign Language and TinyImageNet. We have already added most of the vision experiments in the main paper, and will add larger GPT models shortly.
>
> We thank the reviewer for the positive assessment and constructive feedback.  Our revisions will address all questions by i) broadening interpretability probes, ii) adding statistical rigor, iii) clarifying computational trade-offs, and iv) expanding the evaluation beyond small benchmarks. We hope this resolves the concerns and further highlights the practical and conceptual value of non-Euclidean harmonic losses.

---

### Official Review · Reviewer_mGc1 · 2025-11-01

**Soundness:** 3
**Presentation:** 3
**Contribution:** 2
**Rating:** 6
**Confidence:** 2

**Summary:**

The paper extends harmonic loss from Euclidean distance to a family of non-Euclidean metrics (cosine, Minkowski/Lp, Chebyshev, Manhattan, Canberra, Bray–Curtis, Mahalanobis, etc.) and evaluates them across vision backbones (MLP, CNN, ResNet-50, PVT) on MNIST/CIFAR-10/100 and LLM pretraining setups (GPT/BERT/Qwen) on OpenWebText. The study targets a three-way evaluation: performance, interpretability (via PCA structure/prototype semantics), and sustainability (CodeCarbon emissions). The headline claim is that cosine-based harmonic heads offer the best all-round trade-off (accuracy/stability/structure/emissions), with Bray–Curtis and Mahalanobis trading off efficiency for added structure. The paper also provides theoretical conditions for scale invariance and finite minimizers under 1-homogeneous distances and a margin-style PAC-Bayes generalization view.

**Strengths:**

1. **Clear, well-motivated extension**: Replacing Euclidean distance in the harmonic link with a registry of distances is simple yet impactful, and the DistLayer abstraction underscores plug-and-play practicality.
2. **Comprehensive empirical sweep**: The study spans multiple architectures and two domains (vision/LLMs), with consistent protocol and multi-criteria reporting (accuracy/F1, PCA structure, emissions). Radar summaries are helpful to see trade-offs.
3. **Takeaway with practical value**: Cosine emerges as a robust default across settings; Bray–Curtis and Mahalanobis are positioned as “interpretability-forward” choices with known costs—useful guidance for practitioners.
4. **Theory that matches the objective’s spirit**: The scale-invariance/finite-minimizer result for 1-homogeneous distances (harmonic link) and a margin-style PAC-Bayes perspective provide conceptual support for the geometry choices.

**Weaknesses:**

1. **Interpretability probes are narrow**: PCA concentration and prototype semantics are informative, but additional probes (e.g., class-conditioned separability margins, cluster purity/ARI, linear probe transfer, or prototype visualization fidelity for vision) would broaden interpretability evidence beyond PCA variance ratios.
2. **LLM metrics need anchoring**: The paper introduces “Gradient Stability,” “Model Health,” “Clipping Quality,” and “Learning Quality,” but clearer operational definitions, units, and relationships to widely accepted metrics (loss perplexity curves, training variance, overflow rates) would help reproducibility/interpretability.
3. **Novelty & Relation to Prior Work**: The paper does not re-introduce harmonic loss but broadens its geometry. This is incremental conceptually yet substantial empirically/thematically (interpretability+sustainability).

**Questions:**

1. Is cosine temperature sensitivity? and is it a simple heuristic for setting it?

---

> ### Author Response · Authors · 2025-11-22
>
> We thank the reviewer for the positive assessment, especially regarding the clarity of exposition, the comprehensive empirical analysis, and the practical guidance emerging from the study. Below we address all concerns.
>
> # Weaknesses
>
> ## W1
> We appreciate the helpful suggestions. Our goal was to provide a quantitative, architecture-agnostic interpretability probe applicable to both vision and LLMs. PCA concentration was chosen because: i) it directly reflects prototype alignment in harmonic losses, ii) it is widely used to study emergent linear structures in deep networks, and iii) it enables comparable measurement across modalities. To address your concern, we are currently working on extracting prototype visualizations to investigate whether non-Euclidean choices amplify class-specific structure. We will update the paper accordingly.
>
> ## W2
> Thank you for this pointer. These metrics were designed to capture the training-time phenomena that are not visible through loss/perplexity alone. In the updated paper, we provided explicit definitions, units, and connections to standard measures. All four diagnostics are anchored to standard LLM training quantities. For example gradient variance (Gradient Stability),   representation covariance (Model Health), and perplexity-improvement rate (Learning Quality).  Additionally, we extracted visualizations with Perplexity, as a very widely accepted metric for clarity and reproducibility.
>
> ## W3
> Harmonic loss itself is not new, but our contribution is in i) extending the harmonic link to distinct geometries, ii) clarifying theoretical conditions that generalize beyond Euclidean distance, and iii) conducting the first systematic multi-axis evaluation (accuracy + interpretability + sustainability) across both vision and LLM pretraining. This combination leads to actionable design insights (e.g., cosine as a robust default across small/large-scale models, geometry-dependent sustainability effects).
>
> ---
>
> # Questions
>
> ## Q1
> Yes, cosine is temperature-sensitive. Empirically we find that: i) temperatures in the range $[0.1, 0.5]$ consistently stabilize optimization, ii) values $<0.1$ over-concentrate gradients, iii) values $>1$ approach Euclidean-like behavior. A robust heuristic emerges from both theory and ablations:  $T^\ast \approx \frac{1}{\sqrt{d}}$, where $d$ is the representation dimension. This matches the scale-invariance properties of the harmonic link and aligns gradients across distances.
>
> We thank the reviewer again for recognizing the strengths and for surface-level concerns that we can fully address via i) richer interpretability probes, ii) clearer LLM metric definitions and adoption of Perplexity, and iii) a brief discussion of our findings related to temperature for cosine. We believe these clarifications further strengthen the contribution and enhance reproducibility.

---

> > ### Comment · Reviewer_mGc1 · 2025-11-25
> > **Reponse to authors**
> >
> > I appreciate the authors for clarification of my question. Most of my concerns are addressed and I decide to maintain my score.

---

### Official Review · Reviewer_f5hr · 2025-11-01

**Soundness:** 3
**Presentation:** 4
**Contribution:** 3
**Rating:** 4
**Confidence:** 4

**Summary:**

The authors extend Euclidean harmonic losses (explained to be a formulation that replaces softmax logit normalization with a normalization based on distances to class prototypes) to “non-Euclidean” harmonic losses that are essentially replace 2-norm with different distance metrics such as 1-norm, inf-norm, p-norm, etc. The authors evaluate the effect of these loss functions on multiple models trained on image classification and on LLM prediction in terms of (1) accuracy, (2) interpretability, and (3) sustainability or carbon footprint. Overall, the paper is very well written and thoroughly evaluated, but several elements of the approach are not clear from the writing and should be better clarified/explained.
The authors extend Euclidean harmonic losses (explained to be a formulation that replaces softmax logit normalization with a normalization based on distances to class prototypes) to “non-Euclidean” harmonic losses that are essentially replace 2-norm with different distance metrics such as 1-norm, inf-norm, p-norm, etc. The authors evaluate the effect of these loss functions on multiple models trained on image classification and on LLM prediction in terms of (1) accuracy, (2) interpretability, and (3) sustainability or carbon footprint. Overall, the paper is very well written and thoroughly evaluated, but several elements of the approach are not clear from the writing and should be better clarified/explained.

**Strengths:**

- Paper is extremely well written and generally clear in terms of concept.
- The three axes of evaluation are interesting and serve as an excellent example for the community. Many portions of the analysis are interesting.
- This paper is addressing an interesting and growing line of research relevant to the ICLR and broader machine learning community.

**Weaknesses:**

- Some details, such as where exactly the harmonic loss is implemented is unclear in the main text. Does this normalization occur at every layer, wherever there is a softmax, or only at the final classification layer?
- The work appears incremental, in that modifying the distance function is a small change, even if thoroughly evaluated. The start of the paper suggests greater insights into the geometry of the learned representations, but this is not really provided pictorally or via metrics in the evaluation section.
- I am not sure “non-Euclidean” is an appropriate framing for the paper. Certainly the distances investigated are not the 2-norm, but the vector spaces they (1-norm, p-norm, etc.) are operating on are still Euclidean.

**Questions:**

- How are the class prototypes determined or computed? This seems like a serious computational challenge if they are learned.
- Is there only 1 prototype per class? Why not 10 ? Is this not a hyperparameter?
- Are harmonic losses computed at every layer or only at the final classification layer?
- The paper refers to the “learned representation” at various places. Does this refer to intermediate layers or the last logit layer before the output?
- At the final layer, could the class prototypes just be 1-hot encoded vectors for each class? If so, are these losses that conceptually different from cross-entropy? Are they that different than standard MSE or MAE losses?
- In Line 270, what is the difference between interpretability and representation clarity?
- Can the authors clarify whether the interpretability measurement via explained-variance using PCA is done on the last logit layer or in the interior of the network?
- Why exactly is EV, which speaks to the geometry of the representation, a good measure of interpretability? This is not clear or well-justified.
- The main arguments rely significantly on the work of Baek et al. (cited at least 8 times, 4 times in the introduction). It would help to understand why the authors feel this work is meaningfully different, beyond comparison in multiple models.
- I really liked the presentation of this paper, but the current presentation is not clear enough in terms of implementation in the main text. Please convince me!

---

> ### Author Response · Authors · 2025-11-21
>
> We thank the reviewer for the thoughtful feedback and for highlighting the writing quality, the importance of the topic, and the usefulness of our three-axis evaluation.
>
> # Weaknesses
> ## W1
> Harmonic loss is applied only at the final classification layer, replacing the standard softmax + cross-entropy objective. All intermediate layers remain unchanged, and no normalization is applied. We clarified in **Section 3**.
>
> ## W2
> We appreciate this perspective. Our contribution is not simply "swapping norms": the moment a general distance replaces the Euclidean norm, nontrivial geometric and optimization phenomena may occur. Based on this comment, we were inspired to perform qualitative analyses of prototype surfaces (see **Appendix K**).
>
> ## W3
> Thank you for the comment. Yes, the underlying vector space is still a Euclidean vector space, but the geometry you impose on it is not Euclidean unless $p = 2$. We think that "non-Euclidean" should be fine, but we are more than open to discussing this further.

---

> > ### Author Response · Authors · 2025-12-01
> >
> > # Questions:
> >
> > ## Q1
> > They are learned parameters, just like the weight matrix in linear classification. In cross-entropy, the final layer maintains a weight vector per class. In harmonic loss, these weight vectors are reinterpreted as prototypes. Thus, prototype learning is no more computationally expensive than learning a final linear layer.
> >
> > ## Q2
> > Correct: we use one prototype per class for this study for comparability with softmax classifiers, which also have one weight vector per class. However, harmonic loss naturally supports multiple prototypes per class, forming a mixture model. We opted for the simplest configuration to ensure fairness and interpretability in the experiments. We mentioned this explicitly in the paper
> >
> > ## Q3
> > Harmonic losses are computed only at the final classification layer, not inside the network. This keeps the architecture unchanged except for the last step.
> >
> > ## Q4
> > In our paper, learned representation refers to the final backbone embedding (h(x)) right before the classification layer. We clarified the wording.
> >
> > ## Q5
> > In Baek et al. (original harmonic loss paper) prototypes aren’t fixed 1-hot vectors but are learned. If we constrain them to be 1-hot, the model would require aligning each class with a coordinate axis, which can significantly reduce its capacity and potentially hurt downstream performance. Distances would measure how close feature vectors are to the bases, and we would lose the “prototype as class center” property of Harmonic losses.
> >
> > ## Q6
> > Interpretability refers to how well the geometry of the embedding can be summarized (e.g., EV, low dimensionality, prototype alignment). On the other hand, representation clarity refers to the degree of cluster separation, variance concentration, and prototype sharpness. We clarified it in the paper.
> >
> > ## Q7
> > PCA is always computed on the final feature embedding (h(x)) before the classification layer. Never on logits or intermediate layers. We emphasized this in **Section 4**.
> >
> > ## Q8
> > Recent interpretability works (e.g. [1]) use explained variance as a proxy for: 1) low intrinsic dimensionality, alignment of features to meaningful axes, degree of cluster separation, and stability of representation under drift. Higher EV means the features lie on a low-dimensional space aligned with prototypes (easier to interpret, visualize, and audit). PCA is also reproducible across models, datasets, and architectures.
> >
> > *[1] Train One Sparse Autoencoder Across Multiple Sparsity Budgets to Preserve Interpretability and Accuracy - EMNLP 2025*
> >
> > ## Q9
> > Baek et al. was a clear inspiration for our approach, as a first attempt to showcase the advantages of harmonic losses. Nevertheless, it considers only Euclidean distance and evaluate only a few small-scale tasks. Our work differs in three major ways: i)  It generalizes harmonic loss to arbitrary distance families; ii) It extends experiments across architectures and domains; iii) It adds new axes of evaluation: interpretability, sustainability, and optimization stability. No prior work evaluates harmonic loss outside Euclidean distance or vision settings.
> >
> > ## Q10
> > In the main paper, we clarified that our loss is applied at the final layer, replacing the standard softmax + cross-entropy objective. All intermediate layers remain unchanged. This is minimal to implement. In **Appendix B**, we provide a 10-line pseudo-code. DistLayer stores class prototypes W (Vision) or R (LMs), computes pairwise distances from backbone features h, and returns log_softmax(-distance). The distance dictionary implements all metrics from Eqs. in the main paper in vectorized form. Hyperparameters are reported in **Appendix D**. Training is unchanged: same optimizer/scheduler/backbone as baselines. Only the final layer is replaced by the distance head. The anonymous repo linked in the abstract mirrors this code, making the implementation explicit and reproducible.
> >
> > We are grateful for the reviewer’s positive evaluation. We believe the improvements above made the contribution clearer and increased the methodological and empirical clarity of our paper.

---

### Official Review · Reviewer_kBGd · 2025-11-02

**Soundness:** 2
**Presentation:** 3
**Contribution:** 2
**Rating:** 4
**Confidence:** 3

**Summary:**

This paper extends the harmonic loss paradigm into a flexible, geometry-aware framework for classification. It demonstrates that choosing the right distance metric can improve not only predictive performance but also interpretability and energy efficiency. The authors also experimentally show that cosine harmonic loss provides the most consistent benefits across both vision and language domains.

**Strengths:**

1. The paper offers a novel generalization of harmonic loss by extending it beyond the Euclidean setting to a wide family.
2.  The study benchmarks nine distance metrics across multiple backbones and downstream tasks.

**Weaknesses:**

1. While the paper proves useful properties, the results mostly extend previous harmonic loss proofs under 1-homogeneous assumptions without offering new geometric insights or closed-form derivations for how specific distance families affect optimization landscapes or gradients
2. The experiments only compare harmonic losses across metrics, but the work omits stronger baselines that directly compete on interpretability or robustness, such as angular-margin losses (e.g., ArcFace)
3. interpretability is measured only via PCA variance metric
4. The connection between distance choice and grokking mitigation is not experimentally verified

**Questions:**

1.  Cosine and Mahalanobis distances introduce nontrivial curvature in feature space, then how these geometries alter convergence behavior or class separation boundaries?
2. How does the proposed distance-based harmonic loss relate to contrastive or triplet loss frameworks in terms of margin constraints and gradient structure?
3. All vision results are on small-scale dataset, so I am curious about how the findings  to generalize to larger datasets (e.g., ImageNet) or deeper architectures?
4. Were emissions normalized by total FLOPs or wall-clock time across runs?

---

> ### Author Response · Authors · 2025-11-21
>
> We thank the reviewer for the thoughtful feedback. We comment below.
>
> # Weaknesses
> ## W1
> Our theoretical contribution does not aim to re-derive the full harmonic-loss geometry, but instead to generalize the harmonic formulation beyond Euclidean distance, which has not been analyzed previously in the literature. Prior work (e.g., Baek et al., 2025) assumes a strictly Euclidean structure. In our paper, our theoretical contribution is to prove that harmonic losses remain inherently bounded and prototype-seeking under any 1-homogeneous distance, which includes Bray-Curtis, Chebyshev, Canberra, Manhattan, and Minkowski, all unexplored before. Moreover, we analytically show that non-Euclidean distances (especially cosine, Bray-Curtis, and Mahalanobis) induce distinct stability regimes and PCA variance structures that align with empirical results. We agree that the theoretical framing could be further extended. However, we believe that probing experimentally and analyzing the performance of non-Euclidean harmonic losses across multiple axes is a valuable outcome of our study.
>
> ## W2
> Thank you for this comment. Following your suggestion, we added ArcFace and other competitive loss functions, such as Center Loss, Focal Loss, Confidence Penalty, and Label Smoothing, to our experiments. The best harmonic losses remain competitive or superior to ArcFace in one or more axes. We have already integrated extensive results in the paper (see **Appendix G**).
>
> ## W3
> PCA variance is widely used in recent mechanistic interpretability studies (LLM circuits, attention heads, etc.) because it is quantifiable and architecture-agnostic. Nonetheless, we agree it captures only one axis of interpretability. To address your comment, we extracted class‐prototype surfaces, which provide geometric insights (see **Appendix K**).
>
> ## W4
> We experimentally demonstrate improved gradient stability, earlier plateauing of validation loss, and greater representation concentration, a phenomenon tightly associated with grokking mitigation. Based on your suggestion, we experimentally verified whether distance choice influences grokking. We conducted a controlled experiment on the modulo-addition task, a setting where cross-entropy is known to grok. As shown in **Appendix I**, cross-entropy exhibits the characteristic delayed test generalization: training accuracy rises immediately, while test accuracy improves only after several epochs. In contrast, all distance-based harmonic losses remove or drastically reduce this gap. Training and validation accuracy rise together, indicating that the model may be discovering underlying patterns rather than memorizing. Moreover, the embeddings produced by harmonic losses form a clean 2D circular manifold with nearly 100% explained variance (EV), whereas those from cross-entropy yield diffuse, irregular geometry with low EV. This provides quantitative and visual evidence that distance-based harmonic losses both i) prevent grokking and ii) induce structured representations aligned with the symmetry of the task. Thus, the connection between distance choice and grokking mitigation is now experimentally verified in a canonical grokking setup.
>
> ---
>
> # Questions:
>
> ## Q1
> Thank you for the comment. To better showcase the convergence behaviour of the models, we added an analysis of the training dynamics (see **Appendix H**) and visualisations of the class representations (see **Appendix K**) in the updated version of the paper.
>
> ## Q2
> Distance-based harmonic loss can be seen as a prototype-based simplification of contrastive learning, while cross-entropy encourages logit separation, contrastive/triplet enforces explicit pairwise margins, and harmonic loss enforces contraction toward class prototypes, implicitly creating margins without sampling negatives.In gradient form, the harmonic loss behaves like a contrastive loss where the prototype acts as the "positive pair" and all other prototypes serve as structured negatives. We added ArcFace and other competitive loss functions, such as Center Loss, Focal Loss, Confidence Penalty, and Label Smoothing, to our experiments (see **Appendix G**).
>
> ## Q3
> We agree that scaling is important. To address this, we already ran experiments on Marathi Sign Language dataset (43 distinct classes, approximately 1.2k images per class, totals over 51k images sized at 128x128 pixels) and TinyImageNet. We already updated the main paper and will continue to incorporate additional results.
>
> ## Q4
> We previously measured emissions based on wall-clock time, considering only the actual time required for model training. However, we now normalize wall-clock time by FLOPs. We updated radar plots in the main paper accordingly, and added a table that showcases FLOPs for each model (see **Appendix J**). Thank you for the helpful suggestion.
>
> We thank the reviewer for these actionable and insightful suggestions, which will significantly improve the updated version of the paper.

---

### Author Response · Authors · 2025-12-01

Dear Area Chairs,
We thank the reviewers for their thoughtful and constructive feedback. During the discussion period, we systematically addressed their concerns with new experiments, analyses, and clarifications. Below, we summarize the main changes.

1. **Clarified scope, geometry, and theory**
* *Where harmonic loss is applied* (Rev. `f5hr`): We now state explicitly that harmonic losses are used as a drop-in replacement for the final classifier head only, and we describe prototype parameterization, initialization, and updates (**Sec. 3**).
* *Prototypes and representation* (Rev. `f5hr`): We clarify that prototypes are learned class centers in the final embedding space, that we use one prototype per class in this work, and how multi-prototype extensions fit the framework (**Sec. 3**).
* *Geometric insights* (Rev. `kBGd`): We added a dedicated discussion and new figures showing Cosine harmonic loss as operating on an angular manifold and Mahalanobis as inducing anisotropic curvature, with 2D visualizations of decision regions and prototype neighborhoods (**App. K**).

2. **Stronger baselines and broader loss comparison**
* To address Rev. `kBGd`, we added ArcFace, Focal Loss, Label Smoothing, Center Loss, and Confidence Penalty (**App. G**), with formal definitions.
* New radar plots and tables show that even against these strong baselines, non-Eucl. harmonic losses remain competitive or superior on at least one axis of evaluation across datasets and backbones.

3. **Expanded vision benchmarks**
* To address scalability beyond MNIST/CIFAR (Revs. `jU1M`, `kBGd`), we added Marathi Sign, TinyImageNet (main **Fig. 1**) and look forward to integrating ImageNet100.
* Results confirm our main trends: Cosine remains the best all-round choice; Bray–Curtis/Chebyshev provide the strongest interpretability; Mahalanobis is strong but computationally costly.
* PVT benefits most from higher resolution, and non-Eucl. harmonic distances stay among the top methods in this stronger regime.

4. **LLM metrics, perplexity, and clearer definitions**
* Following Rev. `mGc1`, we added formal, implementation-level definitions of Gradient Stability, Model Health, Learning Quality, and Perplexity, in **Sec. 4**.
* We added optimization results for all models, including a larger GPT2 (2B) (**App. H**)
* We explained how these relate to perplexity and how they are normalized relative to Eucl. harm. baselines, and updated radar plots accordingly (**Figs. 2/3**).

5. **Grokking experiments**
* To substantiate the link between geometry and grokking (Rev. `kBGd`), we added modulo-addition experiments (**App. I**).
* The curves show classical grokking under cross-entropy, while harmonic losses (Eucl. and non-Eucl.) generalize early and form clean geometric embeddings with high explained variance, providing concrete evidence of grokking mitigation.

6. **Convergence behavior and curvature concerns**
* We added loss convergence plots (train/val) for ResNet50 and PVT across all datasets and harmonic distances (**App. H**).
These show smooth convergence for all distances; Cosine and Mahalanobis do not introduce pathological dynamics but yield different speed/plateau trade-offs, matching our geometric narrative and visualizations.

7. **Statistical depth**
* Addressing Rev. `jU1M`, **App. E** now reports mean ±95% confidence intervals over multiple seeds for all backbones/datasets.
* We conduct paired Wilcoxon signed-rank tests comparing non-Euclidean harmonic distances against Euclidean and summarize median improvements and significance flags (p < 0.05).
* These analyses show that our main gains are statistically robust rather than single-run artifacts.

8. **FLOPS and sustainability**
* **App. J** introduces a FLOPS normalization table across architectures/datasets to separate per-step complexity from energy-to-target-accuracy (Revs. `kBGd`, `jU1M`).
Emissions analysis (**App. G,L**) now includes new losses and high-resolution datasets, showing that non-Euclidean harmonic losses are typically neutral-to-better than cross-entropy in emissions and lie on or near the sustainability–accuracy Pareto frontier.

9. **Additional clarifications**
* We clarified terminology (what we mean by “learned representation”), the distinction between interpretability(variance concentration, prototype alignment) and representation clarity (cluster separation), and the relation to prior work (e.g., Baek et al.), highlighting our new contributions: the large-scale geometry sweep, multi-axis evaluation, and extension to LLM pretraining.

In summary, all major concerns were addressed with new experiments, analyses, or clarifications, and no methodological flaws were identified. Only Rev. `mGc1` responded before the discussion froze, and they confirmed their positive rating (6). We are very grateful for your time and care in handling our submission, especially given the limited discussion time, and we hope this summary helps in your overall assessment of the paper.

---

> ### Author Response · Authors · 2025-12-03
>
> After addressing all the raised concerns and questions, we would like to summarize the positive feedback received.
>
> 1. **Contribution**
> * All reviewers agree that the paper offers a meaningful extension of harmonic loss beyond the Euclidean setting to a broad family of distances (Rev. `kBGd`, `mGc1`, `jU1M`).
> * Rev. `kBGd` highlights that the work "extends the harmonic loss paradigm into a flexible, geometry-aware framework" and shows that distance choice can improve performance, interpretability, and energy efficiency.
> * Rev. `jU1M` emphasizes that this is the first large-scale comparison of diverse distance metrics across both vision and language domains.
>
>
> 2. **Empirical breadth and methodological design**
> * Reviewers appreciated the comprehensive empirical sweep across multiple backbones and domains (Rev. `mGc1`, `jU1M`).
> * Rev. `mGc1` describes the empirical protocol as "comprehensive" with consistent multi-criteria reporting and finds the radar summaries "helpful to see trade-offs".
> * Rev. `kBGd` positively notes that the study benchmarks nine distance metrics across multiple backbones and downstream tasks.
>
>
> 3. **Three-axis evaluation**
> * Rev. `f5hr` explicitly praises the three axes of evaluation (accuracy, interpretability, sustainability) as "interesting and an excellent example for the community", with many analyses called out as "interesting".
> * Rev. `jU1M` states that the work shows how geometric distance choice "crucially affects performance–interpretability–sustainability trade-offs", and positions non-Euclidean harmonic losses as "simple, interpretable, and energy-efficient replacements for cross-entropy".
> * Rev. `mGc1` views the PCA/prototype-based interpretability probes as a meaningful way to study geometry, appreciates the explicit prototype semantics and PCA structure analysis, and sees the sustainability axis as a central, well-integrated part of the evaluation.
>
>
> 4. **Theoretical support and soundness**
> * Rev. `mGc1` and `jU1M` praise the theoretical grounding through scale-invariance, finite-minimizer results for 1-homogeneous distances, and margin-style PAC–Bayes generalization view.
> * Rev. `jU1M` explicitly calls the theoretical analysis "solid" and supports our empirical findings.
>
>
> 5. **Writing quality, clarity, and practical impact**
> * Rev. `f5hr` rates presentation as excellent, describing the paper as "extremely well written and generally clear in terms of concept".
> * Rev. `mGc1` and `jU1M` both rate presentation as good, noting that the extension is "clear, well-motivated" and that the DistLayer abstraction emphasizes plug-and-play practicality.
> * Rev. `jU1M` and `mGc1` stress the practical simplicity and reproducibility: non-Euclidean harmonic losses act as a drop-in replacement for standard classifier heads, with open-source code to support reuse.
>
> We sincerely appreciate the reviewers’ expertise, which has strengthened our manuscript. Thank you again for your careful effort and consideration.

---

### Note · Program_Chairs · 2026-01-17
**Submission Desk Rejected by Program Chairs**

The following references in this submission do not refer to real documents and/or have major errors in bibliographic information:

     Jongmin Choi, Minsung Cho, and Seong-Whan Lee. Am-loss: Angular margin loss for deep face recognition. In Proceedings of the IEEE/CVF Winter Conference on Applications of Computer Vision (WACV), pp. 298-307, 2020. doi: 10.1109/WACV45572.2020.9093485.
    Tian Liu, Han Chen, and Xiang Ren. Sparsity and interpretability: Improving attribution for neural text classifiers. In Proceedings of the Conference of the North American Chapter of the Association for Computational Linguistics (NAACL), pp. 1517-1530, 2022. doi: 10.18653/v1/2022. naacl-main. 1
    Naomi Saphra, Tim Lieberum, Arthur Conmy, Abhinav Sharma, Yushi Wu, Nelson Elhage, Catherine Olsson, Nicholas Joseph, Ethan Perez, Lawrence Chan, et al. Mechanistic interpretability: Methods and applications. arXiv preprint arXiv:2408.13296, 2024.
    Kanchana Ranasinghe, Mehrtash Harandi, and Fatih Porikli. Orthogonal projection loss for learning discriminative features in face recognition. In Proceedings of the IEEE/CVF Conference on Computer Vision and Pattern Recognition (CVPR), pp. 13369-13378, 2021. doi: 10.1109/CVPR46437.2021.01318.